# PROVABLE BENEFITS OF REPRESENTATIONAL TRANSFER IN REINFORCEMENT LEARNING

## ABSTRACT

We study the problem of representational transfer in RL, where an agent first pretrains in a number of source tasks to discover a shared representation, which is subsequently used to learn a good policy in a target task. We propose a new notion of task relatedness between source and target tasks, and develop a novel approach for representational transfer under this assumption. Concretely, we show that given a generative access to source tasks, we can discover a representation, using which subsequent linear RL techniques quickly converge to a near-optimal policy, with only online access to the target task. The sample complexity is close to knowing the ground truth features in the target task, and comparable to prior representation learning results in the source tasks. We complement our positive results with lower bounds without generative access, and validate our findings with empirical evaluation on rich observation MDPs that require deep exploration.

## 1 INTRODUCTION

Leveraging historical experiences acquired in learning past skills to accelerate the learning of a new skill is a hallmark of intelligent behavior. In this paper, we study this question in the context of reinforcement learning (RL). Specifically, we consider a setting where the learner is exposed to multiple tasks and ask the following question:

*Can we accelerate RL by sharing representations across multiple related tasks?*

There is rich empirical literature which studies multiple approaches to this question and various paradigms for instantiating it. For instance, in a multi-task learning scenario, the learner has simultaneous access to different tasks and tries to improve the sample complexity by sharing data across them (Caruana, 1997). Other works study a transfer learning setting, where the learner has access to multiple source tasks during a *pre-training* phase, followed by a target task (Pan and Yang, 2009). The goal is to learn features and/or a policy which can be quickly adapted to succeed in the target task. More generally, the paradigms of meta-learning (Finn et al., 2017), lifelong learning (Parisi et al., 2019) and curriculum learning (Bengio et al., 2009) also consider related questions.

On the theoretical side, questions of representation learning have received an increased recent emphasis owing to their practical significance, both in supervised learning and RL settings. In RL, a limited form of transfer learning across multiple downstream reward functions is enabled by several recent reward-free representation learning approaches (Jin et al., 2020a; Zhang et al., 2020; Wang et al., 2020; Du et al., 2019; Misra et al., 2020; Agarwal et al., 2020; Modi et al., 2021). Inspired by recent treatments of representation transfer in supervised (Maurer et al., 2016; Du et al., 2020) and imitation learning (Arora et al., 2020), some works also study more general task collections in bandits (Hu et al., 2021; Yang et al., 2020, 2022) and RL (Hu et al., 2021; Lu et al., 2021). Almost all these works study settings where the representation is *frozen* after pre-training in the source tasks, and a linear policy or optimal value function approximation is trained in the target task using these learned features. This setting, which we call *representational transfer*, is the main focus of our paper.

A crucial question in formalizing representational transfer settings is the notion of similarity between source and target tasks. Prior works in supervised learning make the stringent assumption that the covariates $x$ follow the same underlying distribution in all the tasks, and only the conditional $P(y|x)$ can vary across tasks (Du et al., 2020). This assumption does not nicely generalize to RL settings, where state distributions are typically policy dependent, and prior attempts to extend this assumption to RL (Lu et al., 2021) result in strong assumptions on the learning setup. Other works (Hu et al.,

2021; Yang et al., 2020, 2022) focus on linear representations only, which limits the expressivity of the feature maps, and does not adequately represent the empirical literature in the field.

**Our contributions.** In this context, our work makes the following contributions:

- We propose a new linear span assumption of task relatedness for representational transfer, where the target task dynamics can be expressed as a (state-dependent) linear span of source task dynamics, in addition to the dynamics being low-rank under a shared representation. We give examples captured by this assumption, and it generalizes all prior settings for representational transfer in RL. We do not make any linearity assumptions on our feature maps.
- When we have *generative access* to source tasks, we provide a novel algorithm REPTRANSFER that successfully pretrains a representation for downstream *online learning in any target task* (i.e., no generative access in target task) satisfying the linear span assumption, when the source tasks satisfy a common latent reachability assumption. The regret bound of learning in the target task is close to that of learning in a linear MDP equipped with the ground truth features, the strongest possible yardstick in our setup. The additional terms in our regret largely arise out of the distributional mismatch between source and target tasks, which is expected. We complement the theory with an empirical validation of REPTRANSFER on the challenging rich observation combination lock benchmarks (Misra et al., 2020), confirming our theoretical findings.
- Without generative access to source tasks, we show the statistical hardness of representational transfer under the linear span assumption, and confirm this hardness in our empirical evaluation. We show that an additional assumption that every *observed state* is reachable in every source task is sufficient for allowing fully online learning in source tasks.

The new task relatedness assumption, reward-free learning result for low-rank MDPs and our analysis of LSVI-UCB under average case misspecification may be of independent interest.

## 2 RELATED WORK

In this section, we focus on survey related works that obtained concrete PAC or regret guarantees, and we defer a discussion of the empirical literature to the appendix.

**Multi-task and Transfer Learning in Supervised Learning.** The theoretical benefit of representation learning are well studied under conditions such as the i.i.d. task assumption (Maurer et al., 2016) and the diversity assumption (Du et al., 2020; Tripuraneni et al., 2020). Many works below successfully adopt the frameworks and assumptions to sequential decision making problems.

**Multi-task and Transfer Learning in Bandit and small-size MDPs.** Several recent works study multi-task linear bandits with linear representations ($\phi(s) = A s$ with unknown $A$) (Hu et al., 2021; Yang et al., 2020, 2022). The techniques developed in these works crucially rely on the linear structure and can not be applied to nonlinear function classes. Lazaric et al. (2013) study spectral techniques for online sequential transfer learning. Brunskill and Li (2013) study multi-task RL under a fixed distribution over finitely many MDPs, while Brunskill and Li (2014) consider transfer in semi-MDPs by learning options. Lecarpentier et al. (2021) consider lifelong learning in Lipschitz MDP. All these works consider small size tabular models while we focus on large-scale MDPs.

**Multi-task and Transfer Learning in RL via representation learning.** Beyond tabular MDPs, Arora et al. (2020) and D'Eramo et al. (2019) show benefits of representation learning in imitation learning and planning, but do not address exploration. Lu et al. (2021) study transfer learning in low-rank MDPs with general nonlinear representations, but make a generative model assumption on both the source tasks and the target task, along with other distributional and structural assumptions. We do not require generative access to the target task and make much weaker structural assumptions on the source-target relatedness. Recently and independently, Cheng et al. (2022) also studied transfer learning in low-rank MDPs in the online learning setting, identical to the setting we study in Section 5. However, their analysis relies on an additional assumption that bounds the point-wise TV error with the population TV error, which we show is in fact not necessary (details in Appendix C).

**Efficient Representation Learning in RL.** Even in the single task setting, efficient representation learning is an active area witnessing recent advances with exploration (Agarwal et al., 2020; Modi et al., 2021; Uehara et al., 2021; Zhang et al., 2022) or without (Ren et al., 2021). Other papers study feature selection (e.g. Farahmand and Szepesvári, 2011; Jiang et al., 2015; Pacchiano et al., 2020; Cutkosky et al., 2021; Lee et al., 2021; Zhang et al., 2021) or sparse models (Hao et al., 2021a,b).

## 3 PRELIMINARIES

**Notations:** We denote total variation distance of $P_1$ and $P_2$ by $\|P_1 - P_2\|_{TV}$. Given a vector $a$, we define $\|a\|_B = \sqrt{a^\top B a}$. $c_0, c_1, \cdots$ are universal constants. We use $a \lesssim b$ to mean $a \leq Cb$ for some universal constant $C$. Also, $[K] = \{1, \ldots, K\}$ and $\lambda_{\min}(A)$ is the smallest eigenvalue of matrix $A$. Please see Table 1 for a full list of notations and Table 2 for a list of algorithms.

**Low-rank Markov Decision Processes:** We consider a finite-horizon episodic Markov Decision Process $\mathcal{M} = \langle H, \mathcal{S}, \mathcal{A}, \{P_h^\star\}_{0:H-1}, \{r_h\}_{0:H-1}, d_0 \rangle$, specified by the episode length $H$, state space $\mathcal{S}$, discrete action space $\mathcal{A}$ of size $A$, transition models $P_h^\star : \mathcal{S} \times \mathcal{A} \to \Delta(\mathcal{S})$, reward functions $r_h : \mathcal{S} \times \mathcal{A} \to [0, 1]$, and an initial distribution $d_0 \in \Delta(\mathcal{S})$. Occasionally, we use $P^\star$ to denote $\{P_h^\star\}_{0:H-1}$ and $r$ to denote $\{r_h\}_{0:H-1}$. Starting from an initial state $s_0 \sim d_0$, an agent chooses actions $a_h$ at step $h$ upon observing the state $s_h$, receives a reward $r_h(s_h, a_h)$, and transitions to $s_{h+1} \sim P_h^\star(\cdot|s_h, a_h)$. We assume that $r$ and $d_0$ are known. Given a policy $\pi : \mathcal{S} \to \Delta(\mathcal{A})$ and transition kernel $P$, we use the notation $\mathbb{E}_{\pi,P}[\cdot]$ to denote expectation under the distribution of trajectories when $\pi$ is executed in $P$. The value function $V_{P,r;h}^\pi(s) = \mathbb{E}_{\pi,P}\left[\sum_{\tau=h}^{H-1} r_\tau(s_\tau, a_\tau) \mid s_h = s\right]$ gives the expected reward-to-go of $\pi$ under the MDP with transitions $P$, rewards $r$, starting at state $s$ in step $h$. Similarly, we define the $Q$ function $Q_{P,r;h}^\pi(s, a) := r_h(s, a) + \mathbb{E}_{s' \sim P_h(\cdot|s,a)} V_{P,r;h+1}^\pi(s')$. The expected total reward of a policy $\pi$ under transitions $P$ and rewards $r$ is denoted as $V_{P,r}^\pi := \mathbb{E}_{s_0 \sim d_0} V_{P,r;0}^\pi(s_0)$. We define the state-action occupancy distribution $d_{P;h}^\pi(s, a)$ as the probability of $\pi$ visiting $(s, a)$ at time step $h$ under $\pi$ and $P$. $d_{P;h}^\pi(s)$ is the marginal state visitation, which is equal to $\sum_{a \in \mathcal{A}} d_{P;h}^\pi(s, a)$.

We study low-rank MDPs defined as follows (Jiang et al., 2017; Agarwal et al., 2020). The conditions on the upper bounds of the norm of $\phi^\star, \mu^\star$ are just for normalization.

**Definition 3.1** (Low-rank MDP) *A transition model $P_h^\star : \mathcal{S} \times \mathcal{A} \to \Delta(\mathcal{S})$ admits a low rank decomposition with rank $d \in \mathbb{N}$ if there exist two unknown embedding functions $\phi_h^\star : \mathcal{S} \times \mathcal{A} \mapsto \mathbb{R}^d$, $\mu_h^\star : \mathcal{S} \mapsto \mathbb{R}^d$ such that $\forall s, s' \in \mathcal{S}, a \in \mathcal{A} : P_h^\star(s' \mid s, a) = \mu_h^\star(s')^\top \phi_h^\star(s, a)$, where $\|\phi_h^\star(s, a)\|_2 \leq 1$ for all $(s, a)$ and for any function $g : \mathcal{S} \to [0, 1], \left\|\int g(s) \mathrm{d}\mu_h^\star(s)\right\|_2 \leq \sqrt{d}$. An MDP is a low rank MDP if $P_h^\star$ admits such a low rank decomposition for all $h = 0, 1, \ldots, H - 1$.*

Low-rank MDPs capture the latent variable model (Agarwal et al., 2020) where $\phi^\star(s, a)$ is a distribution over a discrete latent state space $\mathcal{Z}$, and the block-MDP model (Du et al., 2019) where $\phi^\star(s, a)$ is a one-hot encoding vector. Note that the linear MDP model (Yang and Wang, 2020; Jin et al., 2020b) assumes a *known* $\phi^\star$, which significantly simplifies the algorithm design.

**Transfer Learning:** In contrast to the classic single-task learning setting, in this paper, we explore the setting of *transfer learning*, where learning consists of two phases: (1) the **pre-training phase** where the agent interacts with $K - 1$ source tasks with dynamics $P_k^\star$, and (2) the **deployment phase** where the agent is deployed into the $K$-th target task and no longer has access to the source tasks. The performance is measured mainly by the regret incurred in the target task upon deployment, while we also desire small sample complexity in the source tasks. We denote $d_{k;h}^\pi := d_{P_k^\star;h}^\pi$.

In order for the pre-training phase to help with learning in the target task, we must make assumptions on the connections between tasks. In this work, we make the following fundamental structural assumption on all the tasks at hand, namely that they share the same underlying representation $\phi^\star$.

**Assumption 3.1** (Common representation) *We assume that all tasks $P_k^\star$ are low-rank MDPs with a shared representation $\phi_h^\star(s, a)$ but distinct $\mu_{k;h}^\star(s')$, that is $P_{k;h}^\star(s'|s, a) = \phi_h^\star(s, a)^\top \mu_{k;h}^\star(s')$.*

**Assumption 3.2** (Realizability) *For any source task $k \in [K - 1]$ and any $h \in [H]$, we assume that the agent has access to realizable function classes $\Phi_h$ and $\Upsilon_{k;h}$, such that $\phi_h^\star \in \Phi_h$ and $\mu_{k;h}^\star \in \Upsilon_{k;h}$. For normalization, we assume that for all $k, h$, all $\phi \in \Phi_h$ satisfy $\|\phi(s, a)\|_2 \leq 1$, and for all $\mu \in \Upsilon_{k;h}$ and any function $g : \mathcal{S} \to [0, 1], \left\|\int g(s) \mathrm{d}\mu_h^\star(s)\right\|_2 \leq \sqrt{d}$.*

Assumption 3.2 is a standard realizability condition in the function class which is made in almost all prior works on RL with linear and nonlinear function approximation.

Under Assumption 3.1, we expect the agent to learn a good representation during the pre-training phase by interacting with the source tasks, and then learn a good policy for the target task using this representation in the deployment phase. In order to realize this intuition though, we need two additional assumptions which we describe next.

**Assumptions for representational transfer.** In addition to standard assumptions in the source tasks, we make the following structural and relatedness assumptions on the source and target tasks.

**Assumption 3.3** (Feature reachability in the source tasks) *We assume that* $\psi$ $:=$ $\min_{k\in[K-1],h\in[H]}\max_\pi \lambda_{\min}\left(\mathbb{E}_{\pi,P_k^\star}[\phi_h^\star(s_h,a_h)\phi_h^\star(s_h,a_h)^\top]\right)$ *is strictly positive.*

Assumption 3.3 intuitively requires that no subspace in $\mathbb{R}^d$ is unreachable in the source tasks, as otherwise it's impossible to guarantee the quality of the learned representation in that subspace which may contain high rewards in the target task. Note that no reachability is required in the target task.

The next assumption quantifies the relatedness of the target task and the source tasks.

**Assumption 3.4** (Relatedness: Point-wise Linear span) *For any* $h \in [H]$ *and* $s' \in \mathcal{S}_{h+1}$, *there is a vector* $\alpha_h(s') \in \mathbb{R}^{K-1}$ *such that* $\mu_{K;h}^\star(s') = \sum_{k=1}^{K-1}\alpha_{k;h}(s')\mu_{k;h}^\star(s')$, *with* $\alpha_{\max} = \max_{h;k,s'\in\mathcal{S}}|\alpha_{k;h}(s')|$ *and* $\bar\alpha = \max_h \sum_{k=1}^{K-1}\max_{s'\in\mathcal{S}}|\alpha_{k;h}(s')|$.

Assumption 3.4 ensures that if $s'$ is reachable from an $(s,a)$ pair in the target task, then it must be reachable from the same $(s,a)$ pair in at least one of the source tasks. This intuitively is also necessary for transfer learning, as $s'$ could be a high rewarding state in the target. A special case is the convex combination, i.e., for any $s'\in\mathcal{S}, h$, $\alpha_{k;h}(s') = p_k$ with $p_k\geq 0, \sum_k p_k = 1$, which implies $\alpha_{\max}\leq 1$ and $\bar\alpha = 1$. On the other hand, if the target task largely focuses on observations quite rare under the source tasks, then $\bar\alpha$ can grow large, and this is unavoidable in a transfer learning setting.

We remark that unlike prior work (Du et al., 2020; Tripuraneni et al., 2020; Lu et al., 2021), we do not make any assumption on the data generating distribution in either the source or the target tasks. Instead, our approach is end-to-end, i.e., we collect our own data from scratch for representation learning by doing strategic exploration in the source tasks (see Section C for a detailed discussion). In fact, in Theorem D.1 we show that the above assumptions do not permit successful transfer in the supervised learning setting, thus establishing an interesting separation between supervised and reinforcement learning.

We conclude this section with a couple of examples where our assumptions are satisfied.

**Example 3.1** (Mixture of source tasks). Perhaps the simplest example of our assumptions is where $P_K^\star(s'|s,a) = \sum_{k=1}^{K-1}\alpha_k P_k^\star(s'|s,a)$ with the coefficients independent of $s'$, $\alpha_k\geq 0$ and $\sum_{k=1}^{K-1}\alpha_k = 1$. Such mixtures of base models have been considered in several prior works (Modi et al., 2020; Ayoub et al., 2020). While prior works study the case of arbitrary, but *known* base models, we instead allow structured, unknown base models with a shared representation. Here $\bar\alpha = 1$.

**Example 3.2** (Block MDPs with shared latent dynamics). In this example, each MDP $P_k^\star$ is a Block MDP (Du et al., 2019) with a shared latent space $\mathcal{Z}$ and a shared decoder $\psi^\star : \mathcal{S}\to\mathcal{Z}$. In a block MDP, given state action pair $(s,a)$, the decoder $\psi^\star$ maps $s$ to a latent state $z$, the next latent state is sampled from the latent transition $z'\sim P(\cdot|z,a)$, and the next state is generated from an emission distribution $s'\sim o(\cdot|z')$. Recall that $o(s'|z') > 0$ at only one $z'\in\mathcal{Z}$ for any $s'\in\mathcal{S}$ for a block MDP. We assume that the latent transition model $P(z'|z,a)$ is shared across all the tasks, but the emission process differs across the MDPs. For instance, in a typical navigation example used to motivate Block MDPs, the latent dynamics might correspond to navigating in a shared 2-D map, while emission distributions capture different wall colors or lighting conditions across multiple rooms. Then Assumption 3.3 requires that the agent can visit the entire 2-D map, while Assumption 3.4 requires that the color/lighting conditions of the target task resemble that of at least one source task. The coefficients $\alpha$ for any $s'$ are non-zero on the source tasks which can generate that observation.

## 4 TRANSFER LEARNING WITH GENERATIVE ACCESS TO SOURCE TASKS

We first study a setting where we assume generative model access to the source tasks, while having *only online access to the target task*.

**Assumption 4.1** (Generative access to the source tasks) *We assume that we have access to generative models for the $K-1$ source tasks. Specifically, for any $P_k^\star$ with $k\in[K-1]$, we can query any $(s_h,a_h)$ pair, and the generative model will return a next state sample $s_{h+1}\sim P_{k;h}^\star(s_h,a_h)$.*

Having generative model access is not unrealistic, especially in applications where a high-quality simulation environment is available. Generative access also does not trivialize the challenge of efficient exploration in the source tasks, since there are a potentially infinite number of states and the ground truth representation $\phi^\star$ is unknown. Prior works using generative access typically

---

**Algorithm 1** Transfer learning with generative access (REPTRANSFER)

---

**PRE-TRAINING PHASE**
**Input:** exploratory policies $\{\pi_k\}_{k=1}^{K-1}$, size of cross-sampled datasets $n$, failure probability $\delta$.

1: **for** *task pairs $i, j$*, i.e. for all $i, j \in [K-1]$ s.t. $i \neq j$ **do**       ▷ cross sampling procedure
2:     For each $h \in [H-1]$, sample dataset $\mathcal{D}_{ij;h}$ containing $n$ i.i.d. $(s, a, s')$ tuples sampled as:

$$(\tilde{s}, \tilde{a}) \sim d_{i;h-1}^{\pi_i}, s \sim P_{j;h-1}^{\star}(\cdot | \tilde{s}, \tilde{a}), a \sim \mathcal{U}(\mathcal{A}), s' \sim P_{i;h}^{\star}(\cdot | s, a). \tag{1}$$

3: $\forall h \in [H-1]$, learn $\widehat{\phi}_h =$ Multi-task REPLEARN($\{\cup_{j \in [K-1]} \mathcal{D}_{kj;h}\}_{k \in [K-1]}$). (Algorithm 3)
**DEPLOYMENT PHASE Additional Input:** number of deployment episodes $T$.

1: Set $\beta = H\sqrt{d} + \bar{\alpha}dH\sqrt{\log(dHT/\delta)}$.
2: Run LSVI-UCB $\left( \{\widehat{\phi}_h\}_{h=0}^{H-1}, r = r_K, T, \beta \right)$ in the target task $P_K^{\star}$ (Algorithm 6).

---

require either a known representation $\phi^{\star}$ (so that one can perform D-optimal design to construct an exploratory state-action distribution (Agarwal et al., 2019)), or directly assume access to a diverse state-action distribution which provides coverage and from which one can sample. Neither $\phi^{\star}$ nor such a diverse sampling distribution is given in our case. We also note that in Section 5 we will show that without any additional assumptions, generative access in source tasks is necessary.

**Algorithm overview.** Given the above setup, now we present our algorithm REPTRANSFER, detailed in Algorithm 1. REPTRANSFER takes a representation learning approach to the transfer learning problem and operates in two phases. During the pre-training phase, REPTRANSFER performs reward-free exploration in each of the source tasks to learn task-specific policies $\pi_k$, which satisfies for all $h = 0, 1, ..., H-1$, that $\mathbb{E}_{\pi_k, P_k^{\star}} \left[ \phi_h^{\star}(s_h, a_h) \phi_h^{\star \top}(s_h, a_h)^{\top} \right] \succeq \lambda_{\min} I$ for some $\lambda_{\min} > 0$. We call such policies $\lambda_{\min}$-*exploratory* and will use this definition in Lemma 4.1 and Theorem 4.1. In Section 4 we present one particular algorithm that finds such $\pi_k$'s with a specific $\lambda_{\min}$, but our main analysis below is modular to the choice of the reward-free exploration algorithm.

Given the exploratory policies $\{\pi_k\}_{k \in [K-1]}$, REPTRANSFER collects a joint dataset across the source tasks using these policies and cross-sampling across pairs of tasks (1), and then learns a *single* representation $\widehat{\phi}$. In the deployment phase, REPTRANSFER runs optimistic least squares value iteration (Algorithm 6) using the learned representation $\widehat{\phi}$ in the target task[1].

**The cross sampling procedure and learning a shared representation.** We now describe the cross sampling procedure in detail. Given an exploratory policy $\pi_k$ for each source task $k$, the next step is to sample fresh data under a *cross-sampling procedure* using the generative model. Consider a particular $(k, j)$ pair of source environments. For each $h \in [H]$, we first sample $s_{h-1}, a_{h-1} \sim d_{k;h-1}^{\pi_k}$. Then, in the simulator of task $j$, we reset to $(s_{h-1}, a_{h-1})$ and perform a transition step to $s_h$, i.e., $s_h \sim P_{j;h-1}(s_{h-1}, a_{h-1})$. Then we reset the simulator for task $k$ to state $s_h$, sample a uniformly random action $a_h$ and then perform another transition to $s_{h+1}$, i.e., $s_{h+1} \sim P_k^{\star}(s_h, a_h)$. Such a procedure is only possible under the generative model setting. Intuitively, cross-sampling ensures that our training data contains all possible states that can be encountered in the target task, and failure modes without this can be found in the discussion following Theorem 5.1.

Given datasets, each of size $n$, collected using the cross-sampling procedure, we perform a Maximum Likelihood Estimation (MLE)-based representation learning procedure (Algorithm 3) jointly on all source tasks, with the goal of finding a representation function $\hat{\phi}_h$ that can predict the transition probability well across all source tasks. An existing MLE generalization analysis (Agarwal et al., 2020) guarantee that Multi-task REPLEARN (Algorithm 3) achieves the following total variation guarantee with probability at least $1 - \delta$:

$$\sum_{k=1}^{K-1} \mathbb{E}_{\nu_{k;h}} \left\| \widehat{\phi}_h(s, a)^{\top} \widehat{\mu}_{k;h}(\cdot) - \phi_h^{\star}(s, a)^{\top} \mu_{k;h}^{\star}(\cdot) \right\|_{TV}^2 \leq \zeta_N := \mathcal{O}\left( (\log(|\Phi|/\delta) + K \log(|\Upsilon|))/N \right). \tag{2}$$

---

[1]Given any dataset, $\{s, a, r, s'\}$ feature $\phi$, and reward $r$, LSVI learns a $Q$ function backward, i.e., at step $h$ via $\hat{w}_h = \arg\min_w \sum_{s,a,s'} (w^{\top}\phi(s, a) - \hat{V}_{h+1}(s'))^2 + \lambda \|w\|^2$ and set $\hat{V}_h(s) = \max_a(r(s, a) + \hat{w}_h^{\top}\phi(s, a)), \forall s$. UCB, short for Upper Confidence Bound, refers to an exploration bonus added to basic LSVI.

---

**Algorithm 2** REWARDFREE

---

1: **Input:** MDP $P^\star$ with online access, num. LSVI-UCB episodes $N_{\text{LSVI-UCB}}$, num. model-learning episodes $N_{\text{REWARDFREE}}$, failure probability $\delta$.
2: Learn model $\{\widehat{P}_h = (\widehat{\phi}_h, \widehat{\mu}_h)\}_{h=0}^{H-1}$ by running REWARDFREE REP-UCB (Algorithm 4) in $P^\star$ for $N_{\text{REWARDFREE}}$ episodes.
3: Set $\beta = dH\sqrt{\log(dHN_{\text{LSVI-UCB}}/\delta)}$.
4: Return $\rho = \text{LSVI-UCB}\left(\{\widehat{\phi}_h\}_{h=0}^{H-1}, r = 0, N_{\text{LSVI-UCB}}, \beta, \text{UNIFORMACTIONS} = \text{TRUE}\right)$ by simulating in the learned model $\widehat{P}$ (Algorithm 6). Note this step requires no samples from $P^\star$.

---

where $\nu_{k;h}$ denotes the generating distribution of inputted datasets $\mathcal{D}_{k;h}$, each with size $N$. For example, in REPTRANSFER we have $\mathcal{D}_{k;h} = \cup_{j=1}^{K-1} \mathcal{D}_{kj;h}$ and $N = n(K-1)$.

**Representational transfer.** Next we transfer this guarantee to the target task via Assumption 3.4, a key insight of our work. Specifically, we show that under cross sampling and Assumption 3.4, $\widehat{\phi}$ also linearly approximates the true transition in the target task well, under the occupancy measure of *any policy*. Remarkably, this holds before the agent has ever interacted with the target task.

**Lemma 4.1** *Suppose Assumption 3.4 and that for all source tasks $k \in [K-1]$ we have a $\lambda_{\min}$-exploratory policy $\pi_k$. Then, for any $\delta \in (0,1)$, learning features $\widehat{\phi}$ using the cross-sampling procedure of Algorithm 1 satisfies* (2) *w.p.* $1 - \delta$. *Furthermore, for any $h = 0, 1, ..., H - 1$, there exists $\widetilde{\mu}_h : \mathcal{S} \to \mathbb{R}^d$ such that for any function $g : \mathcal{S} \to [0,1]$, $\|\int g(s)\mathrm{d}\widetilde{\mu}_h(s)\|_2 \leq \bar{\alpha}\sqrt{d}$ and*

$$\sup_\pi \mathbb{E}_{\pi, P_K^\star} \left\| \widehat{\phi}_h(s_h, a_h)^\top \widetilde{\mu}_h(\cdot) - \phi_h^\star(s_h, a_h)^\top \mu_{K;h}^\star(\cdot) \right\|_{TV} \leq \varepsilon_{TV} := \sqrt{|\mathcal{A}|\alpha_{\max}^3 K\zeta_n/\lambda_{\min}}.$$

Lemma 4.1 implies that $\widehat{\phi}$ is a feature such that $P_K^\star$ is an *approximatly linear MDP* in $\widehat{\phi}$. Learning in approximately linear MDPs has been studied in Jin et al. (2020b), but under a much stronger $\ell_\infty$ error bound rather than the average misspecification here. Our result for downstream task learning under such an average model misspecification case presents the strongest result for learning in an approximately linear MDP, and the result might be of independent interest.

**Regret bound for REPTRANSFER.** Putting things together, we obtain the following regret bound for online learning in the target task using our learned representation $\widehat{\phi}$.

**Theorem 4.1** (Regret under generative source access) *Suppose Assumptions 3.1-3.4 and 4.1, and suppose the input policies $\pi_k$ are $\lambda_{\min}$-exploratory. Then, for any $\delta \in (0,1)$, w.p. $1 - \delta$, REPTRANSFER when deployed in the target task has regret at most $\widetilde{\mathcal{O}}\left(\bar{\alpha}H^2 d^{1.5}\sqrt{T\log(1/\delta)}\right)$, with at most $Kn$ generative accesses per source task, with $n = \mathcal{O}\left(\lambda_{\min}^{-1}A\alpha_{\max}^3 KT\left(\log\frac{|\Phi|}{\delta} + K\log|\Upsilon|\right)\right)$.*

Remarkably, Theorem 4.1 shows that with the pre-trained features, we achieve the same regret bound on the target task to the setting of linear MDP with known $\phi^\star$ (Jin et al., 2020b), up to the additional $\bar{\alpha}$ factor. The scaling factor $\bar{\alpha}$ only depends on $\alpha$ itself and captures the hardness of transfer learning. For special cases such as convex combination, i.e., $\alpha$ is state-independent and $\alpha_h \in \Delta(K)$, then $\bar{\alpha} = 1$. In the worst-case, some dependence on the scale of $\alpha$ seems unavoidable as we can have a state $s'$ such that $\mu_K(s') = 1$ and $\mu_1(s') \ll 1$ with $\alpha_1(s') \gg 1$. This corresponds to a rarely observed state for the source task encountered often in the target, and our estimates of transitions involving this state can be highly unreliable if it is not seen in any other source, roughly scaling the error between target and source tasks as $|\alpha_1(s')|$. Obtaining formal lower bounds that capture a matching dependence on structural properties of $\alpha$ is an interesting question for future research.

**Reward-free exploration in the source environments.** So far we have assumed that we have a reward-free exploration black box algorithm that can provide an exploratory policy $\pi_k$ for each source task $k$. In this section, we give a detailed algorithm that achieves this goal.

Recall that our transfer learning algorithm, for source task $P_k^\star$, relies on an exploratory policy $\pi_k$ such that the empirical covariance with respect to $\phi_h^\star$ is lower bounded. This ensures good exploration in the underlying *ground truth feature space* in $P_k^\star$. Algorithm 2 achieves this goal by first invoking the REWARDFREE REP-UCB algorithm (Algorithm 4) to learn an estimated linear MDP model $\widehat{P}_k$ for $P_k^\star$. The REWARDFREE REP-UCB algorithm is a reward-free generalization of the recent

REP-UCB algorithm (Uehara et al., 2021) that perform reward-free exploration in low-rank MDPs and can be of independent interest. Subsequently, we perform reward-free exploration (e.g., using LSVI-UCB with zero reward) within each $\widehat{P}_k$ which involves no further environment interactions.

**Lemma 4.2** (Reward-free Exploration) *Fix any source task $k \in [K-1]$. Suppose Assumptions 3.2 and 3.3. Then, for any $\delta \in (0,1)$, w.p. $1 - \delta$, REWARDFREE (Algorithm 2) with $N_{\text{LSVI-UCB}} = \widetilde{\Theta}\left(A^3 d^6 H^8 \psi^{-2}\right)$ and $N_{\text{REWARDFREE}} = \widetilde{\mathcal{O}}\left(A^3 d^4 H^6 \log\left(\frac{|\Phi||\Upsilon|}{\delta}\right) N_{\text{LSVI-UCB}}^2\right)$ returns a $\lambda_{\min}$-exploratory policy $\pi_k$ where $\lambda_{\min} = \widetilde{\Omega}\left(A^{-3} d^{-5} H^{-7} \psi^2\right)$. The sample complexity here is $N_{\text{REWARDFREE}}$ episodes in the source task.*

To the best of our knowledge, Lemma 4.2 is the first result that finds a full-rank policy cover in the low-rank MDP setting, and might be of independent interest. Wagenmaker et al. (2022) recently obtained a related guarantee in the linear MDP setting with known features $\phi^\star$. Incorporating Lemma 4.2 and Theorem 4.1, we get our final sample complexity bound.

**Theorem 4.2** (Regret for REPTRANSFER with REWARDFREE subroutine) *Suppose Assumptions 3.1-3.4 and 4.1 hold, and let $\delta \in (0,1)$. Let $\{\pi_k\}_{k=1}^{K-1}$ be exploratory policies learned from running REWARDFREE on each source task with $N_{\text{LSVI-UCB}}$ and $N_{\text{REWARDFREE}}$ set as in Lemma 4.2. Then, w.p. $1 - \delta$, running REPTRANSFER with $\{\pi_k\}_{k=1}^{K-1}$ has regret in the target task of $\widetilde{\mathcal{O}}\left(\bar{\alpha} H^2 d^{1.5} \sqrt{T \log(1/\delta)}\right)$, with at most $\widetilde{\mathcal{O}}\left(A^4 \alpha_{\max}^3 d^5 H^7 K^2 T \psi^{-2} \left(\log(|\Phi|/\delta) + K \log |\Upsilon|\right)\right)$ generative accesses per source task.*

## 5 TRANSFER LEARNING WITH ONLINE ACCESS TO SOURCE TASKS

In the previous section, we show that efficient transfer learning is possible under very weak structural assumptions, with generative access to the source tasks. One natural question is whether transfer learning is possible with only online access to the source tasks. Somewhat surprisingly, we show that this is impossible without significantly stronger assumption.

**Theorem 5.1** (Impossibility Result) *Let $\mathcal{M}_K$ be a $K$-task multi-set that satisfies (1) all tasks are Block MDPs; (2) all tasks satisfy Assumption 3.3 and Assumption 3.4; (3) the latent dynamics are exactly the same for all source and target tasks. For any pre-training algorithm $\mathcal{A}$ which outputs a feature $\hat{\phi}$ by interacting with the source tasks $k \in [K-1]$, there exists $\{P_k^\star\}_{k \in [K]} \in \mathcal{M}_K$, such that with probability at least $1/2$, $\mathcal{A}$ will output a feature $\hat{\phi}$, such that for any policy taking the functional form of $\pi(s) = f\left(\{\hat{\phi}(s,a)\}_{a \in \mathcal{A}}, \{r(s,a)\}_{a \in \mathcal{A}}\right)$, we have $V_K^\star - V_K^\pi \geq 1/2$.*

The above theorem implies that a representation learned only from online access to source tasks does not enable learning in downstream tasks if the downstream task algorithm is restricted to use the representation as the only information of the state-action pairs (e.g., running LSVI-UCB with $\hat{\phi}$).

We briefly explain the intuition behind the above lower bound. For a Block MDP, for any $(s,a)$, we can model the ground-truth $\phi^\star$ as a one-hot encoding $e_{(z,a)}$ corresponding to the latent state-action pair $(z,a)$ with $z = \psi^\star(s)$ being the encoded latent state. The key observation here is that any permutation of $\phi^\star$ will also be a perfect feature in terms of characterizing the Block MDPs, since it corresponds to simply permuting the indices of the latent states. Therefore, without cross referencing, the agent could potentially learn different permutations in different source tasks, which would collapse in the target task. A precise constructive proof of Theorem 5.1 can be found in Section D.

Part of the reason that the above example fails is that each source task has its own observed subset of raw states, which permits such permutation to happen. In what follows, we show that, indeed, under an additional assumption on the reachability of raw states, a slight variant of the same algorithm (Algorithm 5 in Section G) can achieve the same regret with only online access to the source tasks.

**Assumption 5.1** (Reachability in the raw states) *For all source tasks $k \in [K-1]$, any policy $\pi$ and $h = 0, 1, ..., H-1$, we have $\inf_{s \in \mathcal{S}, a \in \mathcal{A}} d_{k;h}^\pi(s,a) \geq \psi_{raw} \lambda_{\min}\left(\mathbb{E}_{\pi, P_k^\star}\left[\phi_h^\star(s_h, a_h)\phi_h^\star(s_h, a_h)^\top\right]\right).$*

Assumption 5.1 implies that for each source task, any policy that achieves a full-rank covariance matrix also achieves global coverage across the raw state-action space. In addition, in order to apply importance sampling (IS) to transfer the TV error from source task to target task, we need to assume that the target task distribution has bounded density. This is true, for example, when $\mathcal{S}$ is discrete.

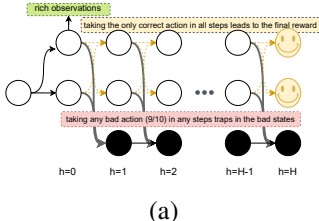

(a)

| | Source | O-REPTRANSFER | G-REPTRANSFER | Oracle | Target |
|---|---|---|---|---|---|
| Comblock | $\infty$ | 8006 (294.7) | 7790 (267.6) | 7048 (164.8) | 181450 (147600.2) |
| Comblock PO | $\infty$ | $\infty$ | 7764 (145.7) | 7336 (270.7) | 85000 (14469.8) |

(b)

Figure 1: **(a):** A visualization of the rich observation comblock environment. Latent states (white and black) emit continuous high-dimensional observations. The reward is sparse (only in white states at $H$). Each white state has 10 actions where one of them is a good action that leads to the next two white states while the other 9 lead to black states (good action for each white state is different). Once the agent is in black states, it stays stuck in black until the end of the episode. Thus, a random exploration strategy has an exponentially small probability of hitting the goal (i.e., roughly $(1/10)^H$). **(b) Top:** Number of episodes required to solve the target environment under the setting from Sec. 6. **(b) Bottom:** Number of episodes required to solve the target environment under the setting from Sec. 6. An algorithm solves the target task if it can achieve the optimal return (i.e., 1) for 5 consecutive iterations with 50 evaluation runs each. We include the mean and standard deviation (in the brackets) for 5 random seeds. $\infty$ denotes that an algorithm can not solve the target task within a fixed sample budget.

**Assumption 5.2** (Bounded density) *For all $(\pi, h, s, a)$, we have $d^{\pi}_{K;h}(s,a) \leq 1$.*

**Theorem 5.2** (Regret with online access) *Suppose Assumptions 3.1-3.4,5.1,5.2 hold. W.p. $1-\delta$, Algorithm 5 with appropriate parameters achieves a regret in the target $\widetilde{\mathcal{O}}\left(\bar{\alpha}d^{1.5}H^2\sqrt{T\log(1/\delta)}\right)$, with at most $\text{poly}\left(A, \alpha_{\max}, d, H, K, T, \psi^{-1}, \psi_{raw}^{-1}, \log(|\Phi||\Upsilon|/\delta)\right)$ online queries in the source tasks.*

Assumption 5.1 is satisfied in a Block MDP, when, for example, the emission function $o(s|z)$ satisfies that $\forall s, \exists z$, s.t., $o(s|z) \geq c$. That is, for any source task, any state in the state space can be generated by at least one latent state.

## 6 EXPERIMENTS

We investigate empirically the benefit of transfer learning under the Block MDP setting, on the challenging Rich Observation Combination Lock (comblock) benchmark. In this environment, one must recover the correct feature from rich observations and perform strategic exploration or otherwise pay for exponential sample complexity. We include a visual overview in Fig. 1(a). This environment is uniquely challenging since it requires strategic exploration and latent state discovery at the same time, which results in failures of common deep RL methods (Misra et al., 2020), and theoretical RL approaches based on linear function approximation (and kernels) (Zhang et al., 2022).

In particular, in this section we study the following questions: i) what are the benefits of representational transfer using multiple source tasks, and ii) whether generative access to source tasks is needed. We design two sets of experiments with various source and target environment configurations. We defer the details of the experiments (the design of vanilla comblock, hyperparameters, etc.) in Appendix J and include the essential design information in the following two sections.

**Baselines.** We denote **Source** as the smallest sample complexity of LSVI-UCB using learned features from any of the source tasks; **O-REPTRANSFER** as REPTRANSFER with only online access to the source tasks; **G-REPTRANSFER** as REPTRANSFER with generative access to the source tasks; **Oracle** as learning in the target task with ground truth features; and **Target** as running BRIEE (Zhang et al., 2022) — the SOTA Block MDP algorithm, in the target task with no pretraining.

**Comblock without partitioned observations** In this section we start off with an easier setting: we use 5 source tasks, each with horizon $H = 25$, 3 latent states per level and 10 actions. The latent transition dynamics are different for each source task, but notice that for comblock, the reachability assumption (c.r. Assumption 3.3) is always satisfied. The emission distribution of all the source and target tasks is identical, so Assumption 5.1 holds here. For the construction of the target task, for each timestep $h$, we choose the latent transition dynamics from one of the sources uniformly at random (thus Assumption 3.4 is satisfied). We record the number of episodes in the

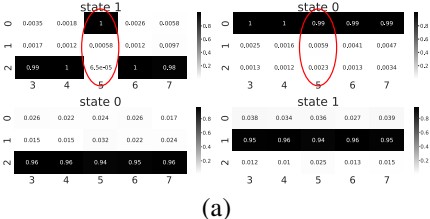 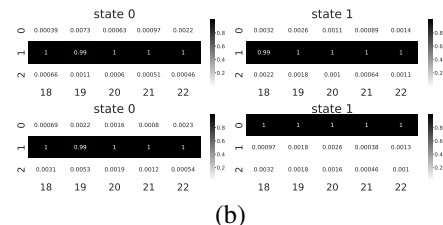

(a)                                                                   (b)

Figure 2: **(a):** Visualization of the decoder source (top) and G-REPTRANSFER (bottom). **(b):** Visualization of the decoder O-REPTRANSFER (top) and G-REPTRANSFER (bottom). For each baseline, The $h$-th column in the $i$-th image denotes the averaged decoded states from the 30 observations generated by latent state $z_{i,h}$, for $i \in \{0, 1, 2\}$ and $h \in [25]$, from the corresponding *target* environment. The optimal decoder should recover the latent states up to a permutation. In Fig a (top), note that the learned features in source task fail to solve the target because of the collapse at timestep 5: both observations from state 0 and 1 are mapped to state 0. Note in the source task where this feature is trained, such collapse can happen when state 0 and 1 have identical latent transition (for detailed discussion we refer to Misra et al. (2020)). In Fig b (top), REPTRANSFER with only online access learns an incorrect decoder when the source tasks' observation spaces are disjoint. This is because the learned feature can decode each source task with a different permutation.

target environment that each method takes to solve the target environment in Table. 1(b). We first observe that REPTRANSFER with either online or generative access can solve the target task (since. Assumption 5.1 holds). Second, we observe that directly applying the learned feature from any *single source task* does not suffice to solve the target environment. This is because the representation learned from a single source task may collapse two latent states into a single one during encoding (e.g., if two latent states at the same time step have exactly identical latent transitions). See the visualization of comparisons between the learned decoders from a single source task and REPTRANSFER in Fig. 2(a). Third, the result shows that REPTRANSFER saves order of magnitude of target samples compared with training in the target environment from scratch using the SOTA Block MDPs algorithm BRIEE. **This set of results verifies the empirical benefits of representation learning from multiple tasks, i.e., resolves ambiguity and speeds up downstream task learning**.

**Comblock with partitioned observation space (Comblock-PO)**   In this section, following the intuition of our lower bound (Theorem 5.1), we construct a setting where the supports of the emission distributions from each task are completely disjoint, while the emission distribution in the target task is a mixture of all source emissions and the latent dynamics are identical across tasks. Hence Assumption 3.4 holds while Assumption 5.1 fails. So we expect that an algorithm without generative access to source tasks will fail based on Theorem 5.1. Specifically, under disjoint emission supports, a representation can decode the latent state correctly for each source, but permute latent state labels across sources, causing decoding errors on the target. We record the number of target episodes for each method to solve the target task in Table. 1(b). We observe that indeed the online version fails while the generative version still succeeds. We show the visualizations of O-REPTRANSFER and G-REPTRANSFER in Fig. 2(b), and we note that the empirical results exactly match our theoretical results. **This ablation verifies that source generative model access is needed**.

## 7    CONCLUSION

We study representational transfer among low rank MDPs which share the same unknown representation. Under a reasonably flexible linear span task relatedness assumption, we propose an algorithm that provably transfers the representation learned from source tasks to the target task. The regret in target task matches the bound obtained with oracle access to the true representation, using only polynomial number of samples from source tasks. Our approach relies on the generative model access in source tasks, which we prove is not avoidable in the worst case under the linear span assumption. To complement the lower bound, we propose a stronger assumption on the conditions of the reachability in raw states, under which online access to source tasks suffices for provably efficient representation transfer. Finding modalities other than generative access which avoid the lower bound, and a more extensive empirical evaluation beyond the proof-of-concept experiments here are important directions for future research

**Reproducibility Statement:** For theory, we include detailed and complete proofs for all of our claims in the appendices. For empirics, we include anonymized code in the supplement. Please refer to Appendix. J for a comprehensive experimental setup and list of hyperparameters. All experiments were run on CPUs and no external datasets were used.

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

# Appendices

## A    NOTATIONS

Table 1: List of Notations

| | |
|---|---|
| $\mathcal{S}, \mathcal{A}, A$ | State and action spaces, and $A = \|\mathcal{A}\|$. |
| $\Delta(S)$ | The set of distributions supported by $S$. |
| $\lambda_{\min}(A)$ | Smallest eigenvalue of matrix $A$. |
| $e_j$ | One-hot encoding of $j$, i.e. 0 at each index except the one corresponding to $j$. Length of vector implied from context. |
| $(x)_{\leq y}$ | $\min\{x, y\}$. |
| $H$ | Episode length of MDPs, a.k.a. time horizon. We index steps as $h = 0, 1, ..., H-1$. |
| $K$ | There are a total of $K$ tasks. The first $k = 1, ..., K-1$ are source tasks, and the $K$-th task is the target task. |
| $d$ | dimension of the low-rank MDP, i.e. dimension of $\phi^\star$. |
| $P_{k;h}^\star$ | Ground truth transition at time $h$ for task $k$. |
| $r_h$ | Reward function of the target task (i.e. task $K$). |
| $\mathbb{E}_{\pi,P}[\cdot]$ | Expectation under the distribution of trajectories when $\pi$ is executed in $P$. We sometimes omit $P$ when the MDP is clear from context. |
| $d_{P;h}^\pi$ | Occupancy distribution of $\pi$ under transitions $P$ at time $h$. |
| $d_{k;h}^\pi$ | Occupancy distribution for the $k$-th task, i.e. $d_{P_k^\star;h}^\pi$. |
| $\phi_h^\star(s,a)$ | Embedding function for $(s,a)$ at time $h$. |
| $\Phi_h$ | Realizable function class for $\phi_h^\star$. |
| $\|\Phi\|$ | Defined as $\max_h \|\Phi_h\|$. |
| $\mu_{k;h}^\star(s')$ | Emission embedding function for $s'$ at time $h$ for environment $k$. |
| $\Upsilon_{k;h}$ | Realizable function class for $\mu_{k;h}^\star$. |
| $\|\Upsilon\|$ | Defined as $\max_{k;h} \|\Upsilon_{k,h}\|$. |
| $\psi$ | Feature reachability in the source task (Assumption 3.3). |
| $\alpha_{\max}, \bar{\alpha}$ | Constants defined in point-wise linear span assumption (Assumption 3.4). |
| $\psi_{raw}$ | Raw states reachability parameter (Assumption 5.1) |

Table 2: List of Algorithms

| | |
|---|---|
| REPTRANSFER (Algorithm 1) | Transfer learning with generative access. |
| REWARDFREE (Algorithm 2) | Reward Free exploration in low-rank MDP. |
| Multi-task REPLEARN (Algorithm 3) | Multi-task Maximum Likelihood (MLE) Representation Learning. |
| REWARDFREE REP-UCB (Algorithm 4) | Modified REP-UCB for reward-free model learning. |
| REPTRANSFER for online access (Algorithm 5) | Simplified version of REPTRANSFER if we have reachability under raw states (see Theorem 5.2). |
| LSVI-UCB (Algorithm 6) | Optimistmic Least Squares Value Iteration (with bonus). |

# B   ADDITIONAL RELATED WORKS

The idea of learning transferable representation has been extensively explored in the empirical literature. Here we don't intend to provide a comprehensive survey of all existing works on this topic. Instead, we discuss a few representative approach that may be of interest.

Towards transfer learning across different environments, progressive neural network (Rusu et al., 2016) is among the first neural-based attempt to learning a transferable representation for a sequence of downstream tasks that tries to overcome the challenge of catastrophic forgetting. It maintains the learned neural models for all previous tasks and introduce additional connections between the network of the current tasks to those of prior tasks to allow information reuse. However, a drawback common to such an approach is that the network size grows linearly with the number of tasks. Other approaches include directly learning a multi-task policy that can perform well on a set of source tasks, with the hope that it will generalize to future tasks (Parisotto et al., 2015). Such an approach requires the tasks to be similar in their optimal policy, which is a much stronger assumption than ours.

Slightly off-topic are the works about "transfer learning" inside the same environment but across different reward functions, which is more restricted than the setting considered in this paper. Several prior works design representation learning algorithms that aim to learn a representation that generalize across multiple reward function/goals (Dayan, 1993; Barreto et al., 2017; Touati and Ollivier, 2021; Blier et al., 2021). These are related to the REWARDFREE REP-UCB we developed in Section E. The key difference is that we concern representation learning along with efficient exploration to derive an end-to-end polynomial sample complexity bound. These prior works do not consider exploration and do not come with provable sample complexity bounds. We refer interested readers to a recent survey (Zhu et al., 2020) for a comprehensive discussion of other empirical approaches.

# C   COMPARISONS TO CLOSELY RELATED WORKS

Recall that in addition to the commonly made shared representation assumption (Assumption 3.1), we made two additional structural assumptions: reachability (Assumption 3.3) and linear span on $\mu^\star$(Assumption 3.4). The reachability assumption is commonly made in prior works even in single agent RL, e.g. (Modi et al., 2021; Misra et al., 2020). It ensures that there is no redundant dimensions in the ground-truth representation, which is a reasonable requirement. The linear span assumption is closely related to the diversity assumptions made in prior works of transfer learning in both supervised learning (Tripuraneni et al., 2020) and reinforcement learning (Lu et al., 2021).

**Lu et al. (2021):**   In the prior work of Lu et al. (2021), which also studies transfer learning in low-rank MDPs with nonlinear function approximations, they need to make the following assumptions:

1. shared representation (identical to our Assumption 3.1).
2. task diversity (similar to our Assumption 3.4).
3. generative model access to both the source and the target tasks. In contrast, we only require generative model access to the source tasks and allow online learning in the target task.
4. a somewhat strong coverage assumption saying that the data covariance matrix (under the generative data distribution) between arbitrary pairs of features $\phi, \phi' \in \Phi$ must be full rank. In contrast, our analysis only requires coverage in the true feature $\phi^\star$ in the source tasks.
5. the existence of an ideal distribution q on which the learned representation can extrapolate. We do not require an assumption of a similar nature. Instead, we show that the data collected from our strategic reward-free exploration phase suffices for successful transfer.
6. the uniqueness for each $\phi$ in the sense of linear-transform equivalence. Two representation functions $\phi$ and $\phi'$ can yield similar estimation result if and only if they differ by just an invertible linear transformation. In contrast, we do not make any additional structural assumptions on the function class $\Phi$ beyond realizability.

In summary, our work present a theoretical framework that permits successful representation transfer based on significantly weaker assumptions. We believe that this is a solid step towards understanding transfer learning in RL.

**Cheng et al. (2022):** A concurrent work of Cheng et al. (2022) also studies the exact same problem as ours. Both works study the setting where the agent performs reward-free exploration in the source tasks for representation learning and used the learned representation for the target task. Both works achieves similar sample complexity in source and target tasks, albeit using very different algorithms. However, in addition to the assumptions that we made [2], Cheng et al. (2022) has to make an additional assumption.

**Assumption C.1** *For any two different models in the model class $\Phi \times \Psi$, say $P^1(s'|s, a) = \langle \phi^1(s, a), \mu^1(s') \rangle$ and $P^2(s'|s, a) = \langle \phi^2(s, a), \mu^2(s') \rangle$, there exists a constant $C_R$ such that for all $(s, a) \in \mathcal{S} \times \mathcal{A}$ and $h \in [H]$,*

$$\|P^1(\cdot|s, a) - P^2(\cdot|s, a)\|_{TV} \le C_R \mathbb{E}_{(s,a) \sim \mathcal{U}(\mathcal{S}, \mathcal{A})} \|P^1(\cdot|s, a) - P^2(\cdot|s, a)\|_{TV}$$

This assumption ensures that the point-wise TV error is bounded, as long as the population-level TV error is bounded. This assumption is used to transfer the MLE error from the source tasks to the target task. This type of assumption is strong in the sense that we typically expect $C_R$ to scale with $|\mathcal{S}|$. In contrast, our analysis (Lemma G.1) shows that this assumption is in fact not necessary assuming online access only to source tasks. The generative access to source task studied here, which enables transfer under weaker reachability assumptions is not studied in their work.

It is worth noting that Cheng et al. (2022) also study offline RL in the target task which we do not cover, while we mainly focus on the setting of generative models in the source tasks and demonstrating a more complete picture by proving generative model access in source tasks is needed without additional assumptions. Comparing to (Cheng et al., 2022), we also further implement and perform experimental evaluations of our algorithm.

## D    IMPOSSIBILITY RESULTS

Here, we present an interesting result showing that the above assumptions we make are so weak that they do not even permit efficient transfer in supervised learning:

**Theorem D.1** (Counter-example in supervised learning) *Assume that we want to perform conditional density estimation, where $P_k^\star(y|x) = \phi^\star(x)^\top \mu_k^\star(y)$. Under Assumption 3.1 (shared representation) and Assumption 3.4 (linear span), and assume that in each source task, one have access to a data generating distribution $\rho_k(x)$ such that $\lambda_{\min} \mathbb{E}_{\rho_k}[\phi^\star(x)\phi^\star(x)^\top] \ge \psi$ (reachability). No algorithm can consistently achieve $\mathbb{E}_{\rho_K}[\|\hat{P}_K^\star(y|x) - P_K^\star(y|x)\|_{TV}] \le 1/2$ on the target task using the feature learned from the source tasks with probability more than $1/2$.*

*Proof of Theorem D.1.* Consider the following example. $X = \mathbb{R}^2$ and we have the following 3 sets.

$$\mathcal{S}_1 = B_{1/2}((-1, -1))$$
$$\mathcal{S}_2 = B_{1/2}((-2, -2))$$
$$\mathcal{S}_3 = B_{1/2}((0, 1))$$

where $B_a((x, y))$ stands for the ball with radius $a$ centered at $(x, y)$. These will be the support of 3 tasks: task 1 and 2 are two source tasks, task 3 is the target task. Let's assume that $P_k^\star(x)$ are uniform distribution on $\mathcal{S}_k$.

Suppose that the feature class $\Phi$ only contains two functions:

$$\phi_1 : \{x_1 \le 0 \ \& \ x_2 \ge x_1\} \to (1, 0), \text{ and } (0, 1) \text{ otherwise}$$
$$\phi_2 : \{x_2 \le 0 \ \& \ x_1 \ge x_2\} \to (0, 1), \text{ and } (1, 0) \text{ otherwise}$$

---

[2]with a the minor difference that they directly assume the state space is compact with bounded measure

That is, the feature maps from $\mathbb{R}^2$ to the set of binary encoding of dimension 2, i.e. $\{(1,0),(0,1)\}$. We further assume that $\mu_k^\star = (p_1(y), p_2(y))$ for some distributions $p_1, p_2$, which is identical for all task $k$, where $\|p_1, p_2\|_{TV} = 1$. We also assume that $\mu_k^\star$ is known to the learner a prior, i.e. $\Upsilon_k = \{\mu_k^\star\}$ for all $k \in [K]$, so all the learner needs to do is to pick the correct $\phi$ out of two candidates.

Given the above setup, it's easy to verify that both Assumption 3.1 and Assumption 3.4 are satisfied, because the decision boundary of both $\phi_1$ and $\phi_2$ passes through the support of the source tasks, and all $\mu_k^\star$'s are identical. However, $\phi_1$ and $\phi_2$ are equivalent in $\mathcal{S}_1$ and $\mathcal{S}_2$ in terms of their representation power, therefore no algorithm can always pick the correct feature function with probability more than $1/2$, regardless of the number of samples. Suppose $\phi_1$ is the true feature and the algorithm incorrectly chooses $\phi_2$. Then, for $x \in S_3 \bigcap \{x_1 \geq 0\}$ which has probability mass $1/2$, $\hat{P}_3(y|x) = p_2$ whereas $P_3^\star(y|x) = p_1$. Thus, the expected total variation distance between $\hat{P}_K$ and $P_K^\star$ is $1/2$. $\qquad\square$

The above construction shows that our assumption are not sufficient to permit reliable representation transfer, even in the supervised learning setting. Yet, surprisingly, these assumptions are sufficient in the RL setting, implying somehow that transfer learning in RL in easier than transfer learning in SL. To understand this phenomenon, observe that in RL, the marginal distribution on $(s, a)$ is not independent from the conditional density $P(s'|s, a)$ we desire to estimate. In particular, if one collects data in the source tasks in an online fashion via running a policy, $\rho(s, a)$ is structurally restricted to be an occupancy distribution generated by the ground-truth transition $P^\star(s'|s, a)$. Such a connection can only exist in Markov chains, and our analysis elegantly utilizes this additional structure to establish the soundness of the learned representation. Also note, crucially, that we never learn a representation to capture $d_0$, which would suffer from similar issues as the supervised learning setting, but is not necessary for sample-efficient RL.

Next, we prove the impossibility result in Theorem 5.1, restated below as Theorem D.3. This result shows that one can not achieve online learning in the source tasks without significantly stronger assumptions such as Assumption 5.1. Before that, we provide a preliminary version, showing that the learned $\hat{\phi}$ is not sufficient to fit the transition model in the target task, which motivates the construction in Theorem D.3.

**Theorem D.2** (Impossibility Result: Model Learning) *Let $\mathcal{M}_K$ be the set of $K$-task multi-set that satisfies*

1. *all tasks are Block MDPs;*

2. *all tasks satisfy Assumption 3.3 and Assumption 3.4;*

3. *the latent dynamics are exactly the same for all source and target tasks.*

*For any pre-training algorithm $\mathcal{A}$, there exists $\{P_k^\star\}_{k=1:K} \in \mathcal{M}_K$ and an occupancy distribution $\rho_K$ on the target task, such that with probability at least $1/2$, $\mathcal{A}$ will output a feature $\hat{\phi}$ and for any $\mu$*

$$\mathbb{E}_{\rho_K}\|\hat{\phi}(s, a)^\top \mu(\cdot) - P_K^\star(\cdot|s, a)\|_{TV} \geq 1/2.$$

*Proof of Theorem D.2.* Consider a tabular MDP with 2 latent states $z_1, z_2$ and an observation state space $\mathcal{S} = R_1 \bigcup R_2 \bigcup B_1 \bigcup B_2$, where in task 1 one can only observe $R_1 \bigcup R_2$ and in task 2 one can only observe $B_1 \bigcup B_2$. Correspondingly, $o_1(s|z)$ is only supported on $R_1 \bigcup R_2$ (i.e., $o_1(R_i|z_i) = 1$) and similar for task 2. Let the latent state transition be such that $P(z_1|z_1, a) = 1$ and $P(z_2|z_2, a) = 1$, i.e. only self-transition regardless of the actions.

Now, consider a 2-element feature class $\Psi = \{\psi_1, \psi_2\}$ such that
$$\psi_1 = \{R_1 \to 1, R_2 \to 2, B_1 \to 1, B_2 \to 2\}$$
$$\psi_2 = \{R_1 \to 1, R_2 \to 2, B_1 \to 2, B_2 \to 1\}$$
Denote $\phi_i(s, a) = e_{(\psi_i(s), a)}$ for $i \in [1, 2]$. Consider for each task $k$, a 2-element $\Upsilon_k$ class in the form of $\Upsilon_k = \{(o_k(s|z_1), o_k(s|z_2)), (o_k(s|z_2), o_k(s|z_1))\}$.

Notice that $\phi_1$ and $\phi_2$ are merely permutations of one another and so given any single task data, the two hypothesis will not be distinguishable by any means. Therefore, for any algorithm, there is at least probability $1/2$ that it will choose the wrong hypothesis if the ground truth $\phi^\star$ is sampled between $\phi_1$ and $\phi_2$ uniformly at random. Suppose $\phi_1$ is the correct hypothesis and $\phi_2$ is the one that the algorithm picks (i.e., $\hat{\phi} = \phi_2$). Let task 3 be such that any state emits to $R_1 \bigcup R_2$ and $B_1 \bigcup B_2$ each with probability $1/2$ (i.e., $o_3(R_i|z_i) = o_3(B_i|z_i) = 0.5$). This construction satisfies Assumption 3.3 and Assumption 3.4.

Then, within task 3, one would encounter observations from both $R_1$ and $B_2$ which should be mapped to latent state $z_1$ and $z_2$ respectively by the true decoder $\phi_1$, but are instead both mapped to latent state $z_1$ by the learned decoder $\phi_2$, and thus $z_1$ and $z_2$ become indistinguishable. Suppose $\rho_K(z_1) = \rho_K(z_2) = 1/2$, then

$$
\mathbb{E}_{\rho_K}[\|\hat{\phi}(s,a)^\top \mu(\cdot) - P^\star(\cdot|s,a)\|_{TV}]
$$
$$
= \frac{1}{4}\|\hat{\phi}(R_1)^\top \mu(\cdot) - \phi(R_1)^{\star\top}\mu^\star(\cdot)\|_{TV} + \frac{1}{4}\|\hat{\phi}(B_1)^\top \mu(\cdot) - \phi(B_1)^{\star\top}\mu^\star(\cdot)\|_{TV}
$$
$$
\quad + \frac{1}{4}\|\hat{\phi}(R_2)^\top \mu(\cdot) - \phi(R_2)^{\star\top}\mu^\star(\cdot)\|_{TV} + \frac{1}{4}\|\hat{\phi}(B_2)^\top \mu(\cdot) - \phi(B_2)^{\star\top}\mu^\star(\cdot)\|_{TV}
$$
$$
= \|o_1 - o_1^\star\|_{TV}/4 + \|o_2 - o_1^\star\|_{TV}/4 + \|o_2 - o_2^\star\|_{TV}/4 + \|o_1 - o_2^\star\|_{TV}/4
$$
$$
\geq \frac{1}{4}\|o_1^\star - o_2^\star\|_{TV} + \frac{1}{4}\|o_1^\star - o_2^\star\|_{TV}
$$
$$
= \frac{1}{2},
$$

where the last second inequality uses triangle inequality, and the last equality comes from the fact that $o_3(\cdot|z_1)$ and $o_3(\cdot|z_2)$ have disjoint support which implies that $\|p_1^\star - p_2^\star\|_{TV} = 1$. □

Now, we are ready to restate and prove Theorem 5.1.

**Theorem D.3** (Impossibility Result: Optimal Policy Identification) *Let $\mathcal{M}_K$ be the set of K-task multi-set that satisfies*

1. *all tasks are Block MDPs;*

2. *all tasks satisfy Assumption 3.3 and Assumption 3.4;*

3. *the latent dynamics are exactly the same for all source and target tasks.*

*For any pre-training algorithm $\mathcal{A}$, there exists $\{P_k^\star\}_{k=1:K} \in \mathcal{M}_K$, such that with probability at least $1/2$, $\mathcal{A}$ will output a feature $\hat{\phi}$, such that for any policy taking the functional form of $\pi(s) = f\left(\{\hat{\phi}(s,a)\}_{a\in\mathcal{A}}, \{r(s,a)\}_{a\in\mathcal{A}}\right)$, we have*

$$V^\star - V^\pi \geq 1/2.$$

*Proof of Theorem D.3.* Consider a tabular MDP with $H = 2$, two latent states $z_1, z_2$ for $h = 1$ and two latent states $z_3, z_4$ for $h = 2$.

- For $h = 1$, let there be two actions $a_1, a_2$. Let the observation state space be $\mathcal{S} = R_1 \bigcup R_2 \bigcup B_1 \bigcup B_2$, where in task 1 one can only observe $R_1 \bigcup R_2$ and in task 2 one can only observe $B_1 \bigcup B_2$. Correspondingly, $o_1(s|z)$ is only supported on $R_1 \bigcup R_2$ (i.e., $o_1(R_i|z_i) = 1$) and similar for task 2. Let the latent state transition be such that $P(z_3|z_1,a_1) = P(z_3|z_2,a_2) = 1$, and $P(z_4|z_1,a_2) = P(z_4|z_2,a_1) = 1$. All rewards are 0 for $h = 1$.

- For $h = 2$, in state $z_3$, all actions have reward 1, and in state $z_4$ all actions have reward 0.

- The initial state distribution is $d_0(z_1) = d_0(z_2) = 1/2$.

---

**Algorithm 3** Multi-task REPLEARN

---

1: **Input:** Datasets $\{\mathcal{D}_k\}_{k=1:K-1}$, model class $\Phi$, $\Upsilon_k$, $k = 1, ..., K-1$.

2: Compute MLE $(\hat{\phi}, \hat{\mu}_{1:k-1}) := \underset{\phi \in \Phi, \mu_i \in \Upsilon_i}{\operatorname{argmax}} \sum_{k=1}^{K-1} \mathbb{E}_{\mathcal{D}_k} \left[ \log \phi(s, a)^\top \mu_k(s') \right]$.

3: **Return** $\hat{\phi}$

---

Now, consider a 2-element feature class $\Psi = \{\psi_1, \psi_2\}$ for $h = 1$, such that
$$\psi_1 = \{R_1 \to 1, R_2 \to 2, B_1 \to 1, B_2 \to 2\}$$
$$\psi_2 = \{R_1 \to 1, R_2 \to 2, B_1 \to 2, B_2 \to 1\}$$
Denote $\phi_i(s, a) = e_{(\psi_i(s), a)}$ for $i \in [1, 2]$. In addition, define $\Upsilon = \{\mu_1, \mu_2\}$ where
$$\mu_1 = \{z_3 \to (1, 0), z_4 \to (0, 1)\}$$
$$\mu_2 = \{z_4 \to (1, 0), z_3 \to (0, 1)\}$$
Notice that $\phi_1$ and $\phi_2$ are merely permutations of one another and so given any single task data, the two hypothesis will not be distinguishable by any means. Therefore, for any algorithm, there is at least probability $1/2$ that it will choose the wrong hypothesis. Suppose $\phi_1$ is the correct hypothesis and $\phi_2$ is the one that the algorithm picks (i.e., $\hat{\phi} = \phi_2$). Let task 3 be such that any state emits to $R_1 \bigcup R_2$ and $B_1 \bigcup B_2$ each with probability $1/2$ (i.e., $o_3(R_i|z_i) = o_3(B_i|z_i) = 0.5$). This construction satisfies Assumption 3.3 and Assumption 3.4.

Then, for any policy that only make decision based on $\hat{\phi}(s, a)$ and $r(s, a)$, $\pi$ would output the same action for observations in $R_1$ and $B_2$, or for $B_1$ and $R_2$. However, notice that the optimal policy, which would try to go to $z_3$ from either $z_1$ or $z_2$, will pick $a_1$ at $R_1$ and $B_1$ while picking $a_2$ at $R_2$ and $B_2$, which means that the optimal policy will not agree on $R_1$ and $B_2$, and it also will not agree on $R_2$ and $B_1$. Thus clearly, no such policy as defined above is capable of capturing the optimal policy. From the reward perspective, notice that $d^\pi(z_1) = d^\pi(z_2) = 1/2$ and $d^\pi(R_1) = d^\pi(R_2) = d^\pi(B_1) = d^\pi(B_1) = 1/4$. Since $\pi(R_1) = \pi(B_2)$, the agent will only be able to collect reward at one of the $R_1$ and $B_2$ (but not at both). Similarly, since $\pi(R_2) = \pi(B_1)$, the agent will only be able to collect reward at one of the $R_2$ and $B_1$ by reaching $z_3$ (but not at both). This means that $\pi$ will have average reward $1/2$. Since the optimal policy will be able to collect reward at all $R_1, R_2, B_1, B_2$, it will have average reward $1$. This concludes the proof. □

Theorem D.2 and Theorem D.3 show that it's impossible to allow online learning in the source tasks without much stronger assumptions. In our paper, we show that our Assumption 5.1, which ensures reachability in the raw states, is sufficient to establish an end-to-ending online transfer learning result. However, it is unclear if Assumption 5.1 is necessary for online learning. We leave this as an important direction of future work.

## E   REWARD-FREE REP-UCB

In this section, we adapt the Rep-UCB algorithm (Uehara et al., 2021) for reward-free exploration in a single task. We drop all task subscripts as this section is for a single task only, i.e. think about the task as being each source task. The original Rep-UCB algorithm was for infinite-horizon discounted MDPs, so we modify it to work for our undiscounted and finite-horizon setting. Our goal is to prove that Rep-UCB can learn a model that satisfies strong TV guarantees, i.e. Theorem E.1 and (5). Note that FLAMBE (Agarwal et al., 2020, Theorem 2) can be used for this directly, but at a worse (polynomial) sample complexity. Thus, we do a bit more work to derive a new model-learning algorithm for low-rank MDPs, based on Rep-UCB, that is more sample efficient in the source tasks.

A finite-horizon analysis of Rep-UCB was done in BRIEE (Zhang et al., 2022), so here we just need to replace BRIEE's RepLearn $\zeta_n$ with that of the MLE, which is how we learn $\hat{\phi}$ and $\hat{\mu}$, as in Rep-UCB. Recall the notation of (Zhang et al., 2022),
$$\rho_{h,n}(s, a) = \frac{1}{n} \sum_{i=0}^{n-1} d_h^{\hat{\pi}_i}(s) \operatorname{Unif}(a)$$

---

**Algorithm 4** REWARDFREE REP-UCB

---

1: **Input:** Regularizer $\lambda_n$, bonus scaling $\alpha_n$, model class $\mathcal{M} = \Phi \times \Upsilon$, number of episodes $N$.
2: Initialize $\widehat{\pi}_0$ as random and $\mathcal{D}_{h,0}, \mathcal{D}'_{h,0} = \emptyset$.
3: **for** episode $n = 1, 2, ..., N$ **do**
4:     Data collection from $\widehat{\pi}_{n-1}$: for $h = 1, 2, ..., H - 1$,

$$s \sim d_h^{\widehat{\pi}_{n-1}}, a \sim \text{Unif}(\mathcal{A}), s' \sim P_h^\star(s, a);$$

$$\widetilde{s} \sim d_{h-1}^{\widehat{\pi}_{n-1}}, \widetilde{a} \sim \text{Unif}(\mathcal{A}), \widetilde{s}' \sim P_{h-1}^\star(\widetilde{s}, \widetilde{a}), \widetilde{a}' \sim \text{Unif}(\mathcal{A}), \widetilde{s}'' \sim P_h^\star(\widetilde{s}', \widetilde{a}');$$

$$\mathcal{D}_{h,n} = \mathcal{D}_{h,n-1} \cup \{(s, a, s')\}, \mathcal{D}'_{h,n} = \mathcal{D}'_{h,n-1} \cup \{(\widetilde{s}', \widetilde{a}', \widetilde{s}'')\}.$$

    For $h = 0$, only collect $\mathcal{D}_{0,n}$.
5:     Learn model via MLE: for all $h = 0, 1, ..., H - 1$,

$$\widehat{P}_{h,n} = (\widehat{\phi}_{h,n}, \widehat{\mu}_{h,n}) = \underset{\phi_h, \mu_h \in \mathcal{M}_h}{\text{argmax}} \ \mathbb{E}_{\mathcal{D}_{h,n} \cup \mathcal{D}'_{h,n}} \left[ \log \phi_h(s, a)^T \mu_h(s') \right].$$

6:     Update exploration bonus: for all $h = 0, 1, ..., H - 1$,

$$\widehat{b}_{h,n}(s, a) = \alpha_n \left\| \widehat{\phi}_{h,n}(s, a) \right\|_{\widehat{\Sigma}_{h,n}^{-1}}$$

$$\widehat{\Sigma}_{h,n} = \sum_{(s,a,\_) \in \mathcal{D}_{h,n}} \widehat{\phi}_{h,n}(s, a) \widehat{\phi}_{h,n}(s, a)^T + \lambda_n I.$$

7:     Learn policy $\widehat{\pi}_n = \text{argmax}_\pi V_{\widehat{P}_n, \widehat{b}_n}^\pi$ and let $\widehat{V}_n$ be its value.
8: Let $\widehat{n} = \text{argmin}_{n \geq N/2} \widehat{V}_n$.
9: **Output:** $\widehat{n}, \widehat{P}_{\widehat{n}}$.

---

$$\beta_{h,n}(s, a) = \frac{1}{n} \sum_{i=0}^{n-1} \mathbb{E}_{\widetilde{s} \sim d_{h-1}^{\widehat{\pi}_i}, \widetilde{a} \sim \text{Unif}(\mathcal{A})} \left[ P_h^\star(s \mid \widetilde{s}, \widetilde{a}) \, \text{Unif}(a) \right]$$

$$\gamma_{h,n}(s, a) = \frac{1}{n} \sum_{i=0}^{n-1} d_h^{\widehat{\pi}_i}(s, a)$$

$$\Sigma_{\rho, \phi, n} = n \mathbb{E}_\rho \left[ \phi(s, a) \phi(s, a)^T \right] + \lambda_n I.$$

By using MLE ([Uehara et al., 2021](), Lemma 18) to learn models, with probability at least $1 - \delta$, for any $n = 1, 2, ..., N$ and $h = 0, 1, ..., H - 1$, we have

$$\max \left\{ \mathbb{E}_{\rho_{h,n}} \left\| \widehat{P}_{h,n}(s, a) - P_h^\star(s, a) \right\|_{TV}^2, \mathbb{E}_{\beta_{h,n}} \left\| \widehat{P}_{h,n}(s, a) - P_h^\star(s, a) \right\|_{TV}^2 \right\} \leq \zeta_n, \quad (3)$$

where

$$\zeta_n = \mathcal{O} \left( \frac{\log(|\mathcal{M}|nH/\delta)}{n} \right),$$

and $|\mathcal{M}| = \max_{h \in [H]} |\Phi_h||\Upsilon_h|$. We also adopt the same choice of $\alpha_n, \lambda_n$ parameters as BRIEE, which we assume from now on.

$$\lambda_n = \Theta \left( d \log(|\mathcal{M}|nH/\delta) \right)$$

$$\alpha_n = \Theta \left( \sqrt{n|\mathcal{A}|^2 \zeta_n + \lambda_n d} \right).$$

As in Rep-UCB, we posit standard assumptions about realizability and normalization, on the (source) task of interest.

**Assumption E.1** *For any $h = 0, 1, ..., H - 1$, we have $\phi_h^\star \in \Phi_h$ and $\mu_h^\star \in \Upsilon_h$. For any $\phi \in \Phi_h$, $\|\phi(s, a)\|_2 \leq 1$. For all $\mu \in \Upsilon_h$ and any function $g : \mathcal{S} \to \mathbb{R}$, we have $\| \int_s g(s) \mathrm{d}\mu(s) \|_2 \leq \|g\|_\infty \sqrt{d}$.*

**Lemma E.1** *Let $r$ be any reward function. Suppose we ran [Algorithm 4]() with [line 7]() having reward $r + \widehat{b}_n$ instead of just $\widehat{b}_n$. Then, for any $\delta \in (0, 1)$, w.p. at least $1 - \delta$, we have*

$$\sum_{n=0}^{N-1} V_{\widehat{P}_n, r+\widehat{b}_n}^{\widehat{\pi}_n} - V_{P^\star, r}^{\widehat{\pi}_n} \leq \mathcal{O} \left( H^2 d^2 |\mathcal{A}|^{1.5} \sqrt{N \log(|\mathcal{M}|NH/\delta)} \right)$$

*Proof.* Start from the third equation of Zhang et al. (2022, Theorem A.4). Following their proof until the last page of their proof, we arrive at the following: for any $n = 1, 2, ..., N$,

$$V_{\widehat{P}_n, r + \widehat{b}_n}^{\widehat{\pi}_n} - V_{P^\star, r}^{\widehat{\pi}_n}$$

$$\lesssim \sum_{h=0}^{H-2} \mathbb{E}_{\widetilde{s}, \widetilde{a} \sim d_{P^\star, h}^{\widehat{\pi}_n}} \|\phi_h^\star(\widetilde{s}, \widetilde{a})\|_{\Sigma_{\gamma_{h,n}, \phi_h^\star}^{-1}} \sqrt{|\mathcal{A}| \alpha_n^2 d + \lambda_n d} + \sqrt{|\mathcal{A}| \alpha_1^2 d / n}$$

$$+ (2H+1) \sum_{h=0}^{H-2} \mathbb{E}_{\widetilde{s}, \widetilde{a} \sim d_{P^\star, h}^{\widehat{\pi}_n}} \|\phi_h^\star(\widetilde{s}, \widetilde{a})\|_{\Sigma_{\gamma_{h,n}, \phi_h^\star}^{-1}} \sqrt{n |\mathcal{A}| \zeta_n + \lambda_n d} + (2H+1) \sqrt{|\mathcal{A}| \zeta_n}.$$

By elliptical potential arguments, we have

$$\sum_{n=0}^{N-1} \mathbb{E}_{\widetilde{s}, \widetilde{a} \sim d_{P^\star, h}^{\widehat{\pi}_n}} \|\phi_h^\star(\widetilde{s}, \widetilde{a})\|_{\Sigma_{\gamma_{h,n}, \phi_h^\star}^{-1}} \leq \sqrt{dN \log\left(1 + \frac{N}{d\lambda_1}\right)}.$$

Thus, summing over $n$, noting that $n\zeta_n, \alpha_n, \lambda_n$ are increasing in $n$, we can combine the above to get,

$$\sum_{n=0}^{N-1} V_{\widehat{P}_n, r + \widehat{b}_n}^{\widehat{\pi}_n} - V_{P^\star, r}^{\widehat{\pi}_n}$$

$$\lesssim \sqrt{dN \log\left(1 + \frac{N}{d\lambda_1}\right)} \left( H \sqrt{|\mathcal{A}| \alpha_N^2 d + \lambda_N d} + H^2 \sqrt{N |\mathcal{A}| \zeta_N + \lambda_N d} \right)$$

$$\lesssim \sqrt{dN \log\left(1 + \frac{N}{d\lambda_1}\right)} \left( H \sqrt{N |\mathcal{A}|^3 \zeta_N d + \lambda_N d^2} + H^2 \sqrt{N |\mathcal{A}| \zeta_N + \lambda_N d} \right)$$

$$\lesssim \sqrt{dN \log\left(1 + \frac{N}{d\lambda_1}\right)} \left( H^2 \sqrt{d |\mathcal{A}|^3 \log(|\mathcal{M}| NH/\delta) + d^3 \log(|\mathcal{M}| NH/\delta)} \right)$$

$$\in \mathcal{O}\left( H^2 d^2 |\mathcal{A}|^{1.5} \sqrt{N \log(|\mathcal{M}| NH/\delta)} \right).$$

$\square$

This gives the following useful corollary for reward free exploration.

**Lemma E.2** *For any $\delta \in (0, 1)$ w.p. at least $1 - \delta$ we have*

$$\widehat{V}_{\widehat{n}} \leq \mathcal{O}\left( H^2 d^2 |\mathcal{A}|^{1.5} \sqrt{\frac{\log(|\mathcal{M}| NH/\delta)}{N}} \right).$$

*Proof.* By definition of $\widehat{V}_{\widehat{n}}$, we have

$$\frac{N}{2} \widehat{V}_{\widehat{n}} \leq \sum_{n=N/2}^{N-1} V_{\widehat{P}_n, \widehat{b}_n}^{\widehat{\pi}_{\widehat{n}}} \leq \sum_{n=0}^{N-1} V_{\widehat{P}_n, \widehat{b}_n}^{\widehat{\pi}_{\widehat{n}}},$$

which is bounded by the previous lemma and the fact that $V_{P^\star, r=0}^{\widehat{\pi}_n} = 0$, since in Algorithm 4, the reward function is zero. $\square$

Conditioning on this, we now show that the environment $\widehat{P}_{\widehat{n}}$ has low TV error for any policy-induced distribution.

**Theorem E.1** *For any policy $\pi$, we have*

$$\sum_{h=0}^{H-1} \mathbb{E}_{d_{P^\star, h}^\pi} \left\| P_h^\star(s, a) - \widehat{P}_{h, \widehat{n}}(s, a) \right\|_{TV} \leq \mathcal{O}\left( H^3 d^2 |\mathcal{A}|^{1.5} \sqrt{\frac{\log(|\mathcal{M}| NH/\delta)}{N}} \right) := \varepsilon_{TV}.$$

*Proof.* In this proof, let $\widehat{P} = \widehat{P}_{\widehat{n}}$, which is the returned environment from the algorithm. Let $r(s,a) = \left\| P_h^\star(s,a) - \widehat{P}_h(s,a) \right\|_{TV} \in [0,2]$. Then,

$$\sum_{h=0}^{H-1} \left( \mathbb{E}_{d_{P^\star,h}^\pi} - \mathbb{E}_{d_{\widehat{P},h}^\pi} \right) [r(s,a)]$$

$$= V_{P^\star,r}^\pi - V_{\widehat{P},r}^\pi$$

$$= \sum_{h=0}^{H-1} \mathbb{E}_{d_{\widehat{P},h}^\pi} \left[ \left( \mathbb{E}_{P_h^\star(s,a)} - \mathbb{E}_{\widehat{P}_h(s,a)} \right) V_{P^\star,r,h+1}^\pi(s') \right] \qquad \text{(Simulation lemma)}$$

$$\leq 2H \sum_{h=0}^{H-1} \mathbb{E}_{d_{\widehat{P},h}^\pi} \left\| P_h^\star(s,a) - \widehat{P}_h(s,a) \right\|_{TV}.$$

Thus,

$$\sum_{h=0}^{H-1} \mathbb{E}_{d_{P^\star,h}^\pi} \left\| P_h^\star(s,a) - \widehat{P}_h(s,a) \right\|_{TV}$$

$$\leq (2H+1) \sum_{h=0}^{H-1} \mathbb{E}_{d_{\widehat{P},h}^\pi} \left\| P_h^\star(s,a) - \widehat{P}_h(s,a) \right\|_{TV} \qquad \text{(by (Zhang et al., 2022, Lemma A.1))}$$

$$\lesssim H \left( \sum_{h=0}^{H-2} \mathbb{E}_{d_{\widehat{P},h}^\pi} \left[ \widehat{b}_{h,\widehat{n}}(s,a) \right] + \sqrt{|\mathcal{A}| \zeta_{N/2}} \right)$$

$$\leq H \left( V_{\widehat{P},\widehat{b}_{\widehat{n}}}^\pi + \sqrt{2|\mathcal{A}| \frac{\log(|\mathcal{M}|NH/\delta)}{N}} \right)$$

$$\leq H \left( V_{\widehat{P},\widehat{b}_{\widehat{n}}}^{\widehat{\pi}_{\widehat{n}}} + \sqrt{2|\mathcal{A}| \frac{\log(|\mathcal{M}|NH/\delta)}{N}} \right)$$

$$\lesssim H \left( H^2 d^2 |\mathcal{A}|^{1.5} \sqrt{\frac{\log(|\mathcal{M}|NH/\delta)}{N}} + \sqrt{|\mathcal{A}| \frac{\log(|\mathcal{M}|NH/\delta)}{N}} \right) \qquad \text{(by Lemma E.2)}$$

$$\in \mathcal{O} \left( H^3 d^2 |\mathcal{A}|^{1.5} \sqrt{\frac{\log(|\mathcal{M}|NH/\delta)}{N}} \right).$$

$\square$

This also gives us a guarantee on the TV distance between the visitation distributions induced by $P^\star$ vs. by $\widehat{P}$.

**Lemma E.3** *Suppose $\widehat{P}$ satisfies the following for all $h = 0, 1, ..., H-1$,*

$$\forall \pi : \mathbb{E}_{d_{P^\star,h}^\pi} \left\| \widehat{P}_h(s,a) - P_h^\star(s,a) \right\|_{TV} \leq \varepsilon_h. \qquad (4)$$

*Then, for any $h = 0, 1, ..., H-1$, we have*

$$\forall \pi : \left\| d_{\widehat{P},h}^\pi - d_{P^\star,h}^\pi \right\|_{TV} \leq \sum_{t=0}^{h-1} \varepsilon_t.$$

*Note, for $h = 0$, the sum is empty so the right hand side is 0.*

*Proof.* We proceed by induction for $h = 0, 1, ..., H-1$. For the base case of $h = 0$, no transition has been taken, so that $d_{\widehat{P},0}^\pi = d_{P^\star,0}^\pi$. Now let $h \in \{0, 1, ..., H-2\}$ be arbitrary, and suppose that the claim is true for $h$ (IH). We want to show the claim holds for $h+1$. One key fact we'll use is that, for any measure $\mu$, we have $\|\mu\|_{TV} = \sup_{\|f\|_\infty \leq 1} |\mathbb{E}_\mu[f]|$. Below we use the notation that $f(s,\pi) = \mathbb{E}_{a \sim \pi(s)} f(s,a)$.

$$\|d_{\widehat{P},h+1}^\pi - d_{h+1}^\pi\|_{TV}$$

$$
\begin{aligned}
&= \sup_{\|f\|_\infty \le 1} \left| \mathbb{E}_{d^\pi_{\widehat{P},h+1}}[f(s,a)] - \mathbb{E}_{d^\pi_{h+1}}[f(s,a)] \right| \\
&= \sup_{\|f\|_\infty \le 1} \left| \mathbb{E}_{(\widetilde{s},\widetilde{a})\sim d^\pi_{\widehat{P},h},(s,a)\sim \widehat{P}_h(\widetilde{s},\widetilde{a})}[f(s,\pi_{h+1})] - \mathbb{E}_{(\widetilde{s},\widetilde{a})\sim d^\pi_h,(s,a)\sim P^\star_h(\widetilde{s},\widetilde{a})}[f(s,\pi_{h+1})] \right| \\
&\le \sup_{\|f\|_\infty \le 1} \left| \left( \mathbb{E}_{(\widetilde{s},\widetilde{a})\sim d^\pi_{\widehat{P},h}} - \mathbb{E}_{(\widetilde{s},\widetilde{a})\sim d^\pi_h} \right) \mathbb{E}_{\widehat{P}_h(\widetilde{s},\widetilde{a})} f(s,\pi_{h+1}) \right| \\
&\quad + \sup_{\|f\|_\infty \le 1} \left| \mathbb{E}_{(\widetilde{s},\widetilde{a})\sim d^\pi_h} \left[ \mathbb{E}_{\widehat{P}_h(\widetilde{s},\widetilde{a})} f(s,a) - \mathbb{E}_{P^\star_h(\widetilde{s},\widetilde{a})} f(s,\pi_{h+1}) \right] \right| \\
&\le \sum_{t=0}^{h-1} \varepsilon_t + \mathbb{E}_{(\widetilde{s},\widetilde{a})\sim d^\pi_h} \left[ \sup_{\|f\|_\infty \le 1} \left| \left( \mathbb{E}_{\widehat{P}_h(\widetilde{s},\widetilde{a})} - \mathbb{E}_{P^\star_h(\widetilde{s},\widetilde{a})} \right) f(s,\pi_{h+1}) \right| \right] \qquad \text{(by IH and Jensen)} \\
&\le \sum_{t=0}^{h-1} \varepsilon_t + \varepsilon_h, \qquad\qquad\qquad\qquad\qquad\qquad\qquad\qquad \text{(by (4) and } \|f(s,\pi_{h+1})\| \le 1)
\end{aligned}
$$

as desired. □

Thus, when combined with Theorem E.1, we have for $h = 0, 1, ..., H-1$ and any policy $\pi$,

$$
\|d^\pi_{\widehat{P},h} - d^\pi_{P^\star,h}\|_{TV} \le \mathcal{O}\left( H^3 d^2 |\mathcal{A}|^{1.5} \sqrt{\frac{\log(|\mathcal{M}|NH/\delta)}{N}} \right) = \varepsilon_{TV}. \tag{5}
$$

In other words, the sample complexity needed for a model-error of $\varepsilon_{TV}$ is

$$
\mathcal{O}\left( \frac{H^6 d^4 |\mathcal{A}|^3 \log(|\mathcal{M}|NH/\delta)}{\varepsilon_{TV}^2} \right).
$$

Note this is much better than FLAMBE's guarantee (Agarwal et al., 2020, Theorem 2) which requires,

$$
\mathcal{O}\left( \frac{H^{22} d^7 |\mathcal{A}|^9 \log(|\mathcal{M}|NH/\delta)}{\varepsilon_{TV}^{10}} \right).
$$

## F  REWARD-FREE EXPLORATION

In this section, we show that the mixture policy returned by Algorithm 2 has good coverage. Recall that Algorithm 2 contains two main steps:

Step 1  Learn a model $\widehat{P}$. This was the focus of the previous section, where our modified REP-UCB method obtained a strong TV guarantee ((5)) by requiring number of episodes at most,

$$
N_{\text{REWARDFREE}} = \mathcal{O}\left( \frac{H^6 d^4 |\mathcal{A}|^3 \log(|\mathcal{M}|NH/\delta)}{\varepsilon_{TV}^2} \right).
$$

Step 2  Run LSVI-UCB (Algorithm 6) in the *learned* model $\widehat{P}$ with reward at the $e$-th episode being $b_{h,e}$ and UNIFORMACTIONS = TRUE. The optimistic bonus pushes the algorithm to explore directions that are not well-covered yet by the mixture policy up to this point. With elliptical potential, we can establish that this process will terminate in polynomial number of steps.

We now focus on Step 2. Let $\pi^{+1}_h$ denote rolling-in $\pi$ for $h$ steps and taking uniform actions on the $h+1$ step, thus inducing a distribution over $s_{h+1}, a_{h+1}$. We abuse the notation a little and use $\pi^{+1}_{-1}$ for a policy that just takes one uniform action from the initial distribution $d_0$.

**Lemma F.1** *Let $\delta \in (0,1)$ and run REWARDFREE (Algorithm 2). Let $\Lambda_{h,N}$ be the empirical covariance at the $N$-th iteration of LSVI-UCB (Algorithm 6). Then, w.p. at least $1 - \delta$ we have,*

$$
\sup_\pi \sum_{h=0}^{H-1} \mathbb{E}_{\pi^{+1}_{h-1},\widehat{P}} \left\| \widehat{\phi}_h(s_h,a_h) \right\|_{\Lambda_{h,N}^{-1}} \lesssim A d^{1.5} H^3 \sqrt{\log(dNH/\delta)/N}.
$$

*Proof.* In this proof, we'll treat the empirical MDP as $P^\star$, as that is the environment we're running in. Thus, we abuse notation and $\widehat{P}_{h,e}$ is the model-based perpsective of the linear MDP, i.e. $\widehat{\phi}_h\widehat{\mu}_{h,e}$ where $\widehat{\mu}_{h,e}$ is $\Lambda_{h,e}^{-1}\sum_{k=1}^{e-1}\widehat{\phi}_h(s_h^k,a_h^k)\delta(s_{h+1}^k)$. Also, in Algorithm 2, we set reward to be zero, but for the purpose of this analysis, suppose the reward function is precisely the (unscaled) bonus in Lsvi-Ucb, i.e. $r_{h,e}(s_h,a_h) = b_{h,e}(s_h,a_h)$. This does not change the algorithm at all since the $\beta$-scaling of the bonus dominates this reward in the definition of $\widehat{Q}_{h,e}$, but thinking about the reward in this way will make our analysis simpler.

Recall the high-level proof structure of reward free guarantee of linear MDP (with known features $\widehat{\phi}$) (Wang et al., 2020, Lemma 3.2).

Step 1 Show that $\widehat{V}_{h,e} \in \mathcal{V}_h$ and w.p. $1 - \delta$, for all $h, e$,
$$\forall s_h, a_h : \sup_{f\in\mathcal{V}_h}\left|\widehat{P}_{h,e}(s_h,a_h) - P_h^\star(s_h,a_h)\right|f \le \beta b_{h,e}(s_h,a_h).$$

This step only uses self-normalized martingale bounds. So, line 9 can use any martingale sequence of states and actions, and this claim still holds, with bonus $b_{h,e}$ using the appropriate covariance under the data.

Step 2 Show optimism conditioned on Step 1. Specifically, for all $e = 1, 2, ..., N$, we have $\mathbb{E}_{d_0}\left[V_0^\star(s_0, r_e) - \widehat{V}_{0,e}(s_0)\right] \le 0$. To show this, we need that $\widehat{V}_{h,e}(s_h) = \widehat{Q}_{h,e}(s_h, \pi_h^e(s_h)) \ge \widehat{Q}_{h,e}(s_h, \pi_h^\star(s_h))$ (this is for the unclipped case of $V$-optimism), which we have satisfied in the algorithm, i.e. $\pi_h^e$ is greedy w.r.t. $\widehat{Q}_{h,e}$.

Step 3 Bound the sum $\sum_e \widehat{V}_{h,e}$, where we decompose it as a sum of expected bonuses with the expectation is under $\pi^e$.

Step 3 is the only place where we use the fact that $s_h^k, a_h^k$ are data sampled from rolling out $\widehat{\pi}^e$. For Step 1 and 2, please refer to existing proofs in (Agarwal et al., 2019; Jin et al., 2020b; Wang et al., 2020).

Now we show Step 3 for our modified algorithm with uniform actions. First, let us show a simulation lemma. For any episode $e = 1, 2, ..., N$, for any $s$, recalling definition of reward being $b_{h,e}$, we have
$$\widehat{V}_{0,e}(s_0) \le (1+\beta)b_{0,e}(s_0,\pi_0^e(s_0)) + \widehat{P}_{0,e}(s_0,\pi^e(s_0))\widehat{V}_{1,e}$$
$$\le (1+2\beta)b_{0,e}(s_0,\pi_0^e(s_0)) + P_{0,e}^\star(s_0,\pi^e(s_0))\widehat{V}_{1,e},$$

where the first inequality is due to the thresholding on $\widehat{V}_{h,e}$'s and the second inequality is due to Step 1. Continuing in this fashion, we have
$$\mathbb{E}_{d_0}\left[\widehat{V}_{0,e}(s_0)\right] \le (1+2\beta)\sum_{h=0}^{H-1}\mathbb{E}_{\pi^e}\left[b_{h,e}(s_h,a_h)\right].$$

Summing over $e = 1, 2, ..., N$, we have
$$\sum_{e=1}^N\mathbb{E}_{d_0}\left[\widehat{V}_{0,e}(s_0)\right] \lesssim \beta\sum_{h=0}^{H-1}\sum_{e=1}^N\mathbb{E}_{\pi^e}\left[b_{h,e}(s_h,a_h)\right]$$
$$\le A\beta\sum_{h=0}^{H-1}\sum_{e=1}^N\mathbb{E}_{\left(\pi_{h-1}^e\right)^{+1}}\left[b_{h,e}(s_h,a_h)\right]$$

For each $h = 0, 1, ..., H - 1$, apply Azuma's inequality to the martingale difference sequence $\Delta_e = \mathbb{E}_{\left(\pi_{h-1}^e\right)^{+1}}\left[b_{h,e}(s_h,a_h)\right] - b_{h,e}(s_h^e,a_h^e)$. The envelope is at most 2. So, w.p. $1 - \delta$,
$$\le A\beta\sum_{h=0}^{H-1}\sum_{e=1}^N b_{h,e}(s_h^e,a_h^e) + A\beta\sqrt{N\log(H/\delta)}.$$

Now apply a self-normalized elliptical potential bound to the first term, giving that

$$\sum_{h=0}^{H-1} \sum_{e=1}^{N} b_{h,e}(s_h^e, a_h^e) \leq \sum_{h=0}^{H-1} \sqrt{N} \sqrt{\sum_{e=1}^{N} b_{h,e}(s_h^e, a_h^e)^2} \lesssim H\sqrt{dN \log(N)}.$$

Thus, we finally have

$$\sum_{e=1}^{N} \mathbb{E}_{d_0} \left[ \widehat{V}_{0,e}(s_0) \right] \lesssim A\beta H \sqrt{dN \log(NH/\delta)}.$$

Consider any episode $e = 1, 2, ..., N$. By definition, $\Lambda_{h,N} \succeq \Lambda_{h,e}$, so for all $s, a$ we have pointwise that $b_{h,N}(s,a) \leq b_{h,e}(s,a)$. Hence, for all $s$, we have $V_0^\star(s; r^N) \leq V_0^\star(s; r^e)$, and further using optimism, we have

$$N\mathbb{E}_{d_0} [V_0^\star(s_0; r_N)] \leq \sum_{e=1}^{N} \mathbb{E}_{d_0} [V_0^\star(s_0; r_e)] \leq \sum_{e=1}^{N} \mathbb{E}_{d_0} \left[ \widehat{V}_{0,e}(s_0) \right] \lesssim A\beta H \sqrt{dN \log(NH/\delta)}.$$

Now consider any $h$ and policy $\pi$, and consider rolling it out for $h - 1$ steps and taking a random action. Then we have

$$\mathbb{E}_{\pi_{h-1}^{+1}, \widehat{P}} \left\| \widehat{\phi}_h(s_h, a_h) \right\|_{\Lambda_{h,N}^{-1}} \leq \mathbb{E}_{d_0} \left[ V_0^{\pi_{h-1}^{+1}}(s_0; r_N) \right] \leq A\beta H \sqrt{d \log(NH/\delta)/N}.$$

Summing over $h$ incurs an extra $H$ factor on the right. This concludes the proof. $\qquad\square$

**Lemma F.2** (One-step back for Linear MDP) *Suppose $P_h = (\phi_h, \mu_h)$ is a linear MDP. Suppose $\rho$ is any mixture of $n$ policies, and let $\Sigma_h := n\mathbb{E}_\rho \left[ \phi_h(s_h, a_h)\phi_h(s_h, a_h)^\top \right] + \lambda I$ denote the unnormalized covariance. For any $g : S \times A \to \mathbb{R}$, policy $\pi$, and $h = 0, 1, ..., H - 2$, we have*

$$\mathbb{E}_\pi [g(s_{h+1}, a_{h+1})] \leq \mathbb{E}_\pi \left[ \|\phi_h(s_h, a_h)\|_{\Sigma_h^{-1}} \right] \sqrt{nA\mathbb{E}_{\rho_h^{+1}} [g(s_{h+1}, a_{h+1})^2] + \lambda d\|g\|_\infty^2}$$

*Proof.*

$$\mathbb{E}_\pi [g(s_{h+1}, a_{h+1})] = \left\langle \mathbb{E}_\pi [\phi_h(s_h, a_h)], \int_{s_{h+1}} g(s_{h+1}, \pi_{h+1}) \mathrm{d}\mu_h(s_{h+1}) \right\rangle$$

$$\leq \mathbb{E}_\pi \|\phi_h(s_h, a_h)\|_{\Sigma_h^{-1}} \left\| \int_{s_{h+1}} g(s_{h+1}, \pi_{h+1}) \mathrm{d}\mu_h(s_{h+1}) \right\|_{\Sigma_h},$$

where

$$\left\| \int_{s_{h+1}} g(s_{h+1}, \pi_{h+1}) \mathrm{d}\mu_h(s_{h+1}) \right\|_{\Sigma_h}$$

$$= n\mathbb{E}_\rho \left[ \left( \mathbb{E}_{s_{h+1} \sim P_h(s_h, a_h)} [g(s_{h+1}, \pi_{h+1})] \right)^2 \right] + \lambda \left\| \int_{s_{h+1}} g(s_{h+1}, \pi_{h+1}) \mathrm{d}\mu_h(s_{h+1}) \right\|^2$$

$$\leq n|\mathcal{A}|\mathbb{E}_{\rho_h^{+1}} [g(s_{h+1}, a_{h+1})^2] + \lambda d\|g\|_\infty^2.$$

$\qquad\square$

Under reachability, we can show that small (squared) bonuses and spectral coverage, in the sense of having lower bounded eigenvalues, are somewhat equivalent.

**Lemma F.3** *Let $\Sigma$ be a symmetric positive definite matrix and define the bonus $b_h(s,a) = \|\phi_h^\star(s,a)\|_{\Sigma^{-1}}$. Then we have*

1. *For any policy $\pi$, $\mathbb{E}_{d_h^\pi} \left[ b_h^2(s,a) \right] \leq \frac{1}{\lambda_{\min}(\Sigma)}$. That is, coverage implies small squared bonus.*

2. *Suppose reachability under $\phi^\star$ (Assumption 3.3), then we have the converse: there exists $\widehat{\pi}$, for any policy $\pi$, $\mathbb{E}_{d_h^{\widehat{\pi}}} \left[ b_h^2(s,a) \right] \geq \frac{\psi}{\lambda_{\min}(\Sigma)}$. That is, small squared bonus implies coverage.*

*Proof.* The first claim follows directly from Cauchy-Schwartz. Indeed, for any policy $\pi$, we have

$$\mathbb{E}_{d_h^\pi}\left[b_h^2(s,a)\right] \leq \mathbb{E}_{d_h^\pi}\left[\|\phi_h^\star(s,a)\|_2^2\|\Sigma^{-1}\|_2\right] \leq \frac{1}{\lambda_{\min}(\Sigma)}.$$

For the second claim, Assumption 3.3 implies that there exist a policy $\widetilde{\pi}$ such that for all vectors $v \in \mathbb{R}^d$ with $\|v\|_2 = 1$, we have $\mathbb{E}_{d_h^{\widetilde{\pi}}}\left[(\phi_h^\star(s,a)^\top v)^2\right] \geq \psi$. Now decompose $\Sigma = \sum_{i=1}^d \lambda_i v_i v_i^\top$, where $\lambda_i, v_i$ are eigenvalue/vector pairs with $\|v_i\|_2 = 1$ and $\lambda_1 \geq \lambda_2 \geq ... \geq \lambda_d$. Then substituting this into the definition of the bonus, we have

$$\begin{aligned}
\mathbb{E}_{d_h^{\widetilde{\pi}}}\left[b_h^2(s,a)\right] &= \sum_{i=1}^d \frac{1}{\lambda_i}\mathbb{E}_{d_h^{\widetilde{\pi}}}\left[(\phi_h^\star(s,a)^T v_i)^2\right] \\
&\geq \frac{1}{\lambda_d}\mathbb{E}_{d_h^{\widetilde{\pi}}}\left[(\phi_h^\star(s,a)^T v_d)^2\right] \\
&\geq \frac{\psi}{\lambda_{\min}(\Sigma)}.
\end{aligned}$$

$\square$

We now prove our main lemma for reward-free exploration, Lemma 4.2.

**Lemma 4.2** (Reward-free Exploration) *Fix any source task $k \in [K-1]$. Suppose Assumptions 3.2 and 3.3. Then, for any $\delta \in (0,1)$, w.p. $1-\delta$, REWARDFREE (Algorithm 2) with $N_{\text{LSVI-UCB}} = \widetilde{\Theta}\left(A^3 d^6 H^8 \psi^{-2}\right)$ and $N_{\text{REWARDFREE}} = \widetilde{\mathcal{O}}\left(A^3 d^4 H^6 \log\left(\frac{|\Phi||\Upsilon|}{\delta}\right) N_{\text{LSVI-UCB}}^2\right)$ returns a $\lambda_{\min}$-exploratory policy $\pi_k$ where $\lambda_{\min} = \widetilde{\Omega}\left(A^{-3} d^{-5} H^{-7} \psi^2\right)$. The sample complexity here is $N_{\text{REWARDFREE}}$ episodes in the source task.*

*Proof of Lemma 4.2.* In this proof, let

$$\Lambda_h^\star = N_{\text{LSVI-UCB}}\mathbb{E}_{\rho_{h-1}^{+1}}\left[\phi_h^\star(s_h,a_h)\phi_h^\star(s_h,a_h)^\top\right] + \lambda I,$$

$$\widehat{\Lambda}_h = N_{\text{LSVI-UCB}}\mathbb{E}_{\rho_{h-1}^{+1}}\left[\widehat{\phi}_h(s_h,a_h)\widehat{\phi}_h(s_h,a_h)^\top\right] + \lambda I,$$

where $\lambda = dH\log(N_{\text{LSVI-UCB}}/\delta) \geq 1$. This setting of $\lambda$ satisfies the precondition for the Concentration of Inverse Covariances Zanette et al. (2021, Lemma 39), which implies w.p. at least $1-\delta$ that

$$\widehat{\Lambda}_h^{-1} \preceq 2\left(\sum_{e=1}^{N_{\text{LSVI-UCB}}} \widehat{\phi}_h(s_h^e,a_h^e)\widehat{\phi}_h(s_h^e,a_h^e)^\top + \lambda I\right)^{-1} \preceq 2\Lambda_{h,N_{\text{LSVI-UCB}}}^{-1},$$

where we've also used the fact that $\lambda \geq 1$, so $(A + \lambda I)^{-1} \preceq (A + I)^{-1}$.

Under this event, for any $\pi$, we have,

$$\sum_{h=1}^H \mathbb{E}_{\pi,\widehat{P}}\left\|\widehat{\phi}_h(s_h,a_h)\right\|_{\widehat{\Lambda}_h^{-1}} \lesssim \sum_{h=1}^H \mathbb{E}_{\pi,\widehat{P}}\left\|\widehat{\phi}_h(s_h,a_h)\right\|_{\widehat{\Lambda}_{h,N_{\text{LSVI-UCB}}}^{-1}}. \tag{6}$$

Now let $h = 0, 1, ..., H-2$ be arbitrary. By Assumption 3.3 (there exists some policy $\widetilde{\pi}$ with coverage) such that,

$$\begin{aligned}
&\frac{\psi}{\lambda_{\min}\left(\Lambda_{h+1}^\star\right)} \\
&\leq \mathbb{E}_{\widetilde{\pi}}\left\|\phi_{h+1}^\star(s_{h+1},a_{h+1})\right\|_{(\Lambda_{h+1}^\star)^{-1}}^2 && \text{(by Lemma F.3)} \\
&\leq \mathbb{E}_{\widetilde{\pi}}\left\|\phi_{h+1}^\star(s_{h+1},a_{h+1})\right\|_{(\Lambda_{h+1}^\star)^{-1}} && \text{(by } \lambda \geq 1) \\
&\leq \mathbb{E}_{\widetilde{\pi},\widehat{P}}\left\|\widehat{\phi}_h(s_h,a_h)\right\|_{\widehat{\Lambda}_h^{-1}}\sqrt{A(2d + \varepsilon_{TV}N_{\text{LSVI-UCB}})} + \varepsilon_{TV} && \text{(by Corollary F.1)}
\end{aligned}$$

$$\leq \mathbb{E}_{\widetilde{\pi},\widehat{P}} \left\| \widehat{\phi}_h(s,a) \right\|_{\widehat{\Lambda}_h^{-1}} \sqrt{A(2d+1)} + 1/N_{\text{LSVI-UCB}} \qquad (\text{by } \varepsilon_{TV} = 1/N_{\text{LSVI-UCB}})$$

$$\lesssim A^{1.5} d^2 H^3 \sqrt{\log(dH N_{\text{LSVI-UCB}}/\delta)/N_{\text{LSVI-UCB}}} + 1/N_{\text{LSVI-UCB}} \qquad (\text{by (6) and Lemma F.1})$$

$$\lesssim A^{1.5} d^2 H^3 \sqrt{\log(dH N_{\text{LSVI-UCB}}/\delta)/N_{\text{LSVI-UCB}}}.$$

Recall that $\lambda = dH \log(N_{\text{LSVI-UCB}}/\delta)$, we have,

$$\lambda_{\min}\left( \mathbb{E}_{\rho_h^{+1}} \left[ \phi_{h+1}^\star(s,a) \phi_{h+1}^\star(s,a)^T \right] \right)$$

$$= \frac{\lambda_{\min}\left( \Lambda_{h+1}^\star \right) - \lambda}{N_{\text{LSVI-UCB}}}$$

$$\geq \frac{1}{N_{\text{LSVI-UCB}}} \left( \frac{C\psi}{A^{1.5} d^2 H^3 \sqrt{\log(dH N_{\text{LSVI-UCB}}/\delta)/N_{\text{LSVI-UCB}}}} - dH \log(N_{\text{LSVI-UCB}}/\delta) \right)$$

$$\gtrsim \frac{C\psi}{A^{1.5} d^2 H^3 \sqrt{N_{\text{LSVI-UCB}}}} - \frac{dH}{N_{\text{LSVI-UCB}}},$$

where we've omitted the log terms for simplicity in the $\gtrsim$. Now we optimize $N_{\text{LSVI-UCB}}$ to maximize this bound. For $a, b > 0$, to maximize a function of the form $f(x) = \frac{a}{\sqrt{x}} - \frac{b}{x}$, it's best to set $x^\star$ such that $\sqrt{x^\star} = \frac{2b}{a}$, resulting in value $f(x^\star) = \frac{a^2}{4b}$. Setting,

$$x = N_{\text{LSVI-UCB}},$$

$$a = \frac{C\psi}{A^{1.5} d^2 H^3},$$

$$b = dH.$$

Hence, we need to set

$$N_{\text{LSVI-UCB}} = \widetilde{\Theta}\left( b^2/a^2 \right) = \widetilde{\Theta}\left( \frac{A^3 d^6 H^8}{\psi^2} \right),$$

which results in a $\lambda_{\min}$ lower bound of

$$\lambda_{\min}\left( \mathbb{E}_{\rho_h^{+1}} \left[ \phi_{h+1}^\star(s_{h+1}, a_{h+1}) \phi_{h+1}^\star(s_{h+1}, a_{h+1})^T \right] \right) = \widetilde{\Omega}\left( a^2/b \right) = \widetilde{\Omega}\left( \frac{\psi^2}{A^3 d^5 H^7} \right).$$

Finally, we used the fact that $\varepsilon_{TV} = 1/N_{\text{LSVI-UCB}}$, which is set by the choice of $N_{\text{REWARDFREE}}$ in the lemma statement to satisfy (5).

The above proves coverage of $\rho_h^{+1}$ for $h = 0, 1, ..., H-2$. Finally to argue for $\rho_{-1}^{+1}$, which is simply taking a random action at time $h$, we can simply invoke Assumption 3.3 for $h = 0$ to get a policy $\widetilde{\pi}$ that

$$\mathbb{E}_{\rho_{-1}^{+1}} \left[ \phi_0^\star(s_0, a_0) \phi_0^\star(s_0, a_0)^\top \right] \succeq \frac{1}{A} \mathbb{E}_{\widetilde{\pi}} \left[ \phi_0^\star(s_0, a_0) \phi_0^\star(s_0, a_0)^\top \right] \succeq \frac{\psi}{A}.$$

$\square$

**Corollary F.1** *Let $\lambda, \Lambda_h^\star, \widehat{\Lambda}_h$ be defined as in the proof of Lemma 4.2. For any $h = 0, 1, ..., H-2$ and any policy $\pi$, we have*

$$\mathbb{E}_\pi \left[ \left\| \phi_{h+1}^\star(s_{h+1}, a_{h+1}) \right\|_{(\Lambda_{h+1}^\star)^{-1}} \right] \leq \mathbb{E}_{\pi,\widehat{P}} \left[ \left\| \widehat{\phi}_h(s_h, a_h) \right\|_{\widehat{\Lambda}_h^{-1}} \right] \sqrt{|\mathcal{A}|(2d + \varepsilon_{TV} N_{\text{LSVI-UCB}})} + \varepsilon_{TV}.$$

*Intuitively, this means that coverage in the learned features implies coverage in the true features.*

*Proof.* For shorthand, let $N = N_{\text{LSVI-UCB}}$. Apply Lemma F.2 (one-step back) to the learned model $\widehat{P}$ and the function $(s,a) \mapsto \| \phi_{h+1}^\star(s,a) \|_{(\Lambda_{h+1}^\star)^{-1}}$, which is bounded by $\lambda^{-1/2} \leq 1$. We have,

$$\mathbb{E}_{\pi,\widehat{P}} \left\| \phi_{h+1}^\star(s_{h+1}, a_{h+1}) \right\|_{(\Lambda_{h+1}^\star)^{-1}}$$

$$\leq \mathbb{E}_{\pi,\widehat{P}} \left\| \widehat{\phi}_h(s_h, a_h) \right\|_{\widehat{\Lambda}_h^{-1}} \sqrt{N |\mathcal{A}| \mathbb{E}_{\rho_h^{+1}, \widehat{P}} \| \phi_{h+1}^\star(s_{h+1}, a_{h+1}) \|_{(\Lambda_{h+1}^\star)^{-1}}^2 + d}$$

$$\leq \mathbb{E}_{\pi,\widehat{P}} \left\| \widehat{\phi}_h(s_h,a_h) \right\|_{\widehat{\Lambda}_h^{-1}} \sqrt{N|\mathcal{A}|\mathbb{E}_{\rho_h^{+1}}\|\phi_{h+1}^\star(s_{h+1},a_{h+1})\|^2_{\left(\Lambda_{h+1}^\star\right)^{-1}} + N|\mathcal{A}|\varepsilon_{TV} + d}$$

$$\leq \mathbb{E}_{\pi,\widehat{P}} \left\| \widehat{\phi}_h(s_h,a_h) \right\|_{\widehat{\Lambda}_h^{-1}} \sqrt{d|\mathcal{A}| + N|\mathcal{A}|\varepsilon_{TV} + d},$$

where we used the fact that

$$\mathbb{E}_{\rho_h^{+1}}\|\phi_{h+1}^\star(s_{h+1},a_{h+1})\|^2_{\left(\Lambda_{h+1}^\star\right)^{-1}}$$

$$= \mathrm{Tr}\left( \mathbb{E}_{\rho_h^{+1}}\left[\phi_{h+1}^\star(s_{h+1},a_{h+1})\phi_{h+1}^\star(s_{h+1},a_{h+1})^\top\right] \left(N\mathbb{E}_{\rho_h^{+1}}\left[\phi_h^\star(s_h,a_h)\phi_h^\star(s_h,a_h)^\top\right] + \lambda I\right)^{-1}\right)$$

$$= \frac{1}{N}\mathrm{Tr}(I - M) \leq \frac{d}{N},$$

where $M$ is a positive definite matrix. Thus, doing an initial change from $d_{h+1}^\pi$ to $d_{\widehat{P},h+1}^\pi$ concludes the proof. $\qquad\square$

## G   REPRESENTATION TRANSFER

First, we prove Lemma 4.1, restated below.

**Lemma 4.1** *Suppose Assumption 3.4 and that for all source tasks $k \in [K-1]$ we have a $\lambda_{\min}$-exploratory policy $\pi_k$. Then, for any $\delta \in (0,1)$, learning features $\widehat{\phi}$ using the cross-sampling procedure of Algorithm 1 satisfies* (2) *w.p.* $1-\delta$. *Furthermore, for any $h = 0,1,...,H-1$, there exists $\widetilde{\mu}_h : \mathcal{S} \to \mathbb{R}^d$ such that for any function $g : \mathcal{S} \to [0,1]$, $\|\int g(s)\mathrm{d}\widetilde{\mu}_h(s)\|_2 \leq \bar{\alpha}\sqrt{d}$ and*

$$\sup_\pi \mathbb{E}_{\pi,P_K^\star} \left\| \widehat{\phi}_h(s_h,a_h)^\top \widetilde{\mu}_h(\cdot) - \phi_h^\star(s_h,a_h)^\top \mu_{K;h}^\star(\cdot) \right\|_{TV} \leq \varepsilon_{TV} \coloneqq \sqrt{|\mathcal{A}|\alpha_{\max}^3 K\zeta_n/\lambda_{\min}}.$$

*Proof of Lemma 4.1.* Fix an arbitrary $\pi$. Denote $\mu_h(s') = \sum_{k=0}^{K-1} \alpha_{k;h}(s')\widehat{\mu}_{k;h}(s')$. First, note that

$$\max_{g:\mathcal{S}\to[0,1]} \left\| \int \mu_h(s)g(s)\mathrm{d}(s) \right\|_2$$

$$\leq \max_{g:\mathcal{S}\to[0,1]} \sum_{k=0}^{K-1} \left\| \int \widehat{\mu}_{k;h}(s)\alpha_{k;h}(s)g(s)\mathrm{d}(s) \right\|_2$$

$$\leq \sum_{k=0}^{K-1} \max_s \alpha_{k;h}(s)\sqrt{d} \qquad\qquad \text{(Since } \int \widehat{\mu}_{k;h}(s)g(s)\mathrm{d}(s) \leq \sqrt{d} \text{ by 3.2)}$$

$$= \bar{\alpha}\sqrt{d}$$

For any $h = 0,1,...,H-1$, we have

$$\mathbb{E}_{\pi,P_K^\star} \left\| \widehat{\phi}_h(s_h,a_h)^\top \mu_h(\cdot) - \phi_h^\star(s_h,a_h)^\top \mu_{K;h}^\star(\cdot) \right\|_{TV}$$

$$= \mathbb{E}_{\pi,P_K^\star} \left[ \sum_{s_{h+1}} \left| \sum_{k=1}^{K-1} \alpha_{k;h}(s_{h+1})\left( \widehat{\phi}_h(s_h,a_h)^\top \widehat{\mu}_{k;h}(s_{h+1}) - \phi_h^\star(s_h,a_h)^\top \mu_{k;h}^\star(s_{h+1}) \right) \right| \right]$$

$$\leq \mathbb{E}_{\pi,P_K^\star} \left[ \sum_{s_{h+1}} \sum_{k=1}^{K-1} |\alpha_{k;h}(s_{h+1})| \left| \widehat{\phi}_h(s_h,a_h)^\top \widehat{\mu}_{k;h}(s_{h+1}) - \phi_h^\star(s_h,a_h)^\top \mu_{k;h}^\star(s_{h+1}) \right| \right]$$

$$\leq \alpha_{\max} \sum_{k=1}^{K-1} \mathbb{E}_{\pi,P_K^\star} \left\| \widehat{\phi}_h(s_h,a_h)^\top \widehat{\mu}_{k;h}(\cdot) - \phi_h^\star(s_h,a_h)^\top \mu_{k;h}^\star(\cdot) \right\|_{TV}.$$

First consider the case when $h = 0$. At $h = 0$, the distribution under $P_K^\star$ is the same as $\nu_{k,h}$, and so, we directly get that the above quantity is at most $\alpha_{\max}\zeta_n^{1/2} \leq \varepsilon$, which proves the $h = 0$ case.

Now consider any $h = 1, 2, ..., H - 1$. To simplify notation, let us denote

$$\text{err}_{k;h}(s_h, a_h) = \left\| \widehat{\phi}_h(s_h, a_h)^\top \widehat{\mu}_{k;h}(\cdot) - \phi_h^\star(s_h, a_h)^\top \mu_{k;h}^\star(\cdot) \right\|_{TV},$$

$$w_{k;h} = \int_{s_h} d\mu_{K;h-1}^\star(s_h) \mathbb{E}_{a_h \sim \pi_h(s_h)} \text{err}_{k;h}(s_h, a_h),$$

$$\Sigma_{k,h} = \mathbb{E}_{\pi_k, P_k^\star} \left[ \phi_h^\star(s_h, a_h) \phi_h^\star(s_h, a_h)^\top \right].$$

Note that $\lambda_{\min}(\Sigma_{k,h}) \geq \lambda_{\min}$ by assumption. Now continuing from where we left off, we take a one-step back as follows,

$$\alpha_{\max} \sum_{k=1}^{K-1} \mathbb{E}_{\pi, P_K^\star} \text{err}_{k;h}(s_h, a_h)$$

$$= \alpha_{\max} \sum_{k=1}^{K-1} \mathbb{E}_{\pi, P_K^\star} \left\langle \phi_{h-1}^\star(s_{h-1}, a_{h-1}), w_{k;h} \right\rangle$$

$$\leq \alpha_{\max} \sum_{k=1}^{K-1} \left( \mathbb{E}_{\pi, P_K^\star} \left\| \phi_{h-1}^\star(s_{h-1}, a_{h-1}) \right\|_{\Sigma_{k;h-1}^{-1}} \right) \left\| w_{k;h} \right\|_{\Sigma_{k;h-1}}$$

By $\lambda_{\min}$ guarantee of $\Sigma_{k,h}$, and Jensen's inequality to push the square inside,

$$\leq \frac{\alpha_{\max}}{\sqrt{\lambda_{\min}}} \sum_{k=1}^{K-1} \sqrt{\mathbb{E}_{s_{h-1}, a_{h-1} \sim \pi_k, P_k^\star} \mathbb{E}_{s_h \sim P_{K;h-1}^\star(s_{h-1}, a_{h-1}), a_h \sim \pi_h(s_h)} \text{err}_{k;h}(s_h, a_h)^2}$$

$$\leq \frac{A^{1/2} \alpha_{\max}}{\sqrt{\lambda_{\min}}} \sum_{k=1}^{K-1} \sqrt{\mathbb{E}_{s_{h-1}, a_{h-1} \sim \pi_k, P_k^\star} \mathbb{E}_{s_h \sim P_{K;h-1}^\star(s_{h-1}, a_{h-1}), a_h \sim \mathcal{U}(\mathcal{A})} \text{err}_{k;h}(s_h, a_h)^2}$$

By Assumption 3.4, the expectation over $P_{K;h-1}^\star$ is a linear combination of expectations over $P_{j;h-1}^\star$,

$$\leq \frac{A^{1/2} \alpha_{\max}^{3/2}}{\sqrt{\lambda_{\min}}} \sum_{k=1}^{K-1} \sqrt{\sum_{j=1}^{K-1} \mathbb{E}_{s_{h-1}, a_{h-1} \sim \pi_k, P_k^\star} \mathbb{E}_{s_h \sim P_{j;h-1}^\star(s_{h-1}, a_{h-1}), a_h \sim \mathcal{U}(\mathcal{A})} \text{err}_{k;h}(s_h, a_h)^2}$$

$$\leq \frac{A^{1/2} \alpha_{\max}^{3/2} K^{1/2}}{\sqrt{\lambda_{\min}}} \sqrt{\sum_{k=1}^{K-1} \sum_{j=1}^{K-1} \mathbb{E}_{s_{h-1}, a_{h-1} \sim \pi_k, P_k^\star} \mathbb{E}_{s_h \sim P_{j;h-1}^\star(s_{h-1}, a_{h-1}), a_h \sim \mathcal{U}(\mathcal{A})} \text{err}_{k;h}(s_h, a_h)^2}$$

$$\leq \frac{A^{1/2} \alpha_{\max}^{3/2} K^{1/2} \zeta_n^{1/2}}{\sqrt{\lambda_{\min}}},$$

where we used the MLE guarantee (2) in the last step. $\qquad \square$

---

**Algorithm 5** Transfer learning with online access

---

**PRE-TRAINING PHASE**

**Input:** num. LSVI-UCB episodes $N_{\text{LSVI-UCB}}$, num. model-learning episodes $N_{\text{REWARDFREE}}$, size of cross-sampled datasets $n$, failure probability $\delta$.

1: **for** source task $k = 1, ..., K - 1$ **do**
2:      Find policy cover $\pi_k = \text{REWARDFREE}(P_k^\star, N_{\text{LSVI-UCB}}, N_{\text{REWARDFREE}}, \delta)$. (Algorithm 2)
3: **for** source task $k = 1, ..., K - 1$ **do**
4:      For each $h = 0, 1, ..., H - 1$, sample $\mathcal{D}_k$ as $n$ i.i.d. $(s_h, a_h, s_{h+1})$ tuples from $\pi_k$.
5: For each $h = 0, 1, ..., H - 1$, learn $\widehat{\phi}_h = \text{Multi-task REPLEARN}(\{\mathcal{D}_{k;h}\}_{k \in [K-1]})$. (Algorithm 3)

**DEPLOYMENT PHASE**

**Additional Input:** number of deployment episodes $T$.

1: Set $\beta = H\sqrt{d} + \bar{\alpha} dH \sqrt{\log(dHT/\delta)}$.
2: Run LSVI-UCB $\left( \{\widehat{\phi}_h\}_{h=0}^{H-1}, r = r_K, T, \beta \right)$ in the target task $P_K^\star$ (Algorithm 6).

---

Next we state an analogous lemma for when we don't need generative access to the source task, but instead assume Assumption 5.1, and Assumption 5.2.

**Lemma G.1** *Suppose Assumption 5.1, and Assumption 5.2. Now take the setup of Lemma 4.1 with the only difference being that $\widehat{\phi}$ is learned as in Algorithm 5. Then, the same guarantee of Lemma 4.1 holds with a slightly different right hand side for the bound on the TV-error,*

$$\sup_\pi \mathbb{E}_{\pi, P_K^\star} \left\| \widehat{\phi}_h(s_h, a_h)^\top \widehat{\mu}_h(\cdot) - \phi_h^\star(s_h, a_h)^\top \mu_{K;h}^\star(\cdot) \right\|_{TV} \leq \frac{\alpha_{\max} K^{1/2} \zeta_n^{1/2}}{(\psi_{raw} \lambda_{\min})^{1/2}}.$$

*Proof of Lemma G.1.* Fix an arbitrary $\pi$. Denote $\mu_h(s') = \sum_{k=0}^{K-1} \alpha_{k;h}(s') \hat{\mu}_{k;h}(s')$. Then, some algebra with importance sampling gives us the bound,

$$\mathbb{E}_{\pi, P_K^\star} \left\| \widehat{\phi}_h(s_h, a_h)^\top \mu_h(\cdot) - \phi_h^\star(s_h, a_h)^\top \mu_{K;h}^\star(\cdot) \right\|_{TV}$$

$$\leq \mathbb{E}_{\pi, P_K^\star} \left[ \sum_{s_{h+1}} \left| \sum_{k=1}^{K-1} \alpha_{k;h}(s_{h+1}) \left( \widehat{\phi}_h(s_h, a_h)^\top \widehat{\mu}_{k;h}(s_{h+1}) - \phi_h^\star(s_h, a_h)^\top \mu_{k;h}^\star(s_{h+1}) \right) \right| \right]$$

$$\leq \alpha_{\max} \sum_{k=1}^{K-1} \mathbb{E}_{\pi, P_K^\star} \left\| \widehat{\phi}_h(s_h, a_h)^\top \widehat{\mu}_{k;h}(\cdot) - \phi_h^\star(s_h, a_h)^\top \mu_{k;h}^\star(\cdot) \right\|_{TV}$$

$$\leq \alpha_{\max} K^{1/2} \sqrt{\sum_{k=1}^{K-1} \mathbb{E}_{\pi, P_K^\star} \left\| \widehat{\phi}_h(s_h, a_h)^\top \widehat{\mu}_{k;h}(\cdot) - \phi_h^\star(s_h, a_h)^\top \mu_{k;h}^\star(\cdot) \right\|_{TV}^2}$$

By Assumption 5.1, Assumption 5.2, for any $s, a$, we have $\frac{d_{K;h}^\pi(s,a)}{d_{k;h}^{\pi_k}(s,a)} \leq \frac{1}{\psi_{raw} \lambda_{\min} \left( \mathbb{E}_{\pi_k, P_k^\star} [\phi_h^\star(s_h, a_h) \phi_h^\star(s_h, a_h)^\top] \right)} \leq \frac{1}{\psi_{raw} \lambda_{\min}}$, where we used the coverage-under-$\pi_k$ assumption in the last inequality. In other words, for each $k = 1, 2, ..., K-1$, we have $\left\| \frac{\mathrm{d} d_{K;h}^\pi}{\mathrm{d} d_{k;h}^{\pi_k}} \right\|_\infty \leq \frac{1}{\psi_{raw} \lambda_{\min}}$, hence we can importance sample,

$$\leq \frac{\alpha_{\max} K^{1/2}}{(\psi_{raw} \lambda_{\min})^{1/2}} \sqrt{\sum_{k=1}^{K-1} \mathbb{E}_{\pi_k, P_k^\star} \left\| \widehat{\phi}_h(s_h, a_h)^\top \widehat{\mu}_{k;h}(\cdot) - \phi_h^\star(s_h, a_h)^\top \mu_{k;h}^\star(\cdot) \right\|_{TV}^2}$$

$$\leq \frac{\alpha_{\max} K^{1/2} \zeta_n^{1/2}}{(\psi_{raw} \lambda_{\min})^{1/2}}.$$

$\square$

# H  PROOFS FOR DEPLOYMENT PHASE

## H.1  ONLINE RL LEMMAS

**Lemma H.1** (Self-normalized Martingale) *Consider filtrations $\{F_i\}_{i=1,2,...}$, so that $\mathbb{E}[\varepsilon_i \mid F_{i-1}] = 0$ and $\{\varepsilon_i \mid F_{i-1}\}_{i=1,2,...}$ are sub-Gaussian with parameter $\sigma^2$. Let $\{X_i\}_{i=1,2,...}$ be random variables in a hilbert space $\mathcal{H}$. Suppose a linear operator $\Sigma_0 : \mathcal{H} \to \mathcal{H}$ is positive definite. For any $t$, define $\Sigma_t = \Sigma_0 + \sum_{i=1}^t X_i X_i^T$. Then w.p. at least $1 - \delta$, we have,*

$$\forall t \geq 1 : \left\| \sum_{i=1}^t X_i \varepsilon_i \right\|_{\Sigma_t^{-1}}^2 \leq \sigma^2 \log \left( \frac{\det(\Sigma_t) \det(\Sigma_0)^{-1}}{\delta^2} \right).$$

*Proof.* Lemma A.8 of (Agarwal et al., 2019). $\square$

**Lemma H.2** *Let $\Lambda_t = \lambda I + \sum_{i=1}^t x_i x_i^T$ for $x_i \in \mathbb{R}^d$ and $\lambda > 0$. Then $\sum_{i=1}^t x_i^T (\Lambda_t)^{-1} x_i \leq d$.*

*Proof.* Lemma D.1 of (Jin et al., 2020b). $\square$

## H.2 PROOF OF MAIN RESULTS

Let $(x)_{\leq y}$ refer to the clamping operator, i.e. $(x)_{\leq y} = \min\{x, y\}$. Let $M_V$ be the maximum possible value in the MDP with the given reward function.

---

**Algorithm 6** LSVI-UCB

---

1: **Input:** Features $\{\widehat{\phi}_h\}_{h=0,1,...,H-1}$, reward $\{r_h\}_{h=0,1,...,H-1}$, number of episodes $N$, bonus scaling parameter $\beta$, UNIFORMACTIONS = FALSE.
2: **for** episode $e = 1, 2, ..., N$ **do**
3:     Initialize $\widehat{V}_{H,e}(s) = 0, \forall s$
4:     **for** step $h = H - 1, H - 2..., 0$ **do**
5:         Learn best predictor for $\widehat{V}_{h+1}^e$,

$$\Lambda_{h,e} = \sum_{k=1}^{e-1} \widehat{\phi}_h(s_h^k, a_h^k)\widehat{\phi}_h(s_h^k, a_h^k)^\top + I,$$

$$\widehat{w}_{h,e} = \Lambda_{h,e}^{-1} \sum_{k=1}^{e-1} \widehat{\phi}_h(s_h^k, a_h^k)\widehat{V}_{h+1,e}(s_{h+1}^k).$$

6:         Set bonus and value functions,

$$b_{h,e}(s, a) = \left\|\widehat{\phi}_h(s, a)\right\|_{\Lambda_{h,e}^{-1}},$$

$$\widehat{Q}_{h,e}(s, a) = \widehat{w}_{h,e}^\top \widehat{\phi}_h(s, a) + r_h(s, a) + \beta b_{h,e}(s, a),$$

$$\widehat{V}_{h,e}(s) = \left(\max_a \left\{\widehat{Q}_{h,e}(s, a)\right\}\right)_{\leq M_V}.$$

7:         Set $\pi_h^e(s) = \text{argmax}_a \widehat{Q}_{h,e}(s, a)$.
8:     Execute $\pi^e$ to collect a trajectory $(s_h^e, a_h^e)_{h=0}^{H-1}$.
9:     If UNIFORMACTIONS = TRUE, discard $a_h^e$ and draw freshly sampled uniform actions independently for all $h$, i.e. $a_h^e \sim \text{Unif}(\mathcal{A})$.
10: **Return:** uniform mixture $\rho = \text{Uniform}(\{\pi^e\}_{e=1}^N)$.

---

Previously, Jin et al. (2020b) analyzed LSVI-UCB under point-wise model-misspecification. Here, we show that similar guarantees hold under a more general *policy-distribution* model-misspecification $\varepsilon_{ms}$, captured by Assumption H.1.

**Assumption H.1** *Suppose for every $h = 0, 1, ..., H - 1$, there exist $\widetilde{\mu}_h$ such that for any policy $\pi$,*

$$\mathbb{E}_\pi \left[\left\|\widetilde{\mu}_h(\cdot)^T \widehat{\phi}_h(s_h, a_h) - P_h^\star(\cdot \mid s_h, a_h)\right\|_{TV}\right] \leq \varepsilon_{ms}.$$

*We further assume that $\sup_{s,a,h} \left\|\widetilde{\mu}_h(\cdot)^T \widehat{\phi}(s, a)\right\|_{TV} \leq M_\mu$ and $\|f^T \widetilde{\mu}_h\|_2 \leq M_\mu\sqrt{d}\|f\|_\infty \ \forall f : \mathcal{S} \to \mathbb{R}$, for some positive constant $M_\mu$.*

In other words, we only need the model to be accurate *on average* under the occupancy distributions realizable by policies. We also make a slight generalization on the regularization constant $M_\mu$, which is set to 1 in the original linear MDP definition (Jin et al., 2020b). Later, we will later instantiate the above assumption with our transferred $\widetilde{\mu}_h(s') = \sum_{k=1}^{K-1} \alpha_{k;h}(s')\widehat{\mu}_{k;h}(s')$, then for any $s, a$, we have

$$\|\widetilde{\mu}_h \widehat{\phi}_h(s, a)\|_{TV} = \sum_{s'} \left|\sum_{k=1}^{K-1} \alpha_{k;h}(s')\widehat{\mu}_{k;h}(s')^T \widehat{\phi}_h(s, a)\right|$$

$$\leq \sum_{s'} \sum_{k=1}^{K-1} |\alpha_{k;h}(s')||\widehat{\mu}_{k;h}(s')^T \widehat{\phi}_h(s, a)|$$

$$\leq \sum_{k=1}^{K-1} \max_{s'} |\alpha_{k;h}(s')| \qquad \qquad (\text{by } \left\|\widehat{\mu}_{k;h}\widehat{\phi}_h(s, a)\right\|_{TV} \leq 1)$$

$$\le \bar{\alpha}.$$

Also,

$$
\begin{aligned}
\|f^T \widetilde{\mu}_h\|_2 &= \left\| \sum_{s'} \sum_{k=1}^{K-1} \alpha_{k;h}(s') \widehat{\mu}_{k;h}(s') f(s') \right\|_2 \\
&= \sum_{k=1}^{K-1} \max_{s'} |\alpha_{k;h}(s')| \left\| \sum_{s'} \widehat{\mu}_{k;h}(s') f(s') \right\|_2 \\
&\le \bar{\alpha} \sqrt{d} \|f\|_\infty. \qquad \text{(by } \|f^T \widehat{\mu}_{k;h}\|_2 \le \sqrt{d} \|f\|_\infty)
\end{aligned}
$$

So we will set $M_\mu = \bar{\alpha}$.

Note that we only need the existence of $\widetilde{\mu}_h$ here, and $\widetilde{\mu}_h(\cdot)^T \widehat{\phi}_h(s, a)$ need not be a valid probability kernel. In fact, it may even be negative valued.

In this section, we make a model-based analysis of LSVI. Similar approaches have been used in prior works, e.g. Lykouris et al. (2021); Agarwal et al. (2019); Zhang et al. (2022). For simplicity, we suppose that $\mathcal{S}$ is finite, but may be exponentially large, as we suffer no dependence on $|\mathcal{S}|$. The proof can be easily extended to infinite state spaces by replacing inner products with $P$ by integrals.

Consider the following quantity,

$$
\widehat{\mu}_{h,e} = \left( \sum_{k=1}^{e-1} \delta(s_{h+1}^k) \widehat{\phi}_h(s_h^k, a_h^k)^T \right) (\Lambda_{h,e})^{-1} \in \operatorname*{argmin}_{\mu \in \mathbb{R}^{S \times d}} \sum_{k=1}^{e-1} \| \mu \widehat{\phi}_h(s_h^i, a_h^i) - \delta(s_{h+1}^i) \|_2 + \|\mu\|_F^2,
$$

where $\delta(s)$ is a one-hot encoding of the state $s$. In words, this is the best choice for linearly (in $\widehat{\phi}_h(s, a)$) predicting $\mathbb{E}_{s' \sim P_h^\star(s,a)}[\delta(s')] = P_h^\star(s' \mid s, a)$. We highlight that this is just a quantity for analysis and not computed in the algorithm. Finally, denote

$$
\begin{aligned}
\widehat{P}_{h,e} &= \widehat{\mu}_{h,e} \widehat{\phi}_h, \\
\widetilde{P}_h &= \widetilde{\mu}_h \widehat{\phi}_h.
\end{aligned}
$$

We will also sometimes use the shorthand $Pf(s, a)$ for $\mathbb{E}_{s' \sim P(\cdot|s,a)}[f(s')]$.

For each $h = 0, 1, ..., H - 1$, let $\mathcal{V}_h$ denote the class of functions

$$
\left\{ s \mapsto \left( \max_a \left\{ w^T \widehat{\phi}_h(s, a) + r_h(s, a) + \widetilde{\beta} \| \widehat{\phi}_h(s, a) \|_{\Lambda^{-1}} \right\} \right)_{\le M_V} \middle| \|w\|_2 \le N M_V, \widetilde{\beta} \in [0, B], \Lambda \succeq I \text{ symmetric} \right\}
$$

The motivation behind this construction is that $\mathcal{V}_h$ satisfies the key property that all of the learned value functions $\widehat{V}_{h,e}$ during Algorithm 6 are captured in this class.

**Lemma H.3** *For any $h = 0, 1, ..., H - 1$,*

1. $\sup_s \left| \widehat{V}_{h,e}(s) \right| \le M_V$.

2. *For any $e = 1, 2, ..., N$, we have $\widehat{V}_{h,e} \in \mathcal{V}_h$.*

3. $\forall f \in \mathcal{V}_h$, *we have $\sup_s |f(s)| \le M_V$.*

*Proof.* Recall that

$$
\widehat{V}_{h,e}(s) = \left( \max_a \left\{ \widehat{w}_{h,e}^T \widehat{\phi}_h(s, a) + r_h(s, a) + \beta b_{h,e}(s, a) \right\} \right)_{\le M_V}
$$

$$
\text{where } \widehat{w}_{h,e} = \Lambda_{h,e}^{-1} \sum_{k=1}^{e-1} \widehat{\phi}_h(s_h^k, a_h^k) \widehat{V}_{h+1,e}(s_{h+1}^k).
$$

From the thresholding, we have

$$
\left| \widehat{V}_{h,e}(s) \right| \le M_V.
$$

We can bound the norm of $\widehat{w}_{h,e}$ as follows,

$$\|\widehat{w}_{h,e}\| \leq \left\|\Lambda_{h,e}^{-1}\right\|_2 \sum_{k=1}^{e-1} \left|\widehat{V}_{h+1,e}(s_{h+1}^k)\right| \leq N \sup_s \left|\widehat{V}_{h+1,e}(s)\right| \leq NM_V.$$

We also required $\beta \leq B$, and we regularized the covariance with $I$, so $\lambda_{\min}$ is at least 1. Hence $\widehat{V}_{h,e}$ satisfies all the conditions to be in $\mathcal{V}_h$. $\qquad\square$

Now we control the metric entropy of $\mathcal{V}_h$ in $\ell_\infty$, i.e. $d(f_1, f_2) = \sup_s |f_1(s) - f_2(s)|$ for $f_i \in \mathcal{V}_h$.

**Lemma H.4** *Let $\varepsilon > 0$ be arbitrary and let $\mathcal{N}_\varepsilon$ be the smallest $\varepsilon$-net with $\ell_\infty$ of $\mathcal{V}_h$. Then,*

$$\log|\mathcal{N}_\varepsilon| \leq d\log(1 + 6L/\varepsilon) + \log(1 + 6B/\varepsilon) + d^2\log(1 + 18B^2\sqrt{d}/\varepsilon^2).$$

*Proof.* Let $f_1, f_2 \in \mathcal{V}_h$. Then,

$|f_1(s) - f_2(s)|$

$\leq \max_a \left|(w_1 - w_2)^T \widehat{\phi}_h(s,a) + \beta_1 \left\|\widehat{\phi}_h(s,a)\right\|_{\Lambda_1^{-1}} - \beta_2 \left\|\widehat{\phi}_h(s,a)\right\|_{\Lambda_2^{-1}}\right|$

$\leq \|w_1 - w_2\|_2 + \max_a \left|(\beta_1 - \beta_2)\left\|\widehat{\phi}_h(s,a)\right\|_{\Lambda_1^{-1}}\right| + \beta_2 \max_a \left|\left\|\widehat{\phi}_h(s,a)\right\|_{\Lambda_1^{-1}} - \left\|\widehat{\phi}_h(s,a)\right\|_{\Lambda_2^{-1}}\right|$

$\leq \|w_1 - w_2\|_2 + |\beta_1 - \beta_2| + B\max_a \sqrt{\left|\left\|\widehat{\phi}_h(s,a)\right\|_{\Lambda_1^{-1}} - \left\|\widehat{\phi}_h(s,a)\right\|_{\Lambda_2^{-1}}\right|} \qquad (\lambda_{\min}(\Lambda_1) \geq 1)$

$\leq \|w_1 - w_2\|_2 + |\beta_1 - \beta_2| + B\sqrt{\left\|\Lambda_1^{-1} - \Lambda_2^{-1}\right\|_2},$

where we used for any $a, b \geq 0$, we have $\left|\sqrt{a} - \sqrt{b}\right| = \frac{\sqrt{|a-b|}}{\sqrt{a}+\sqrt{b}}\sqrt{|a-b|} \leq \sqrt{|a-b|}$. Now proceeding like the Lemma 8.6 in the RL Theory Monograph (Agarwal et al., 2019), we have the result. $\qquad\square$

In this section, we'll use the following bonus scaling parameter,

$$\beta := \mathcal{O}\left(\sqrt{Nd}\varepsilon_{ms}M_V + M_V M_\mu d\sqrt{\log(dNM_V/\delta)}\right). \qquad (7)$$

The following high probability event ($\mathcal{E}_{model}$) is a key step in our proof. Essentially, Theorem H.1 guarantees that, for all functions in $\mathcal{V}_h$, the model we learn is an accurate predictor of the expectation, up to a bonus and some vanishing terms.

For all the following lemmas and theorems, suppose Assumption H.1 and the bonus scaling $\beta$ is set as in (7). Throughout the section, $\zeta_h(\tau_h)$ refers to indicator functions of the trajectory $\tau_h$, where $\tau_h = (s_0, s_1, ..., s_h)$. As before, the expectations $\mathbb{E}_\pi[g(\tau_h)]$ are with respect to the distribution of trajectories when $\pi$ is executed in the environment $P^\star$.

**Theorem H.1** *Let $\delta \in (0,1)$. Then, w.p. $1 - \delta$, for any time $h$, episode $e$, indicator functions $\zeta_1, \ldots, \zeta_H$, and policy $\pi$, we have*

$$\sup_{f \in \mathcal{V}} \left|\mathbb{E}_\pi\left[\left(\widehat{P}_{h,e}(s_h, a_h) - P_h^\star(s_h, a_h)\right)f\zeta_h(\tau_h)\right]\right| \leq \beta\mathbb{E}_\pi\left[b_h^e(s_h, a_h)\zeta_h(\tau_h)\right] + \|\mathcal{V}_h\|_\infty \varepsilon_{ms}.$$

$$(\mathcal{E}_{model})$$

*Proof.* Condition on the outcome of Lemma H.5, which implies that w.p. $1 - \delta$, for any $h, e, \pi, \zeta_h$, we have

$$\sup_{f \in \mathcal{V}_h} \left|\mathbb{E}_\pi\left[\left(\widehat{P}_{h,e}(s_h, a_h) - \widetilde{P}_h(s_h, a_h)\right)f\zeta_h(\tau_h)\right]\right| \leq \beta\mathbb{E}_\pi\left[b_h^e(s_h, a_h)\zeta_h(\tau_h)\right].$$

Also, for any $h, e, \pi, \zeta_h$, by Assumption H.1, we have (w.p. 1) that

$$\sup_{f \in \mathcal{V}_h} \left|\mathbb{E}_\pi\left[\left(\widetilde{P}_h(s_h, a_h) - P_h^\star(s_h, a_h)\right)f\zeta_h(\tau_h)\right]\right| \leq \mathbb{E}_\pi\left[\sup_{f \in \mathcal{V}_h}\left|\left(\widetilde{P}_h(s_h, a_h) - P_h^\star(s_h, a_h)\right)f\right|\zeta_h(\tau_h)\right]$$

$$\leq \mathbb{E}_\pi \left[ \sup_{f \in \mathcal{V}_h} \left| \left( \widetilde{P}_h(s_h, a_h) - P_h^\star(s_h, a_h) \right) f \right| \right]$$

$$\leq \|\mathcal{V}_h\|_\infty \varepsilon_{ms}.$$

Combining these two yields the result, as

$$\sup_{f \in \mathcal{V}_h} \left| \mathbb{E}_\pi \left[ \left( \widehat{P}_{h,e}(s_h, a_h) - P_h^\star(s_h, a_h) \right) f \zeta_h(\tau_h) \right] \right|$$

$$\leq \sup_{f \in \mathcal{V}_h} \left| \mathbb{E}_\pi \left[ \left( \widetilde{P}_h(s_h, a_h) - P_h^\star(s_h, a_h) \right) f \zeta_h(\tau_h) \right] \right| + \sup_{f \in \mathcal{V}_h} \left| \mathbb{E}_\pi \left[ \left( \widehat{P}_{h,e}(s_h, a_h) - \widetilde{P}_h(s_h, a_h) \right) f \zeta_h(\tau_h) \right] \right|.$$

$\square$

**Lemma H.5** *Suppose [Assumption H.1](#) and the bonus scaling $\beta$ is set as in [(7)](#). For any $\delta \in (0,1)$, w.p. at least $1 - \delta$, we have for any time $h$, episode $e$, and policy $\pi$,*

$$\forall s_h, a_h : \sup_{f \in \mathcal{V}_h} \left| \left( \widehat{P}_{h,e}(s_h, a_h) - \widetilde{P}_h(s_h, a_h) \right) f \right| \leq \beta b_{h,e}(s_h, a_h).$$

*Proof.* Consider any $h, e, \pi$. Define $\varepsilon_h^k := -\delta(s_{h+1}^k) + P_h^\star(s_{h+1}^k | s_h^k, a_h^k)$, so that $\mathbb{E}[\varepsilon_h^k \mid \mathcal{H}_{k-1}] = 0$, where $\mathcal{H}_{k-1}$ contains the states and actions before episode $k$. In what follows, we slightly abuse notation, as $P(s,a)\widehat{\phi}^T(s,a)$ will denote the outer product, and hence a $\mathbb{R}^{S \times d}$ quantity.

$$\widehat{\mu}_{h,e}\Lambda_{h,e} = \sum_{k=1}^{e-1} \delta(s_{h+1}^k)\widehat{\phi}_h(s_h^k, a_h^k)^T$$

$$= \sum_{k=1}^{e-1} \left( P_h^\star(s_h^k, a_h^k) - \widetilde{P}_h(s_h^k, a_h^k) \right) \widehat{\phi}_h(s_h^k, a_h^k)^T + \sum_{k=0}^{e-1} \left( \widetilde{P}_h(s_h^k, a_h^k) - \varepsilon_h^k \right) \widehat{\phi}_h(s_h^k, a_h^k)^T$$

$$= \sum_{k=1}^{e-1} \left( P_h^\star(s_h^k, a_h^k) - \widetilde{P}_h(s_h^k, a_h^k) \right) \widehat{\phi}_h(s_h^k, a_h^k)^T + \widetilde{\mu}_h(\Lambda_{h,e} - I) - \left( \sum_{k=0}^{e-1} \varepsilon_h^k \widehat{\phi}_h(s_h^k, a_h^k)^T \right).$$

Rearranging, we have

$$\widehat{\mu}_{h,e} - \widetilde{\mu}_h = \left( \sum_{k=0}^{e-1} \left( P_h^\star(s_h^k, a_h^k) - \widetilde{P}_h(s_h^k, a_h^k) \right) \widehat{\phi}_h(s_h^k, a_h^k)^T \right) (\Lambda_{h,e})^{-1}$$

$$- \widetilde{\mu}_h (\Lambda_{h,e})^{-1} - \left( \sum_{k=0}^{e-1} \varepsilon_h^k \widehat{\phi}_h(s_h^k, a_h^k)^T \right) (\Lambda_{h,e})^{-1}.$$

Now let $f \in \mathcal{V}_h$ be arbitrary. For any $s_h, a_h$, multiply the above with $\widehat{\phi}_h(s_h, a_h)$ and multiply with $f$, we have

$$\left| \left( \widehat{P}_{h,e}(s_h, a_h) - \widetilde{P}_h(s_h, a_h) \right) f \right|$$

$$= \left| f^T \left( \widehat{\mu}_{h,e} - \widetilde{\mu}_h \right) \widehat{\phi}_h(s_h, a_h) \right|$$

$$\leq \underbrace{\left| f^T \left( \sum_{k=1}^{e-1} \left( P_h^\star(s_h^k, a_h^k) - \widetilde{P}_h(s_h^k, a_h^k) \right) \widehat{\phi}_h(s_h^k, a_h^k)^T \right) \Lambda_{h,e}^{-1} \widehat{\phi}_h(s_h, a_h) \right|}_{\text{Term(a)}}$$

$$+ \underbrace{\left| f^T \widetilde{\mu}_h \Lambda_{h,e}^{-1} \widehat{\phi}_h(s_h, a_h) \right|}_{\text{Term(b)}}$$

$$+ \underbrace{\left| f^T \left( \sum_{k=1}^{e-1} \varepsilon_h^k \widehat{\phi}_h(s_h^k, a_h^k)^T \right) \Lambda_{h,e}^{-1} \widehat{\phi}_h(s_h, a_h) \right|}_{\text{Term(c)}}.$$

We can deterministically bound Term (b) as follows,

$$\sup_{f \in \mathcal{V}_h} \left| f^T \widetilde{\mu}_h \Lambda_{h,e}^{-1} \widehat{\phi}_h(s_h, a_h) \right|$$

$$= \sup_{f \in \mathcal{V}_h} \left| (\Lambda_{h,e}^{-1/2} f^T \widetilde{\mu}_h)^T \left( \Lambda_{h,e}^{-1/2} \widehat{\phi}_h(s_h, a_h) \right) \right|$$

$$\leq \sup_{f \in \mathcal{V}_h} \left\| \Lambda_{h,e}^{-1/2} \right\|_2 \| f^T \widetilde{\mu}_h \|_2 b_{h,e}(s_h, a_h)$$

$$\leq \| \mathcal{V}_h \|_\infty M_\mu \sqrt{d} b_{h,e}(s_h, a_h). \qquad \text{(by Assumption H.1)}$$

This term will be lower order compared to the other two.

We now derive the bound for Term (c) for any fixed $f \in \mathcal{V}_h$. Observe that

$$\left| f^T \left( \sum_{k=1}^{e-1} \varepsilon_h^k \widehat{\phi}_h(s_h^k, a_h^k)^T \right) \Lambda_{h,e}^{-1} \widehat{\phi}_h(s_h, a_h) \right| = \left| \left( \Lambda_{h,e}^{-1/2} \sum_{k=1}^{e-1} \widehat{\phi}_h(s_h^k, a_h^k)(f^T \varepsilon_h^k) \right)^T \left( \Lambda_{h,e}^{-1/2} \widehat{\phi}_h(s_h, a_h) \right) \right|$$

$$\leq \left\| \sum_{k=1}^{e-1} \widehat{\phi}_h(s_h^k, a_h^k)(f^T \varepsilon_h^k) \right\|_{\Lambda_{h,e}^{-1}} b_{h,e}(s_h, a_h).$$

Now we argue w.p. $1 - \delta$, for any $e, h$ we have

$$\left\| \sum_{k=1}^{e-1} \widehat{\phi}_h(s_h^k, a_h^k)(f^T \varepsilon_h^k) \right\|_{\Lambda_{h,e}^{-1}} \leq \left( 2\| \mathcal{V}_h \|_\infty \sqrt{2 \log(1/\delta) + d \log(N+1)} \right),$$

which implies the claim about all $s_h, a_h$. Indeed, we can apply Lemma H.1. Checking the preconditions, $\mathbb{E}_{P_h^\star(s_h,a_h)} \left[ f^T \varepsilon_h^k \mid \mathcal{H}_{k-1} \right] = 0$, $\sigma \leq |f^T \varepsilon_h^k| \leq \| f \|_\infty \| \varepsilon_h^k \|_1 \leq 2\| \mathcal{V}_h \|_\infty$, $\det(\Sigma_0) = \det I = 1$, and $\det(\Sigma_t) = \det(\Lambda_{h,e}) \leq (e+1)^d$ since the largest eigenvalue is $e + 1$. So, w.p. at least $1 - \delta$, for all $e$, we have the above inequality.

Thus, for any fixed $f \in \mathcal{V}_h$, w.p. $1 - \delta$, for all $e, h$ we have,

$$\left| \left( \widehat{P}_{h,e}(s_h, a_h) - \widetilde{P}_h(s_h, a_h) \right) f \right|$$

$$\leq \text{Term(a)} + \text{Term(b)} + \text{Term(c)}$$

$$\leq \left( 4\| \mathcal{V}_h \|_\infty (1 + M_\mu) \sqrt{\log(1/\delta) + d \log(N)} + \sqrt{dN} \| \mathcal{V}_h \|_\infty \varepsilon_{ms} \right) b_{h,e}(s_h, a_h)$$

$$+ \left( \| \mathcal{V}_h \|_\infty M_\mu \sqrt{d} \right) b_{h,e}(s_h, a_h)$$

$$+ \left( 4\| \mathcal{V}_h \|_\infty \sqrt{\log(1/\delta) + d \log(N)} \right) b_{h,e}(s_h, a_h)$$

$$\lesssim \left( \sqrt{dN} \| \mathcal{V}_h \|_\infty \varepsilon_{ms} + \| \mathcal{V}_h \|_\infty M_\mu \sqrt{\log(1/\delta) + d \log(N)} \right) b_{h,e}(s_h, a_h).$$

Now we apply a covering argument. Namely, union bound the above argument to every element in an $\varepsilon_{net}$-net of $\mathcal{V}_h$. For any $f \in \mathcal{V}_h$, let $\widetilde{f}$ be its neighbor in the net s.t. $\| \widetilde{f} - f \|_\infty \leq \varepsilon_{net}$, so we have

$$\left| \left( \widehat{P}_{h,e}(s_h, a_h) - \widetilde{P}_h(s_h, a_h) \right) f \right| \leq \left| \left( \widehat{P}_{h,e}(s_h, a_h) - \widetilde{P}_h(s_h, a_h) \right) \widetilde{f} \right| + \left| \left( \widehat{P}_{h,e}(s_h, a_h) - \widetilde{P}_h(s_h, a_h) \right) (\widetilde{f} - f) \right|$$

and

$$\left| \left( \widehat{P}_{h,e}(s_h, a_h) - \widetilde{P}_h(s_h, a_h) \right) (\widetilde{f} - f) \right| \lesssim \| \widetilde{f} - f \|_\infty (N + 1) \lesssim \varepsilon_{net} N.$$

Setting $\varepsilon_{net} = N$, the metric entropy is of the order $d \log(N(M_V + B)) + \log(BN) + d^2 \log(BdN)$. The error incurred with this epsilon net is a constant, which is lower order.

Thus, we have

$$\forall s_h, a_h : \sup_{f \in \mathcal{V}_h} \left| \left( \widehat{P}_{h,e}(s_h, a_h) - \widetilde{P}_h(s_h, a_h) \right) f \right|$$

$$\lesssim \left(\sqrt{dN}\|\mathcal{V}_h\|_\infty \varepsilon_{ms} + \|\mathcal{V}_h\|_\infty M_\mu \sqrt{\log(1/\delta) + d\log(M_V) + d^2\log(BdN)}\right) b_{h,e}(s_h, a_h)$$

$$\lesssim \left(\sqrt{dN} M_V \varepsilon_{ms} + M_V M_\mu \sqrt{\log(1/\delta) + d\log(M_V) + d^2\log(BdN)}\right) b_{h,e}(s_h, a_h)$$

Note that $\beta$ scales as $\sqrt{\log B}$, so one can find a valid $B$ by solving $\beta \leq B$ for $B$.

$\square$

**Lemma H.6** *Let $f \in \mathcal{V}_h$. For any $\delta \in (0,1)$, w.p. at least $1 - \delta$, for any time $h$, episode $e$, we have*

$$\forall s_h, a_h : \left| f^T \left(\sum_{k=1}^{e-1} P_h^\star(s_h^k, a_h^k) - \widetilde{P}_h(s_h^k, a_h^k)\right) \widehat{\phi}_h(s_h^k, a_h^k)^T \Lambda_{h,e}^{-1} \widehat{\phi}_h(s_h, a_h)\right|$$

$$\leq \left(4\|\mathcal{V}_h\|_\infty (1 + M_\mu) \sqrt{\log(1/\delta) + d\log(N)} + \sqrt{dN}\|\mathcal{V}_h\|_\infty \varepsilon_{ms}\right) b_{h,e}(s_h, a_h).$$

*Proof.* First observe that

$$\left| f^T \left(\sum_{k=1}^{e-1} (P_h^\star(s_h^k, a_h^k) - \widetilde{\mu}_h \widehat{\phi}_h(s_h^k, a_h^k)) \widehat{\phi}_h(s_h^k, a_h^k)^T\right) \Lambda_{h,e}^{-1} \widehat{\phi}_h(s_h, a_h)\right|$$

$$= \left| \left(\Lambda_{h,e}^{-1/2} \sum_{k=1}^{e-1} \widehat{\phi}_h(s_h^k, a_h^k) f^T (P_h^\star(s_h^k, a_h^k) - \widetilde{\mu}_h \widehat{\phi}_h(s_h^k, a_h^k))\right)^T \left(\Lambda_{h,e}^{-1/2} \widehat{\phi}_h(s_h, a_h)\right)\right|$$

$$\leq \left\|\sum_{k=1}^{e-1} \widehat{\phi}_h(s_h^k, a_h^k) \widetilde{\varepsilon}_k\right\|_{\Lambda_{h,e}^{-1}} b_{h,e}(s_h, a_h),$$

where $\widetilde{\varepsilon}_k = \left(P_h^\star(s_h^k, a_h^k) - \widetilde{P}_h(s_h^k, a_h^k)\right) f$.

Now we will argue that w.p. $1 - \delta$, for all $e, h$,

$$\left\|\sum_{k=1}^{e-1} \widehat{\phi}_h(s_h^k, a_h^k) \widetilde{\varepsilon}_k\right\|_{\Lambda_{h,e}^{-1}} \leq \left(4\|\mathcal{V}_h\|_\infty (1 + M_\mu) \sqrt{\log(1/\delta) + d\log(N)} + \sqrt{dN}\|\mathcal{V}_h\|_\infty \varepsilon_{ms}\right),$$

which will imply the claim for all $s_h, a_h$.

Apply self-normalized martingale concentration (Lemma H.1) to $X_i = \widehat{\phi}_h(s_h^i, a_h^i)$ and $\varepsilon_i = \widetilde{\varepsilon}_i - \mathbb{E}\left[\widetilde{\varepsilon}_i \mid \mathcal{H}_{i-1}\right]$, where the expectation is over $(s_h^i, a_h^i)$ in the definition of $\widetilde{\varepsilon}_i$. To see sub-Gaussianity, bound the envelope, $|\widetilde{\varepsilon}_i| \leq \|f\|_\infty \|P_h^\star(s_h^k, a_h^k) - \widetilde{\mu}_h \widehat{\phi}_h(s_h^k, a_h^k)\|_{TV} \leq \|\mathcal{V}_h\|_\infty (1 + M_\mu)$, and thus $\sigma \leq |\widetilde{\varepsilon}_k| \leq 2\|\mathcal{V}_h\|_\infty (1 + M_\mu)$. Now compute the determinants: $\det(\Lambda_{h,0}) = 1$ and since $\lambda_{\max}(\Lambda_{h,e}) \leq e + 1$, we have that $\log \det(\Lambda_{h,e}) \leq d\log(e + 1)$. Hence, w.p. at least $1 - \delta$, we have

$$\forall e : \left\|\sum_{k=1}^{e-1} \widehat{\phi}_h(s_h^k, a_h^k) \left(\widetilde{\varepsilon}_k - \mathbb{E}\left[\widetilde{\varepsilon}_k \mid \mathcal{H}_{k-1}\right]\right)\right\|_{\Lambda_{h,e}^{-1}} \leq 2\|\mathcal{V}_h\|_\infty (1 + M_\mu) \sqrt{2\log(1/\delta) + d\log(N + 1)}.$$

By Assumption H.1 applied to $\pi^k$ (the data-generating policy for episode $k$), we have $|\mathbb{E}\left[\widetilde{\varepsilon}_k \mid \mathcal{H}_{k-1}\right]| \leq \|\mathcal{V}_h\|_\infty \varepsilon_{ms}$. Recall for any scalars $c_i$ and vectors $x_i$, we have $\|\sum_i c_i x_i\| \leq \sum_i |c_i| \|x_i\| \leq \sqrt{\sum_i c_i^2} \sqrt{\sum_i \|x_i\|^2}$. Thus,

$$\left\|\sum_{k=1}^{e-1} \widehat{\phi}_h(s_h^k, a_h^k) \mathbb{E}\left[\widetilde{\varepsilon}_k \mid \mathcal{H}_{k-1}\right]\right\|_{\Lambda_{h,e}^{-1}}$$

$$\leq \sqrt{\sum_{k=1}^{e-1} \|\widehat{\phi}_h(s_h^k, a_h^k)\|_{\Lambda_{h,e}^{-1}}^2} \sqrt{\sum_{k=1}^{e-1} \mathbb{E}\left[\widetilde{\varepsilon}_k \mid \mathcal{H}_{k-1}\right]^2}$$

$$\leq \sqrt{d}\sqrt{(e-1)}\|\mathcal{V}_h\|_\infty \varepsilon_{ms}. \qquad \text{(by Lemma H.2)}$$

Combining these two bounds concludes the proof.

$\square$

**Lemma H.7** (Optimism) *Suppose ($\mathcal{E}_{model}$) holds. Let $\iota = \|\mathcal{V}_h\|_\infty \varepsilon_{ms}$. Then, for any episode $e = 1, 2, ..., N$, we have*

$$\forall h = 0, 1, ..., H-1 : \mathbb{E}_{\pi^\star}\left[\left(Q_h^\star(s_h, a_h) - \widehat{Q}_{h,e}(s_h, a_h)\right)\zeta_h(\tau_h)\right] \leq (H-h)\iota,$$

*and*

$$\forall h = 0, 1, ..., H-1 : \mathbb{E}_{\pi^\star}\left[\left(V_h^\star(s_h) - \widehat{V}_{h,e}(s_h)\right)\zeta_{h-1}(\tau_{h-1})\right] \leq (H-h)\iota,$$

*where*

$$\zeta_h(s_h) := \mathbb{I}\left[\widehat{Q}_{h,e}(s_h, \widehat{\pi}_h^e(s_h)) \leq M_V\right]$$

$$\zeta_h(\tau_h) = \prod_{h'=0}^{h} \zeta_{h'}(s_{h'}).$$

*Abusing notation, $\zeta_{-1}(\cdot)$ is the constant function 1.*

*In particular, we have that*

$$\mathbb{E}_{d_0}\left[V_0^\star(s_0) - \widehat{V}_{0,e}(s_0)\right] \leq H\iota.$$

*Proof.* Fix any episode $e$. We prove both claims via induction on $h = H, H-1, H-2..., 1, 0$. The base case holds trivially since $\widehat{V}_{H,e}$ and $V_H^\star$ are zero at every state by definition. Indeed, we have that for any $\pi$, including $\pi^\star$, that

$$\mathbb{E}_\pi\left[\left(P_{H-1}^\star(s_{H-1}, a_{H-1})(V_H^\star - \widehat{V}_{H,e})\right)\zeta_{H-1}(\tau_{H-1})\right]$$
$$= \mathbb{E}_\pi\left[(0-0)\zeta_{H-1}(\tau_{H-1})\right] = 0.$$

Now let's show the inductive step. Let $h \in \{H-1, H-2, ..., 1, 0\}$ be arbitrary and suppose the inductive hypothesis. So suppose that $V$-optimism holds at $h+1$ (we don't even need $Q$-optimism in the future), i.e.

$$\mathbb{E}_{\pi^\star}\left[\left(P_h^\star(s_h, a_h)(V_{h+1}^\star - \widehat{V}_{h+1,e})\right)\zeta_h(\tau_h)\right] = \mathbb{E}_{\pi^\star}\left[\left(V_{h+1}^\star(s_{h+1}) - \widehat{V}_{h+1,e}(s_{h+1})\right)\zeta_h(\tau_h)\right]$$
$$\leq (H-h-1)\iota \qquad \text{(IH)}$$

Recalling that $\widehat{Q}_{h,e}(s_h, a_h) = r_h(s_h, a_h) + \widehat{P}_{h,e}(s_h, a_h)\widehat{V}_{h+1,e} + \beta b_{h,e}(s_h, a_h)$, we have

$$\mathbb{E}_{\pi^\star}\left[\left(Q_h^\star(s_h, a_h) - \widehat{Q}_{h,e}(s_h, a_h)\right)\zeta_h(\tau_h)\right]$$
$$= \mathbb{E}_{\pi^\star}\left[\left(P_h^\star(s_h, a_h)V_{h+1}^\star - \widehat{P}_{h,e}(s_h, a_h)\widehat{V}_{h+1,e} - \beta b_{h,e}(s_h, a_h)\right)\zeta_h(\tau_h)\right]$$
$$\leq \mathbb{E}_{\pi^\star}\left[\left(\left(P_h^\star(s_h, a_h) - \widehat{P}_{h,e}(s_h, a_h)\right)\widehat{V}_{h+1,e} - \beta b_{h,e}(s_h, a_h)\right)\zeta_h(\tau_h)\right] + (H-h-1)\iota$$
$$\text{(by (IH))}$$
$$\leq \left|\mathbb{E}_{\pi^\star}\left[\left(\widehat{P}_{h,e}(s_h, a_h) - P_h^\star(s_h, a_h)\right)\widehat{V}_{h+1,e}\zeta_h(\tau_h)\right]\right| - \mathbb{E}_{\pi^\star}\left[\beta b_{h,e}(s_h, a_h)\zeta_h(\tau_h)\right] + (H-h-1)\iota$$
$$\leq \iota + (H-h-1)\iota = (H-h)\iota, \qquad \text{(by ($\mathcal{E}_{model}$) and $\widehat{V}_{h+1,e} \in \mathcal{V}_h$ (Lemma H.3))}$$

which proves the $Q$-optimism claim.

Now let's prove $V$-optimism.

$$\mathbb{E}_{\pi^\star}\left[\left(V_h^\star(s_h) - \widehat{V}_{h,e}(s_h)\right)\zeta_{h-1}(\tau_{h-1})\right]$$

$$
\begin{aligned}
&= \mathbb{E}_{\pi^\star} \left[ \left( Q_h^\star(s_h, a_h) - \left( \widehat{Q}_{h,e}(s_h, \widehat{\pi}_h^e(s_h)) \right)_{\leq M_V} \right) \zeta_{h-1}(\tau_{h-1}) \right] \\
&= \mathbb{E}_{\pi^\star} \left[ (Q_h^\star(s_h, a_h) - M_V) \zeta_{h-1}(\tau_{h-1})(1 - \zeta_h(s_h)) \right] \\
&\quad + \mathbb{E}_{\pi^\star} \left[ \left( Q_h^\star(s_h, a_h) - \widehat{Q}_{h,e}(s_h, \widehat{\pi}_h^e(s_h)) \right) \zeta_{h-1}(\tau_{h-1}) \zeta_h(s_h) \right] \\
&\leq \mathbb{E}_{\pi^\star} \left[ \left( Q_h^\star(s_h, a_h) - \widehat{Q}_{h,e}(s_h, \widehat{\pi}_h^e(s_h)) \right) \zeta_h(\tau_h) \right] \\
&\leq \mathbb{E}_{\pi^\star} \left[ \left( Q_h^\star(s_h, a_h) - \widehat{Q}_{h,e}(s_h, \pi_h^\star(s_h)) \right) \zeta_h(\tau_h) \right] \\
&\leq (H - h)\iota,
\end{aligned}
$$

by $Q$-optimism. $\qquad\square$

**Remark H.1** *We did not require $\widehat{P}_{h,e}$ to be a valid transition! It is in general unbounded and can even have negative entries!*

**Lemma H.8** (Simulation) *For any episode $e = 1, 2, ..., N$, we have*

$$
\mathbb{E}_{d_0} \left[ \widehat{V}_{0,e}(s_0) - V_0^{\pi^e}(s_0) \right] \leq \sum_{h=0}^{H-1} \mathbb{E}_{\widehat{\pi}^e} \left[ b_{h,e}(s_h, a_h) + (\widehat{P}_h(s_h, a_h) - P_h^\star(s_h, a_h)) \widehat{V}_{h+1,e} \right]
$$

*Proof.* We progressively unravel the left hand side. For any $s_0$,

$$
\begin{aligned}
&\widehat{V}_{0,e}(s_0) - V_0^{\pi^e}(s_0) \\
&\leq \widehat{Q}_{0,e}(s_0, \pi_0^e(s_0)) - Q_0^{\pi^e}(s_0, \pi_0^e(s_0)) \\
&= b_{0,e}(s_0, \pi_0^e(s)) + \left( \widehat{P}_{0,e}(s_0, \pi_0^e(s)) - P_0^\star(s_0, \pi_0^e(s_0)) \right) \widehat{V}_{1,e} + P_0^\star(s_0, \pi_0^e(s_0)) \left( \widehat{V}_{1,e} - V_1^{\pi^e} \right),
\end{aligned}
$$

where the inequality is due to the thresholding on the value function. Now, perform this recursively on the $P_0^\star(s_0, \pi_0^e(s_0)) \left( \widehat{V}_{1,e} - V_1^{\pi^e} \right)$ term. Doing this unravelling $h$ times gives the result. $\qquad\square$

**Theorem H.2** *Suppose Assumption H.1 and let $\beta$ and bonus be defined as in (7). Let $\delta \in (0, 1)$. Then w.p. at least $1 - \delta$, we have that the regret of LSVI is sublinear,*

$$
NV^\star - \sum_{e=0}^{N-1} V^{\widehat{\pi}^e} \leq \widetilde{\mathcal{O}} \left( dHNM_V \sqrt{\log(HN/\delta)} \varepsilon_{ms} + d^{1.5} H \sqrt{N} M_V M_\mu \log(dHN/\delta) \right)
$$

*where $\widetilde{\mathcal{O}}$ hides $\log$ dependence.*

*Proof.* We first condition on the high-probability event $(\mathcal{E}_{model})$, which occurs w.p. at least $1 - \delta$. Fix any arbitrary episode $e$. By optimism Lemma H.7 and the simulation lemma Lemma H.8,

$$
\begin{aligned}
\mathbb{E}_{d_0} \left[ V_0^\star(s_0) - V_0^{\widehat{\pi}^e}(s_0) \right] &\leq \mathbb{E}_{d_0} \left[ \widehat{V}_0^e(s_0) - V_0^{\widehat{\pi}^e}(s_0) \right] + H\iota \\
&\leq \sum_{h=0}^{H-1} \mathbb{E}_{\widehat{\pi}^e} \left[ \beta b_h^e(s_h, a_h) + \left( \widehat{P}_{h,e}(s_h, a_h) - P_h^\star(s_h, a_h) \right) \widehat{V}_{h+1,e} \right] + H\iota
\end{aligned}
$$

Applying $(\mathcal{E}_{model})$ with no indicators, i.e. $\zeta_h(\tau_h) = 1$ always, gives,

$$
\leq \sum_{h=0}^{H-1} \mathbb{E}_{\widehat{\pi}^e} \left[ 2\beta b_{h,e}(s_h, a_h) \right] + 2H\iota.
$$

Now, summing over $e = 1, 2, ..., N$, we have

$$
\sum_{e=1}^{N} \mathbb{E}_{d_0} \left[ V_0^\star(s_0) - V_0^{\widehat{\pi}^e}(s_0) \right]
$$

$$
\leq 2HN\iota + 2\beta \sum_{h=0}^{H-1} \sum_{e=1}^{N} \mathbb{E}_{\widehat{\pi}^e} \left[ b_{h,e}(s_h, a_h) \right]
$$

By Azuma's inequality applied to the martingale difference $\Delta_e = \mathbb{E}_{\widehat{\pi}^e}[b_{h,e}(s_h, a_h)] - b_{h,e}(s_h^e, a_h^e)$, which has envelope bounded by 2, implies w.p. $1 - \delta$,

$$\leq 2HN\iota + 2\beta \sum_{h=0}^{H-1}\sum_{e=1}^{N} b_{h,e}(s_h^e, a_h^e) + 4\beta\sqrt{N\log(HN/\delta)}.$$

It remains to bound the sum of expected bonuses. By Lemma H.2, we know that almost surely,

$$\sum_{h=0}^{H-1}\sum_{e=1}^{N} b_{h,e}(s_h^e, a_h^e) \leq H\sqrt{dN\log(N)}.$$

So, putting everything together,

$$\sum_{e=1}^{N} \mathbb{E}_{d_0}\left[V_0^\star(s_0) - V_0^{\widehat{\pi}^e}(s_0)\right]$$
$$\lesssim HN\iota + \beta H\sqrt{dN\log(N)} + \beta\sqrt{N\log(HN/\delta)}$$
$$\lesssim HN\iota + \left(\sqrt{dN}M_V\varepsilon_{ms} + M_V M_\mu d\sqrt{\log(dHN/\delta)}\right)\cdot H\sqrt{dN\log(HN/\delta)}$$
$$= HN\iota + dHNM_V\sqrt{\log(HN/\delta)}\varepsilon_{ms} + d^{1.5}H\sqrt{N}M_V M_\mu\log(dHN/\delta).$$

Note that $H\iota = H\|\mathcal{V}\|_\infty\varepsilon_{ms} = HM_V\varepsilon_{ms}$ is of lower order (with respect to $N$), we can simply drop it. This concludes the proof. $\square$

**Corollary H.1** *By setting $\delta = 1/N$, we have that expected regret also has the same rate as above.*

*Proof.* The expected regret by law of total probability, since regret is at most $NH$,
$$\mathbb{E}\left[\text{Reg}_N\right] \leq \mathbb{E}\left[\text{Reg}_N \mid (\mathcal{E}_{model})\right] + NH(1 - \mathbb{P}((\mathcal{E}_{model})))$$
$$\leq \mathbb{E}\left[\text{Reg}_N \mid (\mathcal{E}_{model})\right] + H.$$

Since $H$ is lower-order, we have the same rate. $\square$

## I    PROOF OF MAIN THEOREMS

First we prove Theorem 4.1 and Theorem 4.2.

**Theorem 4.1** (Regret under generative source access) *Suppose Assumptions 3.1-3.4 and 4.1, and suppose the input policies $\pi_k$ are $\lambda_{\min}$-exploratory. Then, for any $\delta \in (0,1)$, w.p. $1 - \delta$, REPTRANSFER when deployed in the target task has regret at most $\widetilde{\mathcal{O}}\left(\bar{\alpha}H^2 d^{1.5}\sqrt{T\log(1/\delta)}\right)$, with at most $Kn$ generative accesses per source task, with $n = \mathcal{O}\left(\lambda_{\min}^{-1}A\alpha_{\max}^3 KT\left(\log\frac{|\Phi|}{\delta} + K\log|\Upsilon|\right)\right)$.*

**Theorem 4.2** (Regret for REPTRANSFER with REWARDFREE subroutine) *Suppose Assumptions 3.1-3.4 and 4.1 hold, and let $\delta \in (0,1)$. Let $\{\pi_k\}_{k=1}^{K-1}$ be exploratory policies learned from running REWARDFREE on each source task with $N_{\text{LSVI-UCB}}$ and $N_{\text{REWARDFREE}}$ set as in Lemma 4.2. Then, w.p. $1 - \delta$, running REPTRANSFER with $\{\pi_k\}_{k=1}^{K-1}$ has regret in the target task of $\widetilde{\mathcal{O}}\left(\bar{\alpha}H^2 d^{1.5}\sqrt{T\log(1/\delta)}\right)$, with at most $\widetilde{\mathcal{O}}\left(A^4\alpha_{\max}^3 d^5 H^7 K^2 T\psi^{-2}\left(\log(|\Phi|/\delta) + K\log|\Upsilon|\right)\right)$ generative accesses per source task.*

*Proof of Theorem 4.1 and Theorem 4.2.* For the regret bound, set $M_V = H$ and $M_\mu = \bar{\alpha}$ and apply Theorem H.2. This choice of $M_\mu$ is valid by the argument following Assumption H.1. This gives us a regret bound of

$$\widetilde{\mathcal{O}}\left(dH^2 T\varepsilon_{ms} + \bar{\alpha}d^{1.5}H^2\sqrt{T}\log(1/\delta)\right),$$

where $\varepsilon_{ms}$ can be made smaller than $1/\sqrt{T}$, in which the second term dominates.

Now, we calculate the pre-training phase sample complexity in a source task.

First, let's calculate the reward-free model learning sample complexity, i.e. this is the number of samples required for learning $\widehat{P}_k$. Recall that we need this to be sufficiently large such that $\varepsilon_{TV} = 1/N_{\text{LSVI-UCB}}$. As required by Lemma 4.2, we need,

$$
\begin{aligned}
N_{\text{REWARDFREE}} &= \widetilde{\mathcal{O}} \left( A^3 d^4 H^6 \log \left( |\Phi||\Upsilon|/\delta \right) N_{\text{LSVI-UCB}}^2 \right) \\
&= \widetilde{\mathcal{O}} \left( A^3 d^4 H^6 \log \left( |\Phi||\Upsilon|/\delta \right) \left( A^3 d^6 H^8 \psi^{-2} \right)^2 \right) \\
&= \widetilde{\mathcal{O}} \left( A^9 d^{16} H^{22} \psi^{-4} \log \left( |\Phi||\Upsilon|/\delta \right) \right).
\end{aligned}
$$

Second, we calculate the cross-sampling sample complexity. Recall that $n$ is the number of samples in each pairwise dataset. In order to reduce $\epsilon_{ms}$ to $1/\sqrt{T}$, by Lemma 4.1, we need

$$
\begin{aligned}
1/\sqrt{T} \leq \varepsilon_{ms} &\leq \left( A \alpha_{\max}^3 K/\lambda_{\min} \right)^{1/2} \sqrt{\zeta_n} \\
&\leq \left( A \alpha_{\max}^3 K/\lambda_{\min} \right)^{1/2} \sqrt{\frac{1}{n} \left( \log \frac{|\Phi|}{\delta} + K \log |\Upsilon| \right)} \qquad \text{(by (2))}
\end{aligned}
$$

which implies that we need

$$
n \leq \lambda_{\min}^{-1} A \alpha_{\max}^3 KT \left( \log \frac{|\Phi|}{\delta} + K \log |\Upsilon| \right).
$$

Incorporating the coverage result from Lemma 4.2 gives,

$$
n \leq A^4 \alpha_{\max}^3 d^5 H^7 KT \psi^{-2} \left( \log \frac{|\Phi|}{\delta} + K \log |\Upsilon| \right).
$$

Since each task is in at most $K - 2$ pairwise datasets, each of size $n$, the total pre-training sample complexity per task is at most,

$$
\begin{aligned}
& N_{\text{REWARDFREE}} + (K - 2) \cdot n \\
&= \widetilde{\mathcal{O}} \left( A^9 d^{16} H^{22} \psi^{-4} \log \left( |\Phi||\Upsilon|/\delta \right) + A^4 \alpha_{\max}^3 d^5 H^7 K^2 T \psi^{-2} \left( \log \frac{|\Phi|}{\delta} + K \log |\Upsilon| \right) \right).
\end{aligned}
$$

$\square$

Now we prove Theorem 5.2, restated below.

**Theorem 5.2** (Regret with online access) *Suppose Assumptions 3.1-3.4,5.1,5.2 hold. W.p. $1 - \delta$, Algorithm 5 with appropriate parameters achieves a regret in the target $\widetilde{\mathcal{O}} \left( \bar{\alpha} d^{1.5} H^2 \sqrt{T \log(1/\delta)} \right)$, with at most* $\text{poly} \left( A, \alpha_{\max}, d, H, K, T, \psi^{-1}, \psi_{raw}^{-1}, \log(|\Phi||\Upsilon|/\delta) \right)$ *online queries in the source tasks.*

*Proof of Theorem 5.2.* We follows the same format as the proof of Theorem 4.1. The regret bound is identical.

Now let's compute the pre-training sample complexity. The regret bound requires us to set $\varepsilon_{ms} \leq 1/\sqrt{T}$. Here, our $\varepsilon_{ms}$ comes from Lemma G.1, so

$$
1/\sqrt{T} \leq \frac{\alpha_{\max} K^{1/2}}{(\psi_{raw} \lambda_{\min})^{1/2}} \sqrt{\frac{1}{n} \left( \log \frac{|\Phi|}{\delta} + K \log |\Upsilon| \right)},
$$

which implies we need

$$
n \leq \frac{\alpha_{\max}^2 K}{\psi_{raw} \lambda_{\min}} \left( \log \frac{|\Phi|}{\delta} + K \log |\Upsilon| \right).
$$

Plugging in the coverage of Lemma 4.2,

$$
\begin{aligned}
n &\leq \frac{\alpha_{\max}^2 KT}{\psi_{raw}} \left( \log \frac{|\Phi|}{\delta} + K \log |\Upsilon| \right) \left( A^{-3} d^{-5} H^{-7} \psi^2 \right)^{-1} \\
&\leq \frac{A^3 \alpha_{\max}^2 d^5 H^7 KT}{\psi_{raw} \psi^2} \left( \log \frac{|\Phi|}{\delta} + K \log |\Upsilon| \right).
\end{aligned}
$$

Here, we only collect one dataset, so the total pre-training sample complexity is

$$N_{\text{REWARDFREE}} + n$$
$$= \widetilde{\mathcal{O}} \left( A^9 d^{16} H^{22} \psi^{-4} \log\left(|\Phi||\Upsilon|/\delta\right) + A^3 \alpha_{\max}^2 d^5 H^7 KT \psi_{raw}^{-1} \psi^{-2} \left( \log \frac{|\Phi|}{\delta} + K \log|\Upsilon| \right) \right).$$

$\square$

## J  EXPERIMENT DETAILS

### J.1  CONSTRUCTION OF COMBLOCK

In this section we first introduce the vanilla Combination lock (Comblock) environment that is widely used as the benchmark for algorithms for Block MDPs. We provide a visualization of the comblock environment in Fig. 1(a). Concretely, the environment has a horizon $H$, and 3 latent states $z_{i;h}, i \in \{0, 1, 2\}$ for each timestep $h$ and 10 actions. Among the three latent states, we denote $z_0, z_1$ as the good states which leads to the final reward and $z_2$ as the bad states. At the beginning of the task, the environment will uniformly and independent sample 1 out of the 10 actions for each good state $z_{0;h}$ and $z_{1;h}$ for each timestep $h$, and we denote these actions $a_{0;h}, a_{1;h}$ as the optimal actions (corresponding to each latent state). These optimal actions, along with the task itself, determines the dynamics of the environment. At each good latent state $s_{0;h}$ or $s_{1;h}$, if the agent takes the correct action, the environment transits to the either good state at the next timestep (i.e., $s_{0;h+1}$, $s_{1;h+1}$) with equal probability. Otherwise, if the agent takes any 9 of the bad actions, the environment will transition to the bad state $s_{2;h+1}$ deterministicly, and the bad states transit to only bad states at the next timestep deterministicly. There are two situations where the agent receives a reward: one is uponing arriving the good states at the last timestep, the agent receives a reward of 1. The other is upon the first ever transition into the bad state, the agent receives an "anti-shaped" reward of 0.1 with probability 0.5. Such design makes greedy algorithms without strategic exploration such as policy optimization methods easily fail. For the initial state distribution, the environment starts in $s_{0;0}$ or $s_{1;0}$ with equal probability. The dimension of the observation is $2^{\lceil \log(H+|\mathcal{S}|+1) \rceil}$. For the emission distribution, given a latent state $s_{i;h}$, the observation is generated by first concatenate the one hot vectors of the state and the horizon, adding i.i.d. $\mathcal{N}(0,0.1)$ noise at each entry, appending 0 at the end if necessary. Then finally we apply a linear transformation on the observation with a Hadamard matrix. Note that without a good feature or strategic exploration, it takes $10^H$ actions to reach the final goal with random actions.

### J.2  CONSTRUCTION OF TRANSFER SETUP IN SECTION. 6

In this section we introduce the detailed construction of the experiment in Section. 6. For the source environment, we simply generate 5 random vanilla comblock environment described in Section.J.1. Note that in this way we ensure that the emission distribution shares across the sources, but the latent dynamics are different because the optimal actions are independently randomly selected. For the target environment, for each timestep $h$, we randomly acquire the optimal actions at $h$ from one of the sources and set it to be the optimal action of the target environment at timestep $h$, if the selected optimal actions are different for the two good states. Otherwise we keep sampling until they are different. Note that under such construction, since we fix the emission distribution, Assumption 3.4 is satisfied if we set $\alpha = 1$ for the source environment where we select the optimal action and $\alpha = 0$ for the other sources, at each timestep. To see how Assumption 5.1 is satisfied, recall that comblock environment naturally satisfies Assumption 3.3, and identical emission implies that the conditional ratio of all observations between source and target is 1.

### J.3  CONSTRUCTION OF TRANSFER SETUP IN SECTION. 6

Now we introduce the construction of the Comblock with Partitioned Observation (Comblock-PO) environment. Comparing with the vanilla comblock environment, the major difference is in

the observation space. In this setting, the size of the observation depends on the number of source environments $K$. Let the size of the original observation space be $O = |\mathcal{O}|$, the size of the observation for comblock-PO is $KO$. For the $k$-th source environment, where $k \in [K]$, the environment first generates the $O$-dimensional observation vector as in the original comblock, and then embed it to the $(k-1)O$-th to $kO$-th entries of the $KO$-dimensional observation vector, where it is 0 everywhere else. Thus we can see that the observation space for each source environment is disjoint (and thus the name partitioned observations). For the target enviornment, since the latent dynamcis are the same, we only need to design the emission distribution: for each latent state $s_{i;h}$, we assign the emission distribution uniformly at random from one of the sources.

### J.4    IMPLEMENTATION DETAILS

Our implementation builds on BRIEE (Zhang et al., 2022). In the Multi-task REPLEARN stage, we requires our learned feature to predict the Bellman backup of all the sources simultaneously. Therefore, in each iteration we have $k$ discriminators and $k$ sets of linear weights (instead of 1 in BRIEE), where $k$ is the number of source environments. For the deployment stage we implement LSVI following Algorithm 6.

To create the training dataset for Multi-task REPLEARN, for each $(i, j)$ environment pairs where $i \neq j$, we collect $500$ samples for each timestep $h$. For each $(i, i)$ environment pairs, we collect $500 \times (k-1) \times k$ samples for each timestep $h$, where $k$ denotes the number of sources. Thus we ensure that the number of samples from cross transition of different environments is the same as the number of samples from cross transition of the same environment. For the online setting, we simply sample $1000 \times (k-1) \times k$ samples for each $(i, i)$ cross transition to ensure that the total number of samples is the same for G-REPTRANSFER and O-REPTRANSFER.

To sample the initial state action pair (i.e., $(\tilde{s}, \tilde{a})$ pair as in (1)), for $90\%$ of the samples, we follow the final policy from each source environment trained using BRIEE. For the remaining $10\%$, we follow the same policy to state $\tilde{s}$, and then take a uniform random policy to get $\tilde{a}$. With this sampling scheme we ensure that Assumption 3.3 is satisfied. In the setting of Section. 6, we follow a more simple procedure to ensure that the samples are more balanced among the three states: we skip the first sampling step from environment $i$ (i.e., sample $s$ given $(\tilde{s}, \tilde{a})$), and simply reset environment $i$ to $s$, where $s$ is one of the three states with equal probability, and generate the observation accordingly. Note that such visitation distribution is also possible in the online setting with a more nuanced sampling procedure, and in the experiment we use the same sampling procedure for both G-REPTRANSFER and O-REPTRANSFER for a fair comparison.

### J.5    HYPERPARAMETERS

In this section, we record the hyperparameters we try and the final hyperparameter we use for each baselines. The hyperparameters for REPTRANSFER in Section. 6 is in Table. 3. The hyperparameters for REPTRANSFER in Section. 6 is in Table. 4. The hyperparameters for BRIEE is in Table. 5. We use the same set of hyperparameters for G-REPTRANSFER and O-REPTRANSFER.

Table 3: Hyperparameters for REPTRANSFER in Comblock.

| | Value Considered | Final Value |
|---|---|---|
| Decoder $\phi$ learning rate | {1e-2} | 1e-2 |
| Discriminator $f$ learning rate | {1e-2} | 1e-2 |
| Discriminator $f$ hidden layer size | {256} | 256 |
| RepLearn Iteration $T$ | {30} | 30 |
| Decoder $\phi$ number of gradient steps | {64} | 64 |
| Discriminator $f$ number of gradient steps | {64} | 64 |
| Decoder $\phi$ batch size | {256} | 256 |
| Discriminator $f$ batch size | {512} | 512 |
| RepLearn regularization coefficient $\lambda$ | {0.01} | 0.01 |
| Decoder $\phi$ softmax temperature | {1} | 1 |
| Decoder $\phi_0$ softmax temperature | {0.1} | 0.1 |
| LSVI bonus coefficient $\beta$ | $\{1, \frac{H}{5}\}$ | 1 |
| LSVI regularization coefficient $\lambda$ | {1} | 1 |
| Buffer size | {1e5} | 1e5 |
| Update frequency | {50} | 50 |
| Optimizer | {SGD} | SGD |

Table 4: Hyperparameters for REPTRANSFER in Comblock-PO.

| | Value Considered | Final Value |
|---|---|---|
| Decoder $\phi$ learning rate | {1e-2} | 1e-2 |
| Discriminator $f$ learning rate | {1e-2} | 1e-2 |
| Discriminator $f$ hidden layer size | {256,512} | 256 |
| Discriminator $f$ hidden layer number | {2,3} | 3 |
| RepLearn Iteration $T$ | {30,40,50,100,150} | 50 |
| Decoder $\phi$ number of gradient steps | {64,80,128,256} | 64 |
| Discriminator $f$ number of gradient steps | {64,80,128,256} | 64 |
| Decoder $\phi$ batch size | {256,512} | 512 |
| Discriminator $f$ batch size | {256,512} | 512 |
| RepLearn regularization coefficient $\lambda$ | {0.01} | 0.01 |
| Decoder $\phi$ softmax temperature | {1} | 1 |
| Decoder $\phi_0$ softmax temperature | {0.1,1} | 1 |
| LSVI bonus coefficient $\beta$ | $\{1, \frac{H}{5}\}$ | 1 |
| LSVI regularization coefficient $\lambda$ | {1} | 1 |
| Buffer size | {1e5} | 1e5 |
| Update frequency | {50} | 50 |
| Optimizer | {SGD, Adam} | Adam |

## J.6 VISUALIZATIONS

In this section we provide a comprehensive visualization of the decoders for all baselines in the target environment. We observe that the behaviors of all baselines are similar across the 5 random seeds. Thus to avoid redundancy, we only show the visualization from 1 random seed. We provide an example in Fig. 3 on how to interpret the visualization: let the emission function of the target environment be $o$, and let the decoder that we are evaluating be $\phi$, and to generate the blue block in Fig. 3, we sample 30 observations $\{s_n\}_{n=1}^{30}$ from the target environment at $z_{1,13}$, the latent state 1 (the title of the subplot) from timestep 13 (the x-axis). Concretely, $\{s_n\}_{n=1}^{30} \sim o(\cdot \mid z_{1,13})$. The blue block denotes the three-dimensional decoded latent states $\hat{z}$ from these 30 observations: $\hat{z} = \frac{1}{30} \sum_{n=1}^{30} \phi(s_n)$.

Table 5: Hyperparameters for BRIEE in Comblock and Comblock-PO.

|  | Value Considered | Final Value |
|---|---|---|
| Decoder $\phi$ learning rate | {1e-2} | 1e-2 |
| Discriminator $f$ learning rate | {1e-2} | 1e-2 |
| Discriminator $f$ hidden layer size | {256} | 256 |
| RepLearn Iteration $T$ | {30} | 30 |
| Decoder $\phi$ number of gradient steps | {64} | 64 |
| Discriminator $f$ number of gradient steps | {64} | 64 |
| Decoder $\phi$ batch size | {512} | 512 |
| Discriminator $f$ batch size | {512} | 512 |
| RepLearn regularization coefficient $\lambda$ | {0.01} | 0.01 |
| Decoder $\phi$ softmax temperature | {1} | 1 |
| Decoder $\phi_0$ softmax temperature | {0.1} | 0.1 |
| LSVI bonus coefficient $\beta$ | {$\frac{H}{5}$} | $\frac{H}{5}$ |
| LSVI regularization coefficient $\lambda$ | {1} | 1 |
| Buffer size | {1e5} | 1e5 |
| Update frequency | {50} | 50 |

### J.6.1 VISUALIZATIONS FROM SECTION. 6

We record the visualization of the 5 sources from Fig. 3 to Fig. 7; O-REPTRANSFER in Fig.8; G-REPTRANSFER in Fig. 9; running BRIEE on target in Fig. 10.

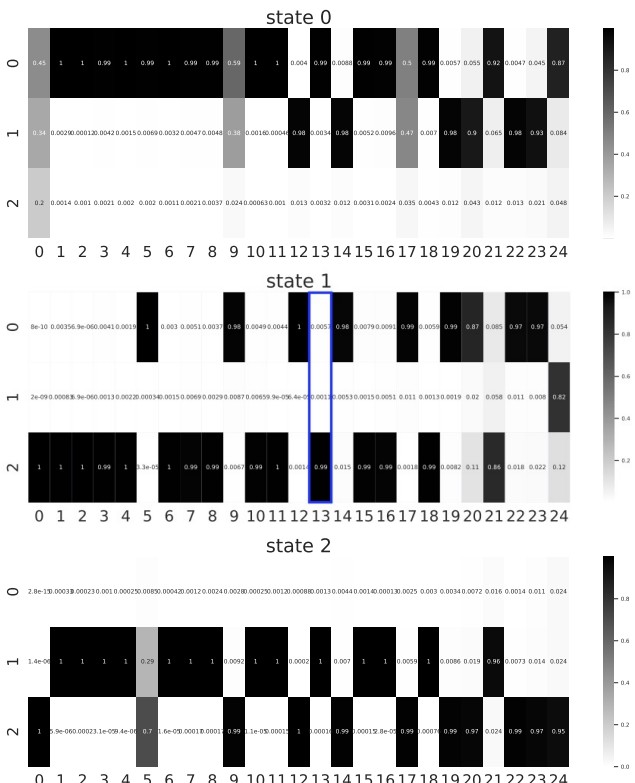

Figure 3: Visualization of decoders from source 1. Note the collapse happens at timestep 5, 9 and 17.

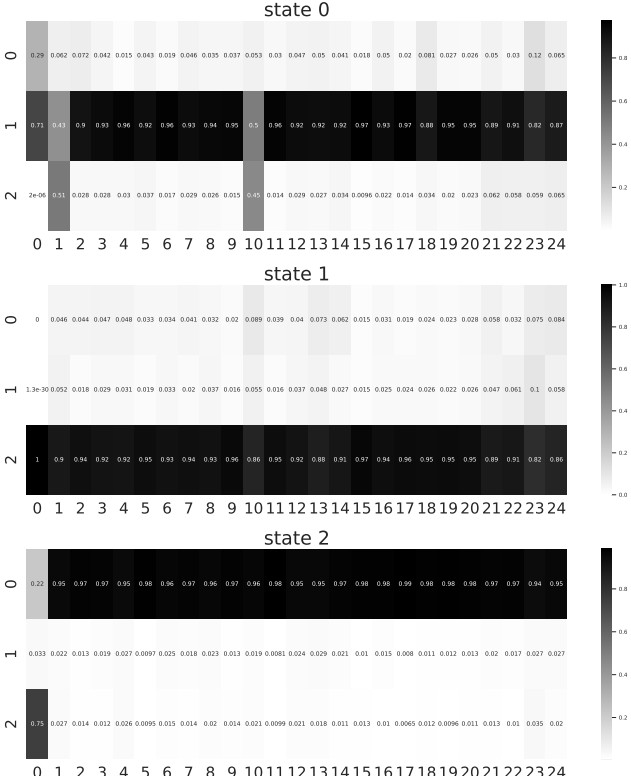

Figure 4: Visualization of decoders from source 2. Note the collapse happens at timestep 1 and 10.

## J.6.2 VISUALIZATIONS FROM SECTION. 6

We record the visualization of the 2 sources from Fig. 11 and Fig. 12; O-REPTRANSFER in Fig.13; G-REPTRANSFER in Fig. 14; running BRIEE on target in Fig. 15. Note that the features collapse at some timesteps in Fig. 14 and Fig. 15, but this is acceptable because the optimal actions at those timesteps are the same for the collapsed states.

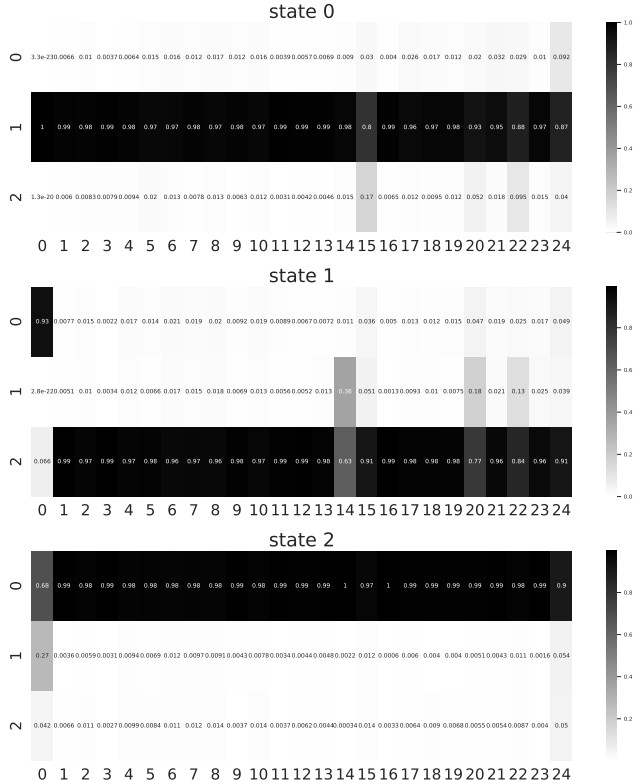

Figure 5: Visualization of decoders from source 3. Note the collapse happens at timestep 14 and 15.

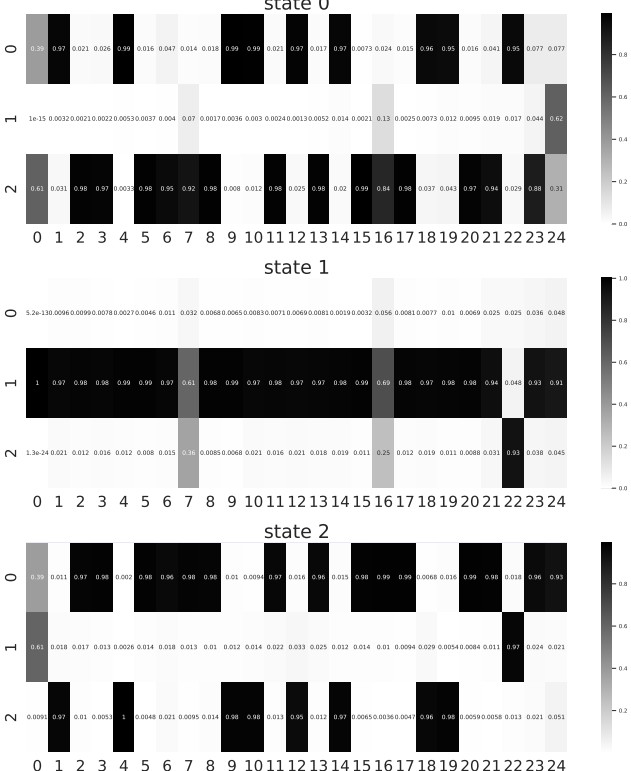

Figure 6: Visualization of decoders from source 4. Note the collapse happens at timestep 7, 16, 24.

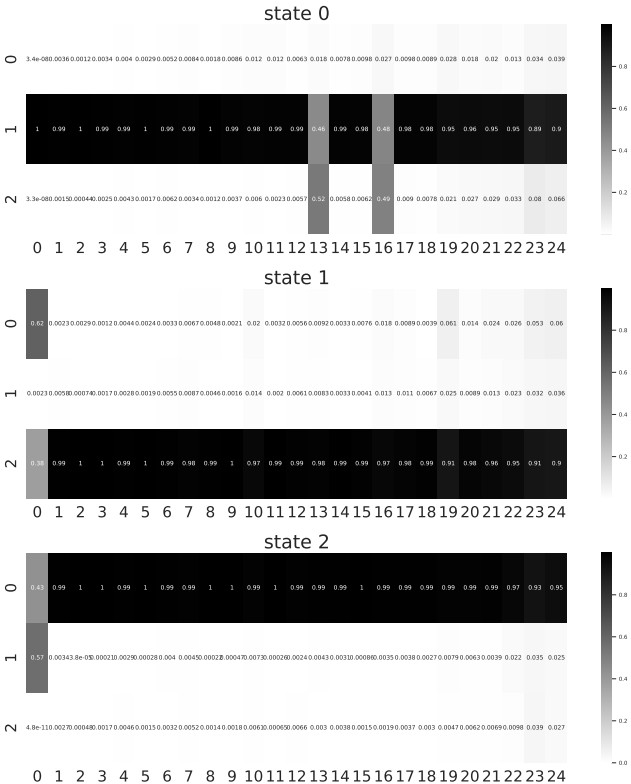

Figure 7: Visualization of decoders from source 3. Note the collapse happens at timestep 13 and 16.

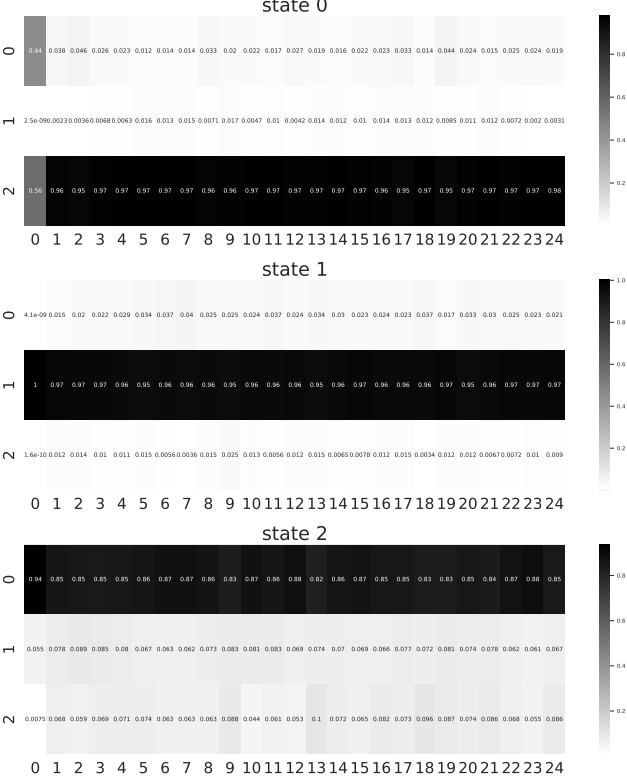

Figure 8: Visualization of decoders from O-REPTRANSFER

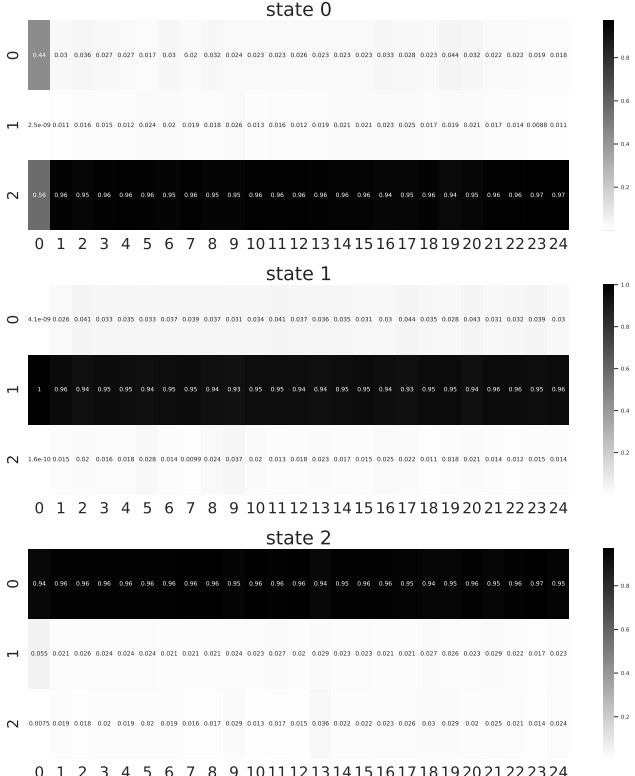

Figure 9: Visualization of decoders from G-REPTRANSFER

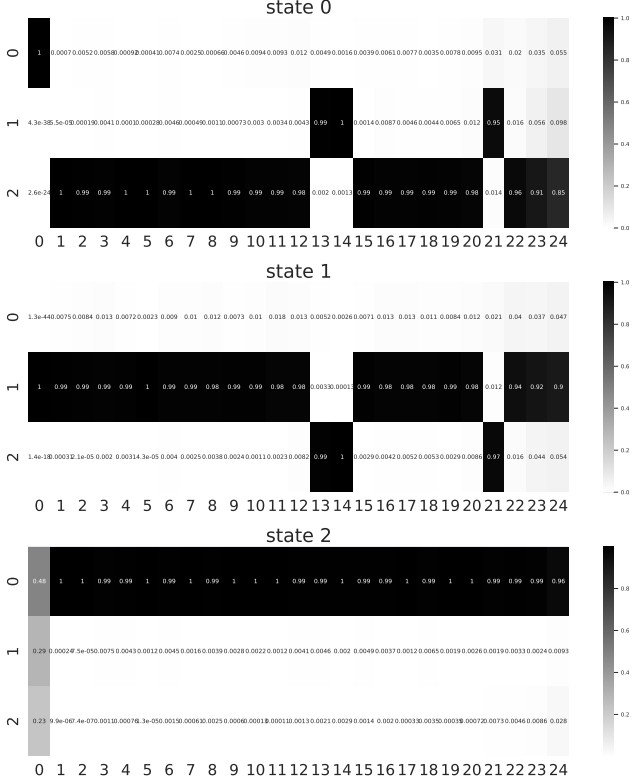

Figure 10: Visualization of decoders from running BRIEE on target.

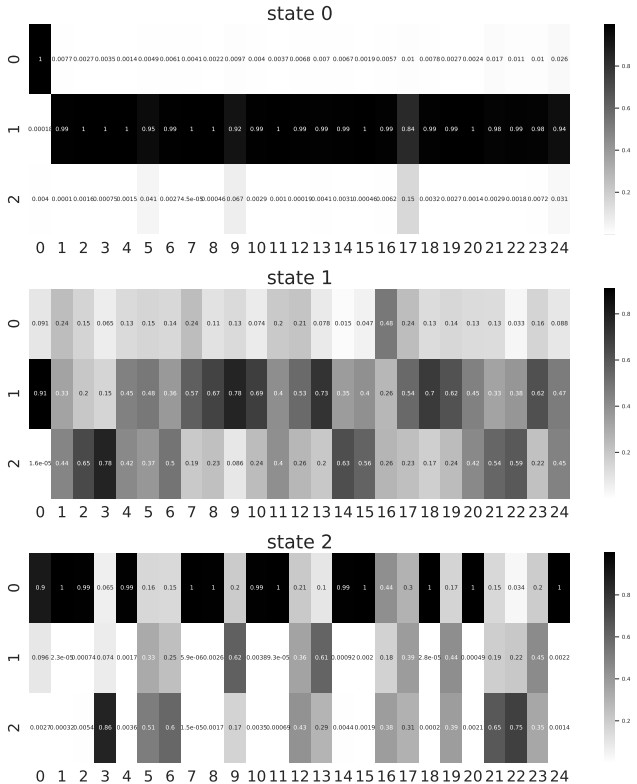

Figure 11: Visualization of decoders from source environment 1.

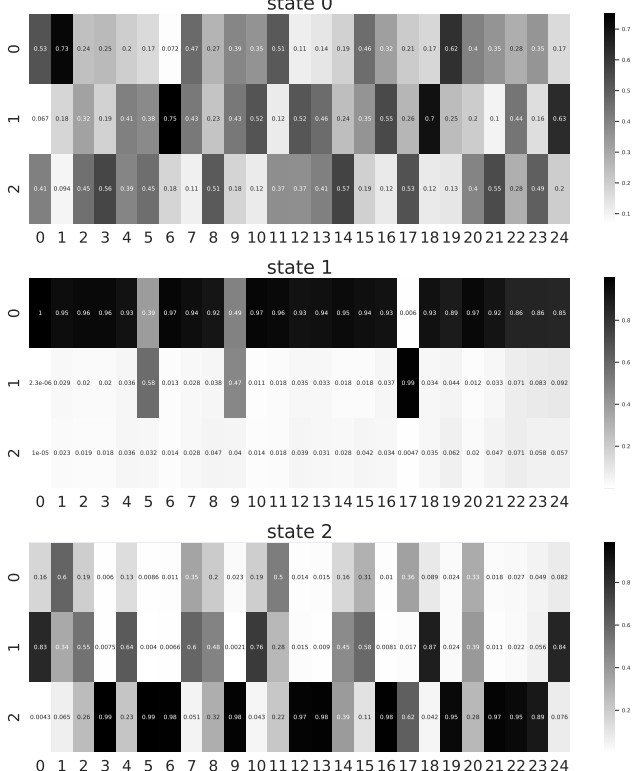

Figure 12: Visualization of decoders from source environment 2.

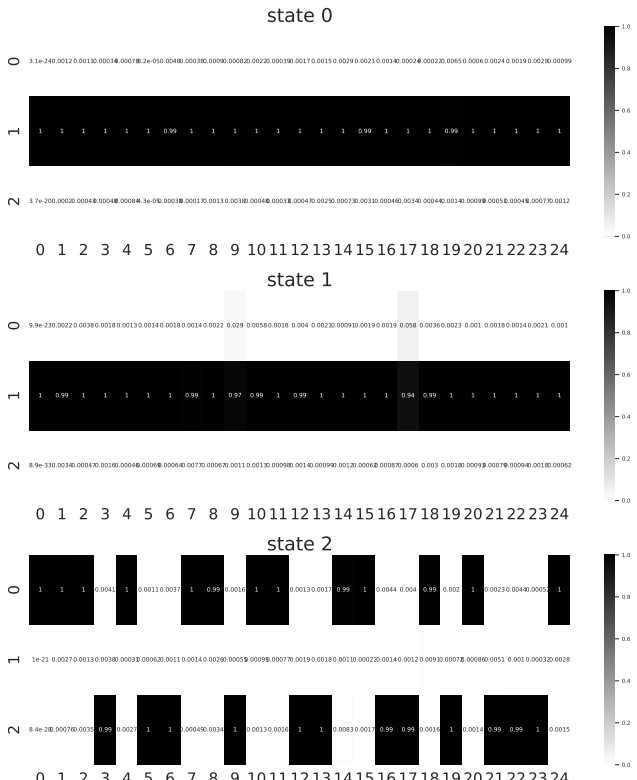

Figure 13: Visualization of decoders from O-REPTRANSFER.

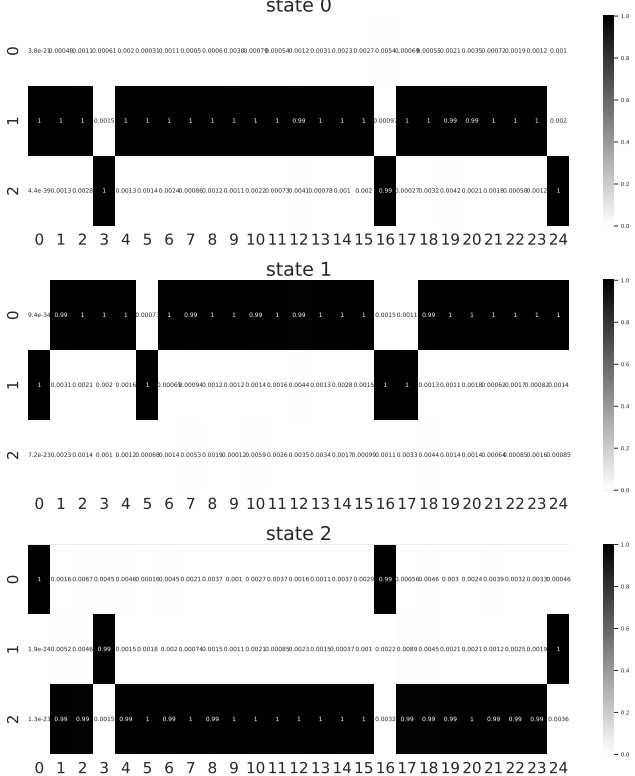

Figure 14: Visualization of decoders from G-REPTRANSFER.

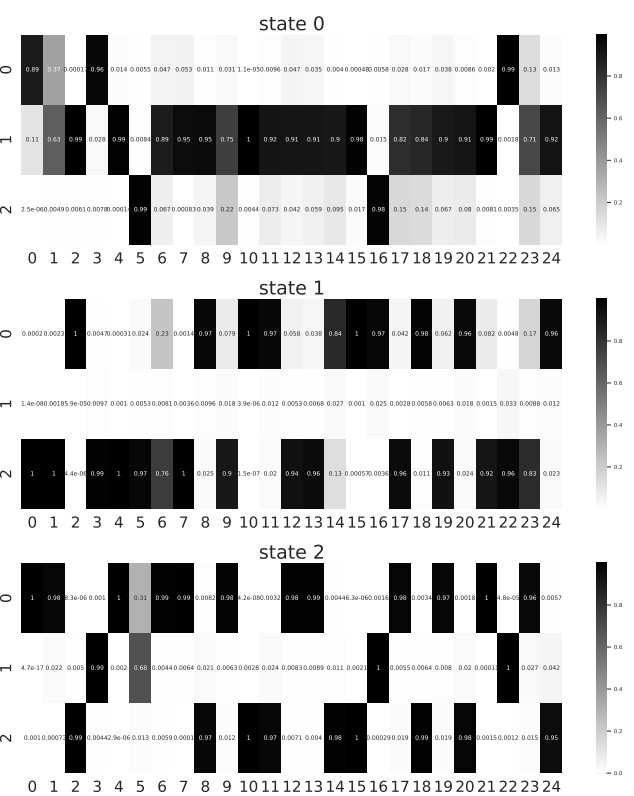

Figure 15: Visualization of decoders running BRIEE in the target.

