# OpenReview forum: "Provable Benefits of Representational Transfer in Reinforcement Learning"
_ICLR.cc/2023/Conference — Submitted to ICLR 2023_

### Official Review · Reviewer_F8xp · 2022-10-25

**Confidence:** 3
**Correctness:** 3
**Technical Novelty And Significance:** 3
**Empirical Novelty And Significance:** 2
**Recommendation:** 6

**Clarity, Quality, Novelty And Reproducibility:**

The paper is mostly of theoretical nature. It shows several interesting results, especially on the impossibility of transfer due to a permutation issue. The presentation is clear.

**Strength And Weaknesses:**

Strengths
- Strong theoretical analysis
- Clear writing

Weaknesses
- It remains unclear about the applicability of the results since one either has to sample from the transition kernel, or requires a stronger assumption. Both are not necessarily satisfied in many practical RL applications.
- The experiments are only conducted on the CombLock benchmark, which is limited. Besides, the experiment results in Fig.2 are hard to parse because the description of the setup (even after reading some of the appendices) is not clear enough.

Typo: In the “special case” after Assumption 3.4, there shouldn’t be summing over k for alphas and the sum for p_k should be over k instead of p.

**Summary Of The Paper:**

This work theoretically analyzes the applicability of feature/representation transfer from source tasks to a target task in RL, under the low-rank MDP setting (Def.3.1). It proposes an algorithm, RepTransfer (Algo.1), that learns the representations from cross-task samples, assuming one can sample from the transition kernel. It also shows that such cross-sampling is necessary (Thm.5.1) unless a stronger assumption is satisfied (Assumption 5.1). Experiments are conducted on the CombLock benchmark.

**Summary Of The Review:**

This paper provides several theoretical results for representational transfer in RL. Such results are important, but the applicability of the algorithm is not very clear, especially due to the limited experiments.

---

### Official Review · Reviewer_1Nh3 · 2022-10-28

**Confidence:** 3
**Correctness:** 3
**Technical Novelty And Significance:** 2
**Empirical Novelty And Significance:** 2
**Recommendation:** 3

**Clarity, Quality, Novelty And Reproducibility:**

Good language quality, bad paper organization and virtual length overflow. Please see main review for details.

**Strength And Weaknesses:**

### Strengths
Informative (although impossible to check in detail) bounds on the behaviour of the RepLearn and RepTransfer procedures which (seems to) permit principled transfer (exemplified on a limited example).

### Weaknesses
Disclaimer: my reviewing policy is always to try to find reasons to accept a paper (and not reasons to reject it), and I strongly believe we, as a community, should lay a kind look upon papers. This "weaknesses" section is long and took me a really long time to write. I hope it helps the authors.

My main concern is that the paper does not stand on its own. There are repeated references to elements of the Appendix. And these elements are not supplementary material, they are necessary to lay an informed eye on the main text's contents. This makes this paper rather difficult to access and to read, despite efforts made in the presentation.

Examples of references to the appendix: Algorithms 3 (RepLearn), 4 (Rep-UCB), 5 (RepTransfer), 6 (LSVI-UCB) are necessary to understand algorithms 1 and 2 or the related theorems from the main text, and they are all in the appendix. The short discussion on transfer in supervised learning (page 4) does not make sense without reading appendix C.

My point is that this should be a conference paper. Not a textbook. Even if all these algorithms are not new contributions per se, the authors cannot expect them to be common knowledge, even within the RL community. So such elements should be part of a background section in the main text, even if in a light form, so that the paper is readable and stand on its own feet without requiring to go back and forth to the 37 pages long appendix.

For the same reason (and because of the insanely short timespan left to ICLR reviewers to complete their reviews), I cannot guarantee a fair evaluation of the proofs and the theoretical statements. In particular, theorems 4.1 and 4.2 (and lemma 4.2) *seem* relevant but I cannot guarantee their soundness. Again: this is a conference paper, not a book.

I think the paper is missing important references and the authors only focus on the niche recent contributions in block MDPs and low rank MDPs. An important trend of research which learns task specific representations, and transfers them downstream to new tasks are progressive neural networks (Rusu et al, 2017). Similarly, the authors quote a number of reward-free exploration schemes (which are somewhat independent of representation learning per se) but dismiss the reward-free representation learning literature which includes at least successor representations (Dayan, 1993; Barreto et al., 2017; and many many others) and successor measures (Blier et al., 2021; Touati & Ollivier, 2021). Even if the paper considers transfer between different transition models, this literature is lacking in Sections 1 and 2 and this should be corrected, in particular given the strong connection with low-rank MDPs. Again, I am surprised to see no references to bisimulation metrics of MDP similarity metrics (e.g. Lecarpentier et al, 2021) when the authors talk about MDP similarity. Unfortunately, this confirms the impression that this paper is "low-rank mdp theoreticians talking to only a very few other low-rank mdp theoreticians" and does not consider the broader field of tranfer in RL (which is an effort other papers make). If that is the case, then this paper should go to a specialized workshop or journal or seminar on the topic (and this should be perfectly acceptable). Otherwise, it needs better foundations, motivation and connexions to work outside the comfort zone of the authors.
Rusu, A. A., Rabinowitz, N. C., Desjardins, G., Soyer, H., Kirkpatrick, J., Kavukcuoglu, K., ... & Hadsell, R. (2016). Progressive neural networks. NeurIPS deep learning symposium.
Dayan, P. (1993). Improving generalization for temporal difference learning: The successor representation. Neural computation, 5(4), 613-624.
Barreto, A., Dabney, W., Munos, R., Hunt, J. J., Schaul, T., van Hasselt, H. P., & Silver, D. (2017). Successor features for transfer in reinforcement learning. Advances in neural information processing systems, 30.
Blier, L., Tallec, C., & Ollivier, Y. (2021). Learning successor states and goal-dependent values: A mathematical viewpoint. arXiv preprint arXiv:2101.07123.
Touati, A., & Ollivier, Y. (2021). Learning one representation to optimize all rewards. Advances in Neural Information Processing Systems, 34, 13-23.
Lecarpentier, E., Abel, D., Asadi, K., Jinnai, Y., Rachelson, E., & Littman, M. L. (2021, May). Lipschitz lifelong reinforcement learning. In Proceedings of the AAAI Conference on Artificial Intelligence (Vol. 35, No. 9, pp. 8270-8278).

The motivation for the cross-sampling part (section 4) is very unclear. Although, after dedicating a long time to understanding it, cross sampling with two successive transitions appears like a reasonable way to feed RepLearn (which is not a contribution from this paper since it is already the work of Agarwal (2020)). But really, the rationale behind this practice is unclear unless one (again) refers to the appendix and looks at the proofs. Not a single line on the topic in the paper: this strongly reduces the value for readers.

Concerning the reward-free exploration part (section 4.1), LSVI-UCB seems a reasonable basis but it is not developped in the paper. Nor is its connection to recent work on reward-free exploration. E.g.:
Kaufmann, E., Ménard, P., Domingues, O. D., Jonsson, A., Leurent, E., & Valko, M. (2021, March). Adaptive reward-free exploration. In Algorithmic Learning Theory (pp. 865-891). PMLR.
Ménard, P., Domingues, O. D., Jonsson, A., Kaufmann, E., Leurent, E., & Valko, M. (2021, July). Fast active learning for pure exploration in reinforcement learning. In International Conference on Machine Learning (pp. 7599-7608). PMLR.
Badia, A. P., Sprechmann, P., Vitvitskyi, A., Guo, D., Piot, B., Kapturowski, S., ... & Blundell, C. (2020). Never give up: Learning directed exploration strategies. ICLR 2020.
Domingues, O. D., Tallec, C., Munos, R., & Valko, M. (2021, June). Density-Based Bonuses on Learned Representations for Reward-Free Exploration in Deep Reinforcement Learning. In ICML 2021 Workshop on Unsupervised Reinforcement Learning.
So overall, besides the "hey, this is an interesting setup upon which we can derive regret bounds" side, this looks like an odd assembly of good ideas, without discussion on the rationale and I question what will remain of this paper in a few years (and this is in no way a negative judgment on the work done, i'm really questioning the impact this presentation will have).

There is a subsection numbered 4.1 but no 4.2. This is very odd and does not help structuring the paper. Either make it a separate section, or introduce a 4.1 at the beginning of section 4.

In theorem 5.1, what is the "set of K-task multi-set"?
In theorem 5.1, the algorithm $\mathcal{A}$ is said to interact with the source tasks, but the word "online" is never mentioned. As far as I understand, this theorem is specific to online interaction (it's the very motivation for this theorem), so this should be made explicit, otherwise the statement make no sense (interaction with the source tasks could use a generative model as in section 4).

Theorem 5.2 directly refers (again) to algorithm 5 which is on page 28. No more comments on that.

There is no conclusion! Again: conference are not a patent registration desk for some new theorems. Papers should be reasonably accessible and feature hindsight views, perspectives, thoughtful discussion which leaves the reader bubbling with ideas and inspiration. It is not the case here. And although the work is of great quality, I strongly question its relevance for the reader.

Typos and phrasing:
Abstract: weird sentence "The sample complexity is close to knowing the ground truth features in the target task". Close to that of learning the representations when knowing...?
page 3, "transitions s_{h+1}", the "to" is missing.
page 3, \phi^* should be \phi^\star.
page 3, "Assumption 3.2 is standard realizability condition". is a standard
page 4, assumption 3.3 does not need the "with \alpha_{max}=..." part. Please make it separate, it's confusing otherwise.
page 7, extra space after "somewhat surprisingly".
page 8, which results the failures -> which results in failure

Notation:
P^\star_h is the transition model at step k, while P^\star_k is the full sequence of time-dependent transition kernels for task k. This is confusing. Why the \star everywhere?
Using \Upsilon for the set of \mu functions is confusing. Why not just M?

**Summary Of The Paper:**

This work considers the question of learning shared representations in low-rank MDPs, so as to permit transfer between tasks. It considers the problem of transfer with access to a generative model for the source tasks, and the online case separately.
Contributions: exploit RepLearn (Agarwal 2020) for transfer, provide regret bounds for separate parts of the algorithm (representation learning, SxA coverage, etc.).

**Summary Of The Review:**

Overall, this seems to be very solid work on deriving interesting bounds for re-using RepLearn in the context of low-rank feature transfer, with a limited experimental validation. But there is no take-away message for the reader and the paper is unreadable without the appendix. To me, that alone is a motive for rejection. I hope the detailed review will clarify this motivation.

Besides this, despite the interesting contents and the (apparent) correctness of the points developed, I question the impact this will have on the community. I think this paper needs profound restructuring to make it valuable to the reader and encourage the authors to make this effort for the community. Thus, as is, this paper is not suited for publication.

---

### Official Review · Reviewer_WTGs · 2022-11-01

**Confidence:** 3
**Correctness:** 3
**Technical Novelty And Significance:** 3
**Empirical Novelty And Significance:** 3
**Recommendation:** 6

**Clarity, Quality, Novelty And Reproducibility:**

$\textbf{Clarity: }$

The writing and presentation of the proposed method is almost clear. The organization of this paper is good. The assumptions along with the limitations are well discussed. The appendix provides sufficient details.

&nbsp;

$\textbf{Novelty: }$

Based on the new assumptions (i.e., mainly Assumption 3.3, 3.4) used in this work, significant and novel contributions are made in learning and transfering representation in Low-rank MDPs with relaxed assumptions.
Correspondingly, the experiments show empirical supports to the new theoretical results.

&nbsp;

$\textbf{Quality: }$

The theoretical results are presented with clear statements on assumptions. The experiments connect to the theoretical results closely and provide good empirical supports.

&nbsp;


$\textbf{Reproductibility:}$

Most experimental details are provided in the appendix. The source codes are also provided.


**Strength And Weaknesses:**

$\textbf{Strengths:}$
+ The paper is well written and organized. . The assumptions along with the limitations are well discussed. Although I am not an expert in low-rank MDP, the paper is still easy to follow.
+ The new theoretical scenario (i.e., Assumption 3.3 and 3.4) makes sense to me (although being kind of limited, I think it is a reasonable choice). The sample complexity under both access settings are discussed in an organized order. The proposed approach is clear and neat to me.
+ The related work and background are well enclosed and connected in the text.
+ The experiments connect to and support the theoretical results well.

&nbsp;

$\textbf{Weaknesses (and Questions): }$


What if we consider a variant of REPTRANSFER that still has the generative access but does not do cross sampling? Does this variant degenerate to the online access case in the pratical implementation? If not, maybe taking this variant as an additional baseline helps the presentation of the experiments.

&nbsp;

After introducing Assumption 4.1, the authors metion ‘Prior works …. or directly assume access to a diverse state-action distribution which provides coverage and from which one can sample’. I am wondering why such a prior assumption is considered to be (maybe) stronger than the generative access.


&nbsp;


For maybe a minor one, at the bottom of page 6, should it be ‘This ensures good exploration … in $P^{\star}_{k}$’ (rather than $K$)?





**Summary Of The Paper:**

This paper studies provably efficient representation transfer in Low-rank MDPs, where multiple source tasks are used to train a shared representation based on which a good policy in the target task is expected to learn efficiently. Different from prior works, this work proposes a new theoretical scenario based on an (arguably more relaxed) assumption on relatedness. In this scenario, this work proposes a novel approach, called REPTRANSFER, consisting of 1) a modular reward-free exploration method for the policies in source tasks that satisfy feature reachability, 2) MLE-based representation pre-training with cross sampling, and 3) representation transfer and policy learning in target task. The sample complexity of the proposed approach with generative access to source tasks or with only online access to source tasks are presented. The theoretical results mainly show the sample complexity is close to knowing the ground truth representation in the target task with the generative access while efficient representation transfer is impossible with only online access if no additional stronger assumption is used. The proposed approach is empirical evaluated in COMBLOCK benchmark, demonstrating the supriority of REPTRANSFER and supporting the theoretical results.

**Summary Of The Review:**

According to my detailed review above, I think this paper is above the acceptance threshold mainly due to the novel contribution made in theory and the high-quality presentation of the paper. Since I am not an expert in low-rank MDP theory, I make a conservative rating of 6 (marginally above the acceptance threshold).

---

### Author Response · Authors · 2022-11-09
**Summary of Revision**

We thank all the reviewers again for their effort in reviewing and sharing constructive reviews on our paper. Below we briefly summarize the changes in the revision.

- All typos brought up by the reviewers are fixed.
- We added additional descriptions/explanations in the main text about the algorithms that are deferred to the appendix, to further improve readability.
- We added an additional related work section in Appendix B discussing some of the representative empirical works on transfer learning and representation learning.

All changes are marked in $\textcolor{red}{\text{red}}$. We hope that our revision has addressed all the concerns from the reviewers.

---

> ### Author Response · Authors · 2022-11-14
> **Discussion Phase Ending Soon**
>
> Dear Reviewers,
>
> Thank you again for reviewing our paper. As the discussion phase is about to end in a few days, we'd like to ask one more time if there are further clarifications/modifications about our paper that you like us to address, or if any of the responses to your questions need further elaboration.
>
> If you believe that our response and revision have addressed your concerns properly, please consider increasing your score.
>
> Thank you,
> Paper 526 Authors

---

### Decision · Program_Chairs · 2023-01-20

**Decision:**

Reject

**Justification For Why Not Higher Score:**

The paper contains interesting results, but it needs significant re-organization of the material to make the paper more accessible.

**Justification For Why Not Lower Score:**

N/A

**Metareview: Summary, Strengths And Weaknesses:**

The paper studies the problem of learning a shared representation from a set of source tasks and then leveraging it to effectively solve a target task.
The authors theoretically and empirically study both the generative model and simulating model cases.
The reviewers agree that this is a solid paper with strong theoretical results.
On the other hand, the reviewers agree that the paper places too much content into the Appendix, making the paper not self-contained and making it difficult for the reader to extract the important messages from the paper.
The paper needs significant re-organization to fit a conference format, and, even if the authors have already made a step in the right direction, much work still needs to be done.
The reviewers also share concerns about the impact of this paper, which could be of interest only to a small subset of the ICLR audience.
We encourage the authors to consider the reviewers' suggestions in preparing a new version of their paper.